# UNLEASHING MASK: EXPLORE THE INTRINSIC OUT-OF-DISTRIBUTION DETECTION CAPABILITY

## ABSTRACT

Out-of-distribution (OOD) detection is an important aspect of safely deploying machine learning models in real-world applications. Previous approaches either design better scoring functions or utilize the knowledge of outliers to equip the well-trained models with the ability of OOD detection. However, few of them explore to excavate the intrinsic OOD detection capability of a given model. In this work, we discover the existence of an intermediate stage of a model trained on in-distribution data having higher OOD detection performance than that of its final stage across different settings and further identify the critical attribution to be learning with *atypical samples*. Based on such empirical insights, we propose a new method, Unleashing Mask (UM), that restores the OOD discriminative capabilities of the model. To be specific, we utilize the mask to figure out the memorized *atypical samples* and fine-tune the model to forget them. Extensive experiments have been conducted to characterize and verify the effectiveness of our method.

## 1 INTRODUCTION

Out-of-distribution (OOD) detection has drawn increasing attention when deploying machine learning models into the open-world scenarios (Nguyen et al., 2015; Lee et al., 2018a). Since the test samples can naturally arise from a label-different distribution, identifying OOD inputs is important, especially for those safety-critical applications like autonomous driving and medical intelligence. Previous studies focus on designing a series of scoring functions (Hendrycks & Gimpel, 2017b; Liang et al., 2018; Lee et al., 2018a; Liu et al., 2020; Sun et al., 2021; 2022) for OOD uncertainty estimation or finetuning with auxiliary outlier data to better distinguish the OOD inputs (Hendrycks et al., 2019c; Tack et al., 2020; Mohseni et al., 2020; Sehwag et al., 2021; Wei et al., 2022; Ming et al., 2022).

Despite the promising results achieved by previous methods (Hendrycks & Gimpel, 2017a; Hendrycks et al., 2019c; Liu et al., 2020; Ming et al., 2022), little attention is paid to considering whether the well-trained given model is the most appropriate for OOD detection. In general, models deployed for various applications have different targets (e.g., multi-class classification) (Goodfellow et al., 2016) instead of OOD detection (Nguyen et al., 2015; Lee et al., 2018a). However, most representative score functions, e.g., MSP (Hendrycks et al., 2019c), ODIN(Liang et al., 2018), and Energy (Liu et al., 2020), uniformly leverage the given models for OOD detection. Considering the target-oriented discrepancy, it arises a critical question: *does the well-trained given model have the optimal OOD detection capability?* If not, *how can we find a more appropriate model for OOD detection?*

In this work, we start by revealing an important observation (as illustrated in Figure 1), i.e., there exists a historical training stage where the model has a higher OOD detection performance than the final well-trained one. This is generally true across different OOD/ID datasets (Netzer et al., 2011; Van Horn et al., 2018; Cimpoi et al., 2014), learning rate schedules (Loshchilov & Hutter, 2017), and model structures (Huang et al., 2017; Zagoruyko & Komodakis, 2016). The empirical results of Figure 1 reflect the inconsistency between gaining better OOD detection capability (Nguyen et al., 2015) and pursuing better performance on ID data. We delve into the differences between the intermediate model and the final model by visualizing the misclassified examples. As shown in Figure 2, one possible attribution for covering the detection capability should be memorizing the *atypical samples* (at the semantic level) that are hard to learn for the model. Seeking zero error on those samples makes the model more confident on OOD data (see Figures 1(b) and 1(c)).

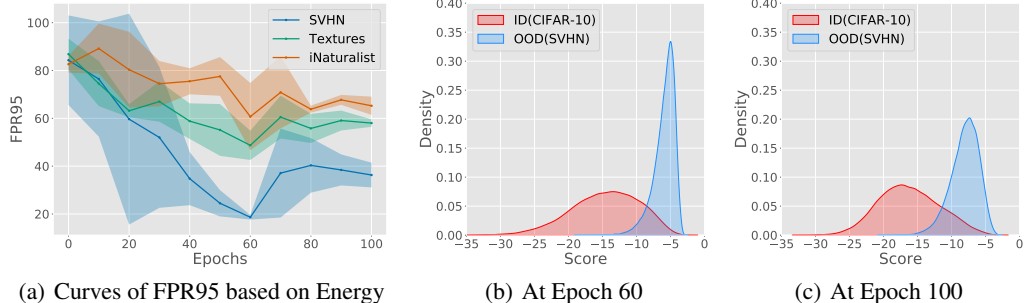

(a) Curves of FPR95 based on Energy     (b) At Epoch 60     (c) At Epoch 100

Figure 1: (a) the curves of FPR95 (false positive rate of OOD examples when the true positive rate of in-distribution examples is at 95%) based on the Energy score (Liu et al., 2020) across three different OOD datasets during the training on the CIFAR-10 dataset. (b) comparison between ID and OOD distribution at Epoch 60. (c) comparison between ID and OOD distribution at Epoch 100. All the experiments testing for OOD detection performance have been conducted multiple times. By backtracking the training phase, we can observe the existence of the model stage with better OOD detection capability using the Energy score to distinguish the OOD inputs. When zooming in the ID and OOD distributions at Epoch 60 and Epoch 100 respectively, it can be seen that, along with the training at the later stage, the overlap between them grows. Figure 2 contains further exploration.

The above analysis inspires us to propose a new strategy, namely, *Unleashing Mask* (UM), to excavate the once-covered detection capability of a well-trained given model by alleviating the memorization of those atypical samples (as illustrated in Figure 3) of ID data. In general, we aim to backtrack its previous stage with a better OOD detection capability. To achieve this target, there are two essential issues: (1) *the model that is well-trained on ID data has already memorized some atypical samples*; (2) *how to forget those memorized atypical samples considering the given model?* Accordingly, our proposed UM contains two parts utilizing different insights to address the above problems. First, as atypical samples are more sensitive to the change of model parameters, we initialize a mask with the specific cutting rate to mine these samples with constructed discrepancy. Then, with the loss reference estimated by the mask, we conduct the constrained gradient ascent (i.e., Eq. 3) for model forgetting. It will encourage the model to finally stabilize around the optimal stage. To avoid the severe sacrifice of the original task performance on ID data, we further propose UM Adopts Pruning (UMAP) which performs the tuning on the introduced mask with the newly designed objective.

For our proposed methods, we conduct extensive experiments to characterize and understand the working mechanism (in Section 4 and Appendix F). The comprehensive results accordingly demonstrate their effectiveness. We have verified the effectiveness of UM with a series of OOD detection benchmarks considering the two different ID datasets, i.e., CIFAR-10 and CIFAR-100. Under the various evaluation metrics, our UM, as well as UMAP, can indeed excavate the better OOD detection capability of given models and the averaged FPR95 can be reduced by a significant margin. Finally, a range of ablation studies and further discussions related to our proposed strategy are provided. We summarize our main contributions as follows,

- Conceptually, we explore the OOD detection performance via a new perspective, i.e., backtracking the model training phase without regularizing by any auxiliary outliers, different from most previous works that start with the well-trained model on ID data.

- Empirically, we reveal the potential detection capability of the well-trained model. We observe the general existence of an intermediate stage where the model has more appropriate discriminative features that can be utilized for OOD detection.

- Technically, we introduce a new strategy, i.e., *Unleashing Mask*, to excavate the once-covered OOD detection capability of a given model. By introducing the mask, we estimate the loss constraint for forgetting the atypical samples and empower the detection performance.

- Experimentally, we conduct extensive explorations to verify the general effectiveness on improving the OOD detection performance of our methods. Using various ID and OOD benchmarks, we provide comprehensive results across different setups and further discussion.

## 2 BACKGROUND

In this section, we briefly introduce the preliminaries and related work about OOD detection.

### 2.1 PRELIMINARIES

We consider multi-class classification as the training task, where $\mathcal{X} \subset \mathbb{R}^d$ denotes the input space and $\mathcal{Y} = \{1, \ldots, C\}$ denotes the label space. Given the model deployed in the real world, the reliable classifier is expected to figure out the OOD input, which can be formulated as a binary classification problem. Given $\mathcal{P}$, the distribution over $\mathcal{X} \times \mathcal{Y}$, we consider $D_{\text{in}}$ as the marginal distribution of $\mathcal{P}$ for $\mathcal{X}$, namely, the distribution of ID data. At test time, the environment can present a distribution $D_{\text{out}}$ over $\mathcal{X}$ of OOD data. In general, the OOD distribution $D_{\text{out}}$ is defined as an irrelevant distribution of which the label set has no intersection with $\mathcal{Y}$ Yang et al. (2021) and therefore should not be predicted by the model. The decision can be made with the threshold $\lambda$:

$$D_\lambda(x; f) = \begin{cases} \text{ID} & S(x) \geq \lambda \\ \text{OOD} & S(x) < \lambda \end{cases}, \tag{1}$$

Building upon the model $f \in \mathcal{H} : \mathcal{X} \to \mathbb{R}^c$ trained on ID data with the logit outputs, the goal of decision is to utilize the scoring function $S : \mathcal{X} \to \mathbb{R}$ to distinguish the inputs of $D_{\text{in}}$ from that of $D_{\text{out}}$ by $S(x)$. Typically, if the score value is larger than the threshold $\lambda$, the associated input $x$ is classified as ID and vice versa. We consider several representative scoring functions designed for OOD detection, e.g., MSP (Hendrycks & Gimpel, 2017b), ODIN (Liang et al., 2018), and Energy (Liu et al., 2020). More detailed definitions and implementation can refer to Appendix A.

To mitigate the issue of over-confident predictions for (Hendrycks & Gimpel, 2017b; Liu et al., 2020) some OOD data, recent works (Hendrycks et al., 2019c; Tack et al., 2020) utilize the auxiliary unlabeled dataset to regularize the model behavior. Among them, one representative baseline is outlier exposure (OE) (Hendrycks et al., 2019c). OE can further improve the detection performance by making the model $f(\cdot)$ finetuned from a surrogate OOD distribution $D_{\text{out}}^s$, and its corresponding learning objective is defined as follows,

$$\mathcal{L}_f = \mathbb{E}_{D_{\text{in}}} \left[ \ell_{\text{CE}}(f(x), y) \right] + \lambda \mathbb{E}_{D_{\text{out}}^s} \left[ \ell_{\text{OE}}(f(x)) \right], \tag{2}$$

where $\lambda$ is the balancing parameter, $\ell_{\text{CE}}(\cdot)$ is the Cross-Entropy (CE) loss, and $\ell_{\text{OE}}(\cdot)$ is the Kullback-Leibler divergence to the uniform distribution, which can be written as $\ell_{\text{OE}}(h(\boldsymbol{x})) = -\sum_k \texttt{softmax}_k\, f(x)/C$, where $\texttt{softmax}_k(\cdot)$ denotes the $k$-th element of a softmax output. The OE loss $\ell_{\text{OE}}(\cdot)$ is designed for model regularization, making the model learn from surrogate OOD inputs to return low-confident predictions. The general formulation of Eq 2 is also adopted in other related works for designing better tuning objectives that use different auxiliary outlier data.

Although previous works show promising results via designing scoring functions or regularizing models based on the model $f$ trained on ID data, few of them investigated the original detection capability of the well-trained given model. In this work, we introduce the layer-wise mask $m$ (Han et al., 2016; Ramanujan et al., 2020) to mine the atypical samples that memorized by the model. Accordingly, the decision can be written as $D(x; m \odot f)$, and the output of masked model is $m \odot f(x)$.

### 2.2 RELATED WORK

**Out-of-distribution Detection without auxiliary data.** (Hendrycks & Gimpel, 2017a) formally shed light on out-of-distribution detection, proposing to use softmax prediction probability as a baseline which is demonstrated to be unsuitable for OOD detection. Subsequent works keep focusing on designing post-hoc metrics to distinguish ID samples from OOD samples, among which ODIN (Liang et al., 2018) introduces small perturbations into input images to facilitate the separation of softmax score, Mahalanobis distance-based confidence score (Lee et al., 2018b) exploits the feature space by obtaining conditional Gaussian distributions, energy-based score (Liu et al., 2020) aligns better with the probability density. Besides designing score functions, many other works pay attention to various aspects to enhance the OOD detection such that LogitNorm (Wei et al., 2022) produces confidence scores by training with a constant vector norm on the logits, and DICE (Sun & Li, 2022) reduces the variance of the output distribution by leveraging the sparsification of the model.

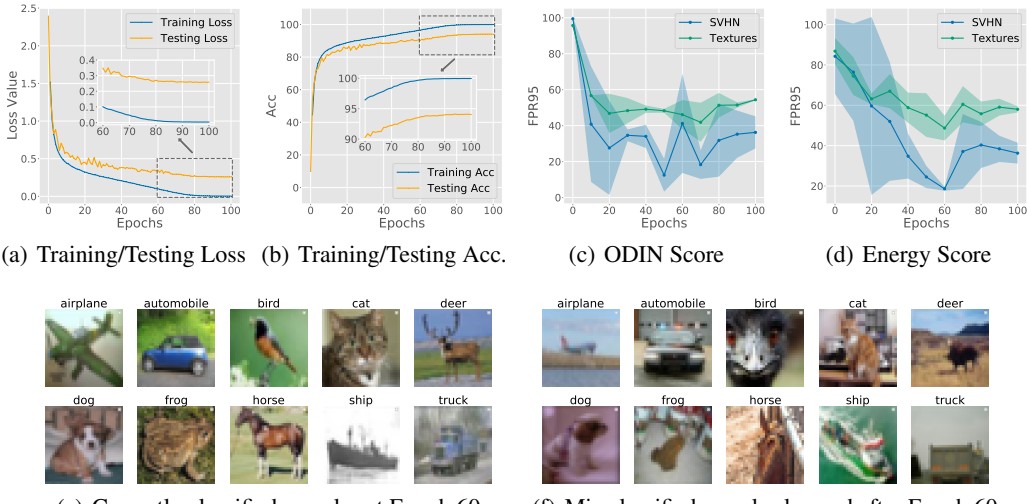

(a) Training/Testing Loss  (b) Training/Testing Acc.  (c) ODIN Score  (d) Energy Score

(e) Correctly classified samples at Epoch 60  (f) Mis-classified samples learned after Epoch 60

Figure 2: We train on `CIFAR-10` for the original multi-class classification and check the details as follows: (a) training/testing loss on ID data; (b) training/testing accuracy on ID data; (c) curves of FPR95 based on ODIN score; (d) curves of FPR95 based on Energy score; (e) visualization of correct classified samples at Epoch 60; (f) visualization of misclassified samples at Epoch 60. We investigate the once-covered OOD detection capability by checking the model behavior during its training phase. We take a closer look at the corresponding training and testing performance with the OOD detection capability indicated by two different scores. Through comparison, we find that achieving a reasonably small loss value (at round Epoch 60) on ID data is enough for OOD detection. However, continually optimizing on those atypical samples can impair the detection performance.

**Out-of-distribution Detection with auxiliary data.** Another promising direction towards OOD detection involves the auxiliary outliers for model regularization. On one hand, some works generate virtual outliers such that Lee et al. (2018a) uses generative adversarial networks to generate boundary samples, VOS (Du et al., 2022a) regularizes the decision boundary by adaptively sampling virtual outliers from the low-likelihood region. On the other hand, other works tend to exploit information from natural outliers, such that outlier exposure (OE) is introduced by Hendrycks et al. (2019b), given that diverse data are available in enormous quantities. (Yu & Aizawa, 2019) train an additional "head" and maximizes the discrepancy of decision boundaries of the two heads to detect OOD samples. Energy-bounded learning (Liu et al., 2020) fine-tunes the neural network to widen the energy gap by adding an energy loss term to the objective. Some other works also highlight the sampling strategy, such that ATOM (Chen et al., 2021) greedily utilizes informative auxiliary data to tighten the decision boundary for OOD detection, and POEM (Ming et al., 2022) adopts Thompson sampling to contour the decision boundary precisely. The performance of training with outliers is usually superior to that without outliers, shown in many recent works (Mohseni et al., 2020; Liu et al., 2020; Fort et al., 2021; Sun et al., 2021; Sehwag et al., 2021; Yang et al., 2021; Chen et al., 2021; Salehi et al., 2021).

## 3 PROPOSED METHOD: UNLEASHING MASK

In this section, we introduce our new method, i.e., *Unleashing Mask* (UM), to reveal the potential OOD detection capability of the well-trained model. First, we present and discuss the important observation that inspires our methods (Section 3.1). Second, we provide the insights behind the two critical parts of our UM (Section 3.2). Lastly, we describe the framework and the learning objective of UM that incorporates the previous component, as well as its variant, i.e., UMAP (Section 3.3).

### 3.1 ONCE-COVERED OOD DETECTION CAPABILITY

First, we present the phenomenon of the inconsistent trend between a better OOD detection capability and smaller training error during training. Empirically, as shown in Figure 1, we plot the OOD detection performance during the model training after multiple runs of the experiments. Across three

different OOD testing datasets, we can observe the existence of a better detection capability using the index of FPR95 metric based on the Energy (Liu et al., 2020) or ODIN (Liang et al., 2018) score. The generality has also been verified using different learning schedules and model structures in our experimental sections. We further show the comparison of the ID/OOD distributions in Figures 1(b) and 1(c). To be specific, the statics of the two distributions indicate that the gap between the ID and OOD data gets narrow as their overlap grows along with the training. After Epoch 60, although the model becomes more confident on ID data which satisfies a part of the calibration target (Hendrycks et al., 2019a), its predictions on the OOD data also become more confident which is unexpected. Without seeing any auxiliary outliers, it motivates us to explore how the model achieves that.

We take a closer look at the model behaviors in Figure 2, where we check its corresponding training/testing loss and accuracy. We find that the training loss has reached a reasonably small value at Epoch 60 where its detection performance also achieves a satisfactory level. However, if we further minimize the training loss, the trend of the FPR95 curve shows almost opposite directions with both training and testing loss or accuracy (see Figures 1(a), 2(b), and 2(c)). Accordingly, we extract those samples that were learned by the model at this period. As shown in Figures 2(e) and 2(f), the misclassified samples learned after Epoch 60 present much atypical semantic features. As deep neural networks tend to first learn the data with typical features (Arpit et al., 2017), we attribute the inconsistent trend to memorizing those atypical data at the later stage. In Appendix C, we provide a detailed discussion between it with the concept of conventional overfitting (Goodfellow et al., 2016).

## 3.2 UNLEASHING THE INTRINSIC DETECTION POWER

In general, the models that are developed for the original classification tasks are always seeking better performance (i.e., higher testing accuracy and lower training loss) in practice. However, the inconsistent trend revealed before indicates that the intrinsic OOD detection capability maybe once-covered during the training. It gives us a chance to unleash the potential detection power only considering the ID data in training. To this end, we have two important issues that need to address: (1) *the model that is well-trained on ID data may have already memorized some atypical samples which can not be figured out*; (2) *how to forget those atypical samples considering the given model?*

**Mining the atypical samples with constructed discrepancy.** According to Figures 2(a) and 2(b), both training accuracy and loss provide limited information that can differentiate the typical and atypical data. Inspired by the learning dynamics (Goodfellow et al., 2016; Arpit et al., 2017) of deep neural networks and the pathway conjecture (Barham et al., 2022) for inference, we try to manually construct the parameter discrepancy to mine the atypical samples from a well-trained model. To be specific, we introduce a novel layer-wise mask to achieve the goal. The masks are applied to all layers, which is consistent with the mask generation in the conventional pruning pipeline (Han et al., 2016). In Figure 3(c), we provide empirical evidence to show that we can figure out the atypical samples via enlarging the mask rate. Utilizing the masked output for loss computation, the atypical samples can be better differentiated. We also provide more discussion about the intuition in Appendix B.

**Forgetting the atypical samples with gradient ascent.** As the training loss achieves zero at the final stage of the given model, we need extra optimization signals to forget those memorized atypical samples. Considering the previous consistent trend before the potential optimal stage (e.g., before Epoch 60 in Figure 1(a)), the optimization signal also needs to control the model update not to be too greedy to drop the discriminative features for OOD detection. Starting with the given model, we can employ the gradient ascent (Sorg et al., 2010; Ishida et al., 2020) to forget the targeted samples, while the tuning phase should also prevent further updates if the model can achieve the expected stage.

## 3.3 METHOD REALIZATION

Based on previous insights, we present our overall framework as well as the learning objective of the proposed *Unleashing Mask* for OOD detection. Lastly, we discuss its compatibility with either the fundamental scoring functions or the outlier exposure approaches utilizing auxiliary outlier data.

**Framework.** As illustrated in Figure 3(a), our framework consists of two critical components for uncovering the intrinsic OOD detection capability: (1) the initialized mask with a specific pruning rate for constructing the output discrepancy with the original model; (2) the fine-tuning procedure for

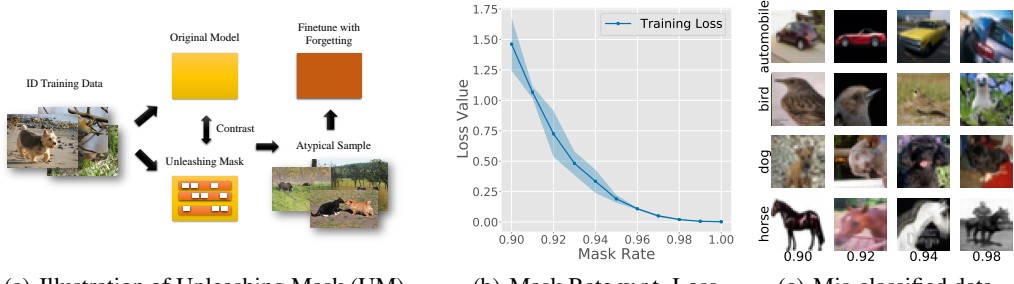

(a) Illustration of Unleashing Mask (UM)  (b) Mask Rate w.r.t. Loss  (c) Mis-classified data

Figure 3: (a) a brief illustration of the proposed Unleashing Mask (UM); (b) the mask rate w.r.t. loss value using the masked outputs; (c) examples of misclassified samples after masking the original well-trained model. As for our framework, given a well-trained model, we initialize an extra mask for mining the atypical samples that are sensitive to the changes in model parameters. Then we fine-tune the original model or adopt pruning with the estimated forgetting threshold, i.e., the loss value estimated by the UM. The final model can serve as the base of various score functions to utilize the discriminative features and also as the new initialization of fine-tuning with the auxiliary outliers.

alleviating the memorization of atypical samples. The overall workflow starts with obtaining the loss value of misclassifying those atypical samples and then conducts tuning with the model to forget.

**Objective for forgetting.**  Based on our framework, we introduce the forgetting objective as,

$$\min \mathcal{L}_{\text{UM}} = \min_{m_\delta \in [0,1]^n} |\ell_{\text{CE}}(f) - \widehat{\ell}_{\text{CE}}(m_\delta \odot f^*)| + \widehat{\ell}_{\text{CE}}(m_\delta \odot f^*), \tag{3}$$

where $m_\delta$ is our proposed layer-wise mask with the pruning rate $\delta$, $\ell_{\text{CE}}$ is the CE loss, $\widehat{\ell}_{\text{CE}}$ is the averaged CE loss over the ID training data, $|\cdot|$ indicates the computation for absolute value and $m_\delta \odot f^*$ denotes the masked output of the fixed pretrained model that is used to estimate the loss constraint for the learning objective of forgetting, which would be a constant value during the whole finetuning process. Concretely, the well-trained model will start to optimize itself again if it memorized the atypical samples and achieved almost zero loss value. We provide a positive gradient signal when the current loss value is lower than the estimated one and vice versa. The model is expected to finally stabilize around the stage that can forget those atypical samples.

**Unleashing Mask Adopts Pruning (UMAP).**  Considering the potential negative effect on the original task performance when conducting tuning for forgetting, we further propose a variant of UM Adopts Pruning, i.e., UMAP, to conduct tuning based on the masked output (e.g., replace $\ell_{\text{CE}}(f)$ to $\ell_{\text{CE}}(\hat{m}_p \odot f^*)$ in Eq 3) using the different mask $\hat{m}_p$ with its pruning rate $p$ as follows,

$$\min \mathcal{L}_{\text{UMAP}} = \min_{\hat{m}_p \in [0,1]^n, m_\delta \in [0,1]^n} |\ell_{\text{CE}}(\hat{m}_p \odot f^*) - \widehat{\ell}_{\text{CE}}(m_\delta \odot f^*)| + \widehat{\ell}_{\text{CE}}(m_\delta \odot f^*), \tag{4}$$

Different from the objective of UM (i.e., Eq 3) that minimizes the loss value over the model parameter, the objective of UMAP minimizes the loss over the mask to achieve the target of forgetting those atypical samples. UMAP provides an extra mask to restore the detection capacity but doesn't affect the model parameter for the inference on original tasks, indicating that UMAP is a more practical choice in real-world applications (as empirically verified in our experiments like Table 1). We summarize the algorithms of UM (in Algorithm 1) and UMAP (in Algorithm 2) in Appendix D.

**Compatible to other methods.**  As we explore the original OOD detection capability of the well-trained model, it is orthogonal and compatible with those promising methods that equip the given model with better detection ability. To be specific, through our proposed methods, we reveal the once-covered OOD detection capability via tuning the original model towards its intermediate training stage. The discriminative feature learned at that stage can be utilized by different scoring functions (Huang et al., 2021; Sun & Li, 2022; Wei et al., 2022), like ODIN (Liang et al., 2018) adopted in Figure 2(c). For those methods (Hendrycks et al., 2019a; Liu et al., 2020; Ming et al., 2022) utilizing the auxiliary outliers to regularize the model, our fine-tuned model obtained by UM and UMAP can also serve as their starting point or adjustment. As our strategy does not require any auxiliary outlier data to be involved in training, adjusting the model using ID data during its developing phase is practical.

## 4 EXPERIMENTS

In this section, we present the performance comparison of the proposed method in the OOD detection scenario. Specifically, we verify the effectiveness of our UM and UMAP with two mainstreams of OOD detection approaches: (i) fundamental scoring function methods; (ii) outlier exposure methods involving auxiliary samples. To better understand and characterize our proposed method, we further conduct extensive explorations on the ablation study and provide the corresponding discussion on each sub-aspect considered in our work. More details and additional results can also refer to Appendix F.

### 4.1 EXPERIMENTAL SETUPS

**Datasets.** Following the common benchmarks used in previous work (Liu et al., 2020; Ming et al., 2022), we adopt `CIFAR-10`, `CIFAR-100` (Krizhevsky, 2009) as our ID datasets. We use a series of different image datasets as the OOD datasets, namely `Textures` (Cimpoi et al., 2014), `Places365` (Zhou et al., 2017), `SUN` (Xiao et al., 2010), `LSUN` (Yu et al., 2015), and `iNaturalist` (Van Horn et al., 2018). We also use the other ID dataset as OOD dataset when training on a specific ID dataset, given that none of them shares any same classes (Yang et al., 2021). e.g. we treat `CIFAR-100` as the OOD dataset when training on `CIFAR-10` in our experiments for comparison.

**Training details.** We conduct all major experiments on DenseNet-101 (Huang et al., 2017) with training epochs fixed to 100. We also include experiment results on other types of models in the Appendix F. The models are trained using stochastic gradient descent (Kiefer & Wolfowitz, 1952) with Nesterov momentum (Duchi et al., 2011). We adopt Cosine Annealing (Loshchilov & Hutter, 2017) to schedule the learning rate which begins at $0.1$. We set the momentum and weight decay to be $0.9$ and $10^{-4}$ respectively throughout all experiments. The size of the mini-batch is $64$ for both ID samples (when training and testing) and OOD samples (when testing). More details and further discussion about choosing the mask ratio in experiments can be referred to at the end of Appendix F.

**Evaluation metrics.** We employ the following three common metrics to evaluate the performance of OOD detection: (i) Area Under the Receiver Operating Characteristic curve (AUROC) (Davis & Goadrich, 2006) can be interpreted as the probability for a positive sample to have a higher discriminating score than a negative sample (Fawcett, 2006); (ii) Area Under the Precision-Recall curve (AUPR) (Manning & Schütze, 1999) is an ideal metric to adjust the extreme difference between positive and negative base rates; (iii) False Positive Rate (FPR) at $95\%$ True Positive Rate (TPR) (Liang et al., 2018) indicates the probability for a negative sample to be misclassified as positive when the true positive rate is at $95\%$. We also include in-distribution testing accuracy (ID-ACC) to reflect the preservation level of the performance for the original classification task on ID data.

**OOD detection baselines.** We compare the proposed method with several competitive baselines in the two directions. Specifically, we adopt Maximum Softmax Probability (MSP) (Hendrycks & Gimpel, 2017a), ODIN (Liang et al., 2018), Mahalanobis score (Lee et al., 2018b), and Energy score (Liu et al., 2020) as scoring function baselines; We adopt OE (Hendrycks et al., 2019b), Energy-bounded learning (Liu et al., 2020), and POEM (Ming et al., 2022) as baselines with outliers. For all scoring function methods, we assume the accessibility of well-trained models. For all methods involving outliers, we constrain all major experiments to a fine-tuning scenario, which is more practical in real cases. Different from training a dual-task model at the very beginning, equipping deployed models with OOD Detection ability is a much more common circumstance, considering the millions of existing and running deep learning systems. We leave more details in Appendix A.

### 4.2 PERFORMANCE COMPARISON

In this part, we present the performance comparison with some representative baseline methods to demonstrate the effectiveness of our UM and UMAP. Our proposed UM is designed for excavating the potential OOD detection capability of the given model. Here we consider several scoring functions to compare the detection performance, and also some outlier exposure methods to further regularize the given model and boost the OOD detection ability. In each category, we choose one with the best detection performance to adopt UM/UMAP and check the detection results with the ID-ACC.

Table 1: Main Results (%). Comparison with competitive OOD detection baselines.

| $D_{in}$ | Method | AUROC↑ | AUPR↑ | FPR95↓ | ID-ACC↑ | w./w.o $D_{aux}$ |
|---|---|---|---|---|---|---|
| **CIFAR-10** | MSP(Hendrycks & Gimpel, 2017a) | $89.90 \pm 0.30$ | $91.48 \pm 0.43$ | $60.08 \pm 0.76$ | $\mathbf{94.01 \pm 0.08}$ | |
| | ODIN(Liang et al., 2018) | $91.46 \pm 0.56$ | $91.67 \pm 0.58$ | $42.31 \pm 1.38$ | $\mathbf{94.01 \pm 0.08}$ | |
| | Mahalanobis(Lee et al., 2018b) | $75.10 \pm 1.04$ | $72.32 \pm 1.92$ | $61.35 \pm 1.25$ | $\mathbf{94.01 \pm 0.08}$ | |
| | Energy(Liu et al., 2020) | $92.07 \pm 0.22$ | $92.72 \pm 0.39$ | $42.69 \pm 1.31$ | $\mathbf{94.01 \pm 0.08}$ | |
| | **Energy+UM** (ours) | $93.73 \pm 0.36$ | $94.27 \pm 0.60$ | $33.29 \pm 1.70$ | $92.80 \pm 0.47$ | |
| | **Energy+UMAP** (ours) | $\mathbf{93.97 \pm 0.11}$ | $\mathbf{94.38 \pm 0.06}$ | $\mathbf{30.71 \pm 1.94}$ | $94.01 \pm 0.08$ | |
| | OE(Hendrycks et al., 2019b) | $97.07 \pm 0.01$ | $97.31 \pm 0.05$ | $13.80 \pm 0.28$ | $92.59 \pm 0.32$ | ✓ |
| | Energy (w. $D_{aux}$)(Liu et al., 2020) | $94.58 \pm 0.64$ | $94.69 \pm 0.65$ | $18.79 \pm 2.31$ | $80.91 \pm 3.13$ | ✓ |
| | POEM(Ming et al., 2022) | $94.37 \pm 0.07$ | $94.51 \pm 0.06$ | $18.50 \pm 0.33$ | $77.24 \pm 2.22$ | ✓ |
| | **OE+UM** (ours) | $\mathbf{97.60 \pm 0.03}$ | $\mathbf{97.87 \pm 0.02}$ | $\mathbf{11.22 \pm 0.16}$ | $93.66 \pm 0.12$ | ✓ |
| | **OE+UMAP** (ours) | $97.48 \pm 0.01$ | $97.74 \pm 0.00$ | $12.21 \pm 0.09$ | $94.01 \pm 0.08$ | ✓ |
| **CIFAR-100** | MSP(Hendrycks & Gimpel, 2017a) | $74.06 \pm 0.69$ | $75.37 \pm 0.73$ | $83.14 \pm 0.87$ | $\mathbf{74.86 \pm 0.21}$ | |
| | ODIN(Liang et al., 2018) | $76.18 \pm 0.14$ | $76.49 \pm 0.20$ | $78.93 \pm 0.31$ | $\mathbf{74.86 \pm 0.21}$ | |
| | Mahalanobis(Lee et al., 2018b) | $63.90 \pm 1.91$ | $64.31 \pm 0.91$ | $78.79 \pm 0.50$ | $\mathbf{74.86 \pm 0.21}$ | |
| | Energy(Liu et al., 2020) | $\mathbf{76.29 \pm 0.24}$ | $\mathbf{77.06 \pm 0.55}$ | $78.46 \pm 0.06$ | $\mathbf{74.86 \pm 0.21}$ | |
| | **Energy+UM** (ours) | $76.22 \pm 0.42$ | $76.39 \pm 1.03$ | $74.05 \pm 0.55$ | $64.55 \pm 0.24$ | |
| | **Energy+UMAP** (ours) | $75.57 \pm 0.59$ | $75.66 \pm 0.07$ | $\mathbf{72.21 \pm 1.46}$ | $\mathbf{74.86 \pm 0.21}$ | |
| | OE(Hendrycks et al., 2019b) | $90.55 \pm 0.87$ | $90.34 \pm 0.94$ | $34.73 \pm 3.85$ | $73.59 \pm 0.30$ | ✓ |
| | Energy (w. $D_{aux}$)(Liu et al., 2020) | $88.92 \pm 0.57$ | $89.13 \pm 0.56$ | $37.90 \pm 2.59$ | $57.85 \pm 2.65$ | ✓ |
| | POEM(Ming et al., 2022) | $88.95 \pm 0.54$ | $88.94 \pm 0.31$ | $38.10 \pm 1.30$ | $56.18 \pm 1.92$ | ✓ |
| | **OE+UM** (ours) | $91.04 \pm 0.11$ | $\mathbf{91.13 \pm 0.24}$ | $34.71 \pm 0.81$ | $\mathbf{75.15 \pm 0.18}$ | ✓ |
| | **OE+UMAP** (ours) | $\mathbf{91.10 \pm 0.16}$ | $90.99 \pm 0.23$ | $\mathbf{33.62 \pm 0.26}$ | $74.76 \pm 0.11$ | ✓ |

Table 2: Fine-grained Results (%). Comparison on different OOD benchmark datasets.

| ID dataset | Method | OOD dataset | | | | | |
|---|---|---|---|---|---|---|---|
| | | **CIFAR-100** | | **Textures** | | **Places365** | |
| | | FPR95↓ | AUROC↑ | FPR95↓ | AUROC↑ | FPR95↓ | AUROC↑ |
| **CIFAR-10** | MSP | $66.43 \pm 1.25$ | $87.73 \pm 0.02$ | $65.20 \pm 1.33$ | $88.06 \pm 0.61$ | $61.34 \pm 0.60$ | $89.63 \pm 0.15$ |
| | ODIN | $55.31 \pm 0.85$ | $87.75 \pm 0.37$ | $53.11 \pm 4.84$ | $87.13 \pm 2.04$ | $43.77 \pm 0.20$ | $91.70 \pm 0.30$ |
| | Mahalanobis | $81.61 \pm 0.96$ | $64.52 \pm 0..73$ | $\mathbf{20.04 \pm 1.43}$ | $\mathbf{94.38 \pm 0.78}$ | $86.21 \pm 1.36$ | $64.00 \pm 1.21$ |
| | Energy | $54.65 \pm 1.24$ | $\mathbf{89.01 \pm 1.18}$ | $57.09 \pm 3.52$ | $87.51 \pm 1.43$ | $38.62 \pm 1.64$ | $93.03 \pm 0.20$ |
| | **Energy+UM** (ours) | $\mathbf{54.62 \pm 1.16}$ | $88.30 \pm 0.30$ | $41.61 \pm 3.67$ | $91.31 \pm 0.01$ | $\mathbf{30.85 \pm 0.58}$ | $\mathbf{94.27 \pm 0.16}$ |
| | Method | **SUN** | | **LSUN** | | **iNaturalist** | |
| | | FPR95↓ | AUROC↑ | FPR95↓ | AUROC↑ | FPR95↓ | AUROC↑ |
| | MSP | $60.27 \pm 0.66$ | $90.00 \pm 0.24$ | $36.43 \pm 1.94$ | $95.17 \pm 0.32$ | $67.53 \pm 1.64$ | $88.01 \pm 0.82$ |
| | ODIN | $41.14 \pm 1.29$ | $92.34 \pm 0.62$ | $5.16 \pm 0.76$ | $98.96 \pm 0.09$ | $54.41 \pm 0.91$ | $90.17 \pm 0.19$ |
| | Mahalanobis | $84.56 \pm 1.51$ | $66.41 \pm 4.57$ | $69.18 \pm 3.52$ | $66.41 \pm 4.57$ | $80.76 \pm 2.48$ | $71.77 \pm 1.12$ |
| | Energy | $36.73 \pm 1.72$ | $93.63 \pm 0.34$ | $6.25 \pm 0.43$ | $98.77 \pm 0.07$ | $59.11 \pm 1.18$ | $89.71 \pm 0.06$ |
| | **Energy+UM** (ours) | $\mathbf{27.88 \pm 0.73}$ | $\mathbf{94.83 \pm 0.11}$ | $\mathbf{2.91 \pm 0.53}$ | $\mathbf{99.22 \pm 0.11}$ | $\mathbf{46.27 \pm 2.74}$ | $\mathbf{92.75 \pm 0.80}$ |

In Table 1, we summarize the results of different OOD test sets using different methods. Note that, here the evaluation results are obtained by averaging several OOD test datasets across multiple independent trials. For the scoring-based methods, our UM can further improve the overall detection performance by alleviating the memorization of atypical ID data, when the ID-ACC keeps comparable with the baseline. For the complex `CIFAR-100` dataset, our UMAP can be adopted as a practical way to empower the detection performance and simultaneously avoid affecting the original performance on ID data. As for those methods of the second category (i.e., involving auxiliary outlier $D_{aux}$ sampling from ImageNet), since we consider a practical workflow, i.e., fine-tuning, on the given model, OE achieves the best performance on the task. Due to the special optimization characteristic, Energy (w.$D_{aux}$) and POEM focus more on the energy loss on differentiating OOD data while performing not well on the preservation of ID-ACC. Without sacrificing much performance on ID data, OE with our UM can still achieve better detection performance. In Table 3, the fine-grained detection performance on each OOD testing set demonstrates the general effectiveness of UM. We have comprehensively verified the significant improvement (up to $18\%$ reduced on averaged FPR95) in OOD detection of our methods across different setups in Appendix F, the complete results can refer to Tables 8 to 20. More fine-grained results of the experiments on CIFAR-100 is provided in Table 16 and 19. In addition, we also provide similar results using another model structure in Table 18.

## 4.3 ABLATION AND FURTHER DISCUSSION

In this part, we conduct various explorations to provide a thorough understanding of our presented Unleashing Mask from different perspectives. To be specific, we first present the general existence of the once-covered intrinsic OOD detection capability across different settings. Second, we present

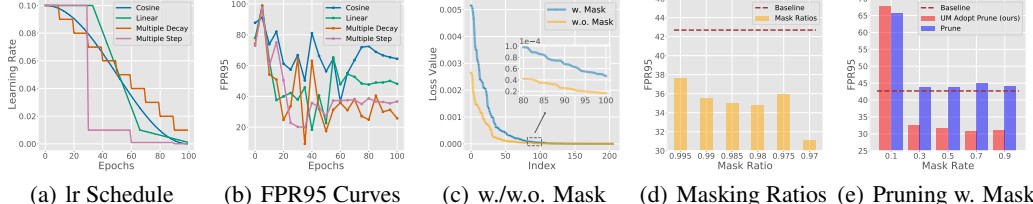

(a) lr Schedule (b) FPR95 Curves (c) w./w.o. Mask (d) Masking Ratios (e) Pruning w. Mask

Figure 4: Ablation studies. (a)-(b) exploring the existence of potential OOD detection capability with different learning rate schedules; (c) comparison of loss value using original output with masked output (the x-axis represents the index of samples within a mini-batch); (d) effects of using different masking ratios in UM; (e) comparison of using original pruning with our proposed UMAP.

the exploration about the mask, which helps us to characterize its effect on figuring out the atypical samples. Third, we provide further exploration on excavating the detection power via pruning.

**General existence of once-covered OOD detection capability.** In Figure 4(a) and Figure 4(b), we explore 4 learning rate schedules to demonstrate the general existence of once-covered OOD detection capability. To be specific, the OOD performance (indicated by FPR95) is evaluated along with the training in every 5 epoch, in which the model takes `CIFAR-10` as ID data and `SVHN` as OOD data. As shown by the curves, a middle stage exists with better OOD performance than that of the final stage across different schedules. We empirically verify the existence of this phenomenon without schedule specificity. More explorations on the other ID dataset and model structure reveal similar results in Figures 5, 6, and 7. The detailed information and discussion can refer to Appendix F.

**Effects of the mask on mining atypical samples.** Following our previous illustration of Figure 3(c), we scrutinize the change of training loss on a random batch of the training set in Figure 4(c). The results further explain why the loss value estimated by the UM can be used to force the model to forget atypical samples. It can be seen from Figure 4(c) that the loss is proportionally increased by randomly knocking out $2.5\%$ weights. In this case, the estimated loss is more influenced by those who have a higher initial loss and are what we termed as atypical samples. By controlling the training loss to the estimated value, the model is encouraged to backtrack to a middle training stage where samples with high loss value have little influence on the forgetting process of the gradient ascent.

**Exploration on revealing detection capability with model pruning.** Although the large constrain on training loss can help reveal the model's OOD performance, the ID-ACC is undermined under such circumstances. Generally speaking, the proposed UM forces the model to forget the atypical samples and may result in lower test performance. To mitigate this issue, we further adopt pruning as a countermeasure to learn a mask instead of tuning the model parameters directly. In Figure 4(e), we experiment with various prune rates $p$ and demonstrate that we can achieve the same or better OOD performance also by pruning. Specifically, our UMAP can achieve a lower FPR95 than pure pruning with the original objective. The prune rate can be selected from a wide range (e.g. $p \in [0.3, 0.9]$) to guarantee a fast convergence and effectiveness. Since pruning doesn't change the well-trained model parameters, it can preserve the performance of the original task. We also provide additional empirical results and corresponding discussion about the effectiveness of our UMAP in Appendix F.

## 5 CONCLUSION

In this work, we explore the intrinsic OOD detection capability of a well-trained model. Without involving any auxiliary outliers in training, we reveal the inconsistent trend between minimizing original training loss and gaining OOD detection capability. We further attribute it to the memorization behavior of atypical samples. To excavate the once-covered capability, we propose a new method, namely, Unleashing Mask (UM). Through UM, we construct model-level discrepancy that figures out the memorized atypical samples and utilizes the constrained gradient ascent to encourage forgetting. It better utilizes the given model for OOD detection via backtracking or sub-structure pruning. We hope our work could provide new insights for revisiting the model development in OOD detection.

REPRODUCIBILITY STATEMENT

For the experimental setups, we have provided the details in Section 4.1 and Appendixes A and E. We will also provide the anonymous repository about our source codes in the discussion phase for reviewing purposes to ensure the reproducibility of our experimental results.

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

APPENDIX

REPRODUCIBILITY STATEMENT

We will provide the anonymous repository about our source codes in the discussion phase for reviewing purposes to ensure the reproducibility of our experimental results. Below we summarize some critical aspects to facilitate the reproducible results:

- **Datasets.** The datasets we used are all publicly accessible, which is introduced in Section 4.1. For methods involving auxiliary outliers, we strictly follow previous works (Sun et al., 2021; Du et al., 2022b) to avoid overlap between the auxiliary dataset (ImageNet-1k) (Deng et al., 2009) and any other OOD datasets.
- **Assumption.** We set our experiments to a post-hoc scenario where a well-trained model is available, and some parts of training samples are also available for subsequent fine-tuning.
- **Open source.** The code repository will be available in an anonymous repository for reviewing purposes. We provide a backbone of our experiments as well as several auxiliary components, such as score estimation.
- **Environment.** All experiments are conducted multiple runs on NVIDIA Tesla V100-SXM2-32GB GPUs with Python 3.6 and PyTorch 1.8.

## A DETAILED INFORMATION ABOUT THE USED BASELINES AND METRICS

In this section, we provide the details about the baselines for the scoring functions and finetuning with auxiliary outliers, and the corresponding hyper-parameters that are considered in our work.

**Maximum Softmax Probability (MSP).** (Hendrycks & Gimpel, 2017a) proposes to use maximum softmax probability to discriminate ID and OOD samples. The score is defined as follows,

$$S_{\text{MSP}}(x; f) = \max_c P(y = c|x; f) = \max \texttt{softmax}(f(x)) \tag{5}$$

where $f$ represents the given well-trained model and $c$ is one of the classes $\mathcal{Y} = \{1, \ldots, C\}$. The larger softmax score indicates the larger probability for a sample to be ID data, reflecting the model's confidence on the sample.

**ODIN.** (Liang et al., 2018) designed the ODIN score, leveraging the temperature scaling and tiny perturbations to widen the gap between the distributions of ID and OOD samples. The ODIN score is defined as follows,

$$S_{\text{ODIN}}(x; f) = \max_c P(y = c|\tilde{x}; f) = \max \texttt{softmax}(\frac{f(\tilde{x})}{T}) \tag{6}$$

where $\tilde{x}$ represents the perturbed samples (controled by $\epsilon$), $T$ represents the temperature. For fair comparison, we adopt the suggested hyperparameters (Liang et al., 2018): $\epsilon = 1.4 \times 10^{-3}$, $T = 1.0 \times 10^4$.

**Mahalanobis.** (Lee et al., 2018b) introduces a Mahalanobis distance-based confidence score, exploiting the feature space of the neural networks by inspecting the class conditional Gaussian distributions. The Mahalanobis distance score is defined as follows,

$$S_{\text{Mahalanobis}}(x; f) = \max_c -(f(x) - \hat{\mu}_c)^T \hat{\Sigma}^{-1}(f(x) - \hat{\mu}_c) \tag{7}$$

where $\hat{\mu}_c$ represents the estimated mean of multivariate Gaussian distribution of class $c$, $\hat{\Sigma}$ represents the estimated tied covariance of the $C$ class-conditional Gaussian distributions.

**Energy.** (Liu et al., 2020) proposes to use the Energy of the predicted logits to distinguish the ID and OOD samples. The Energy score is defined as follows,

$$S_{\text{Energy}}(x; f) = -T \log \sum_{c=1}^{C} e^{f(x)_c/T} \tag{8}$$

where $T$ represents the temperature parameter. As theoretically illustrated in Liu et al. (2020), a lower Energy score indicates a higher probability for a sample to be ID. Following (Liu et al., 2020), we fix the $T$ to 1.0 throughout all experiments.

**Outlier Exposure (OE).** (Hendrycks et al., 2019b) initiates a promising approach towards OOD detections by involving outliers to force apart the distributions of ID and OOD samples. In the experiments, we use the cross-entropy from $f(x_{\text{out}})$ to the uniform distribution as the $\mathcal{L}_{\text{OE}}$ (Lee et al., 2018a),

$$\mathcal{L}_f = \mathbb{E}_{D_{\text{in}}}\left[\ell_{\text{CE}}(f(x), y)\right] + \lambda \mathbb{E}_{D_{\text{out}}^s}\left[\log\sum_{c=1}^{C} e^{f(x)_c} - \mathbb{E}_{D_{\text{out}}^s}(f(x))\right] \tag{9}$$

**Energy (w. $D_{\text{aux}}$).** In addition to using the Energy as a post-hoc score to distinguish ID and OOD samples, (Liu et al., 2020) proposes an Energy-bounded objective to further separate the two distributions. The OE objective is as follows,

$$\mathcal{L}_{\text{OE}} = \mathbb{E}_{D_{\text{in}}^s}(\max(0, S_{\text{Energy}}(x, f) - m_{\text{in}}))^2 + \mathbb{E}_{D_{\text{out}}^s}(\max(0, m_{\text{out}} - S_{\text{Energy}}(x, f)))^2 \tag{10}$$

We keep the thresholds same to (Liu et al., 2020): $m_{\text{in}} = -25.0$, $m_{\text{out}} = -7.0$.

**POEM.** (Ming et al., 2022) explores the Thompson sampling strategy (Thompson, 1933) to make the most use of outliers to learn a tight decision boundary. Though given the POEM's nature to be orthogonal to other OE methods, we use the Energy(w. $D_{\text{aux}}$) as the backbone, which is the same as Eq.( 10) in Liu et al. (2020). The details of Thompson sampling can refer to Ming et al. (2022).

**Detailed formulations of FPR and TPR.** Suppose we have a binary classification task (to predict an image to be an ID or OOD sample in this paper). There are two possible outputs: a positive result (the model predicts an image to be an ID sample); a negative result (the model predicts an image to be an OOD sample). Since we have two possible labels and two possible outputs, we can form a confusion matrix with all possible outputs as follows.

Table 3: Confusion Matrix.

|  | Truth: ID | Truth: OOD |
| --- | --- | --- |
| Predict: ID | True Positive (TP) | False Positive (FP) |
| Predict: OOD | False Negative (FN) | True Negative (TN) |

Therefore, the false positive rate (FPR) is calculated as :

$$FPR = \frac{FP}{FP + TN} \tag{11}$$

The true positive rate (TPR) is calculated as:

$$TPR = \frac{TP}{TP + FN} \tag{12}$$

# B  ADDITIONAL EXPLANATION TOWARDS MINING THE ATYPICAL SAMPLES

First, for identifying those atypical samples using a layer-wise mask with the well-pre-trained model, the core intuition behind is constructing the parameter-level discrepancy to mine the atypical samples. It is inspired by and based on the evidence drawn from previous literature about learning behaviors (Arpit et al., 2017; Goodfellow et al., 2016) of deep neural networks (DNNs) and sparse representation (Frankle & Carbin, 2019; Goodfellow et al., 2013; Barham et al., 2022). To be specific, the atypical samples tend to be learned by the DNNs later than those typical samples (Arpit et al., 2017), and are relatively more sensitive to the changes of the model parameter as the model does not generalize well on that. By the layer-wise mask, the constructed discrepancy can make the model misclassify the atypical samples and estimate loss constraint for the forgetting objective, as visualized in Figure 3(c).

Second, introducing the layer-wise mask has several advantages for achieving the staged target of mining atypical samples in our proposed method, while we would also admit that the layer-wise mask is not an irreplaceable option or maybe not optimal. On the one hand, considering that the model has been trained to approach the zero error on training data, utilizing the layer-wise mask is an integrated strategy to 1) figure out the atypical samples and 2) obtain the loss value computed by the masked output that misclassifies them. The loss constraint is later used in the forgetting objective to fine-tune the model. On the other hand, the layer-wise mask is also compatible with the proposed UMAP to generate a flexible mask for restoring the detection capability of the original model.

Third, we also adopt the unit/weight mask and visualize the misclassified samples in Figure 12. We think they can also be used to mine the atypical samples and can be extended or improved to be a more flexible choice. Further investigating the specific effect of different methods that construct the parameter-level discrepancy would be an interesting sub-topic in future work. For the value of CE loss, although the atypical samples tend to have high CE loss value, they are already memorized and correctly classified as indicated by the zero training error. Only using the high CE error can not provide the loss estimation when the model does not correctly classify those samples.

## C  CONCEPTUAL AND EMPIRICAL COMPARISON WITH OVERFITTING

First of all, we would refer to the concept of the conventional overfitting (Goodfellow et al., 2016; Belkin et al., 2019), i.e., the model "overfit" the training data but fail to generalize and perform well on the test data that is unseen during training. The common empirical reflection of overfitting is that the training error is decreasing while the test error is increasing at the same time, which enlarges the generalization gap of the model. It has been empirically confirmed not the case in our observation as observed in Figure 2(a) and 2(b). To be specific, for the original classification task, there is no conventional overfitting observed as the test performance is still improved at the later training stage, which is a general pursuit of the model development phase on the original tasks.

Then, when we consider the OOD detection performance of the well-pretrained model, our unique observation is about the inconsistency between gaining better OOD detection capability and pursuing better performance on the original classification task for the in-distribution (ID) data. It is worth noting that here the training task is not the binary classification of OOD detection, but the classification task on ID data. It is out of the rigorous concept of the conventional overfitting and has received limited focus and discussion in the previous literature about OOD detection to the best of our knowledge. Considering the practical scenario that exists target-level discrepancy, our revealed observation may encourage us to revisit the detection capability of the well-trained model.

Third, through empirical observation, those strategies designed for preventing the conventional overfitting may need to change the target to the OOD detection based on the important observation. In our experiments, for all the baseline models including that used in Figure 1, we have adopted those strategies (Srivastava et al., 2014; Hastie et al., 2009) (e.g., drop-out, weight decay) to reduce overfitting. It is found to be not enough to restore the OOD detection performance. For another shared issue, on the CIFAR-100 dataset, our UM restore the OOD detection capability of the well-trained model with a significant sacrifice on "ID-ACC". Using those strategies for reducing overfitting in the model development phase maybe not be acceptable to the users that it achieves such a lower performance on the original task. In contrast, our proposed UMAP can be a more practical and flexible way to restore detection performance.

We conduct the extra comparisons between our UM and UMAP with those methods for reducing overfitting. The results are summarized in the following Tables 4, 5, 6 and 7. According to our extra experiments, most conventional methods proposed to prevent conventional overfitting show limited benefits on gaining better OOD detection performance. Based on our important observation, the effective criterion, i.e., early stopping, also need to change its validation target to be the OOD data. However, most of them suffer from higher sacrifice on the performance of the original task and maybe not compatible and practical in the current general setting, i.e., starting from a well-trained model.

Given the concept discrepancy aforementioned, one conclusive message is that "memorization of the atypical samples" are not "memorization in overfitting". Those atypical samples are empirically beneficial in improving the performance on the original classification task as shown in Figure 2. However, this part of knowledge is not very necessary and even harmful to the OOD detection task as

Table 4: Comparison among overfitting methods and ODIN with DenseNet-101 (%). ↑ indicates higher values are better, and ↓ indicates lower values are better.

| $D_{in}$ | Method | AUROC↑ | AUPR↑ | FPR95↓ | ID-ACC↑ |
|---|---|---|---|---|---|
| | Baseline | 91.67 | 91.89 | 40.74 | 93.67 |
| | Early Stopping w. ACC | 92.13 | 92.46 | 38.86 | 93.69 |
| | Early Stopping w. OOD | 92.95 | 93.26 | 35.23 | 93.18 |
| | Weight Decay 0.1 | 86.64 | 86.67 | 60.07 | 88.53 |
| | Weight Decay 0.01 | 90.76 | 91.25 | 44.20 | 92.07 |
| **CIFAR-10** | Weight Decay 0.001 | 88.93 | 88.25 | 48.95 | 94.26 |
| | Drop Rate 0.3 | 91.14 | 92.21 | 46.58 | 90.05 |
| | Drop Rate 0.4 | 84.95 | 86.62 | 62.52 | 82.55 |
| | Drop Rate 0.5 | 83.75 | 85.17 | 62.17 | 75.31 |
| | **UM** (ours) | 92.45 | 93.06 | 37.13 | 92.76 |
| | **UMAP** (ours) | 91.92 | 92.88 | 37.69 | 93.69 |

Table 5: Comparison among overfitting methods and Energy with DenseNet-101 (%). ↑ indicates higher values are better, and ↓ indicates lower values are better.

| $D_{in}$ | Method | AUROC↑ | AUPR↑ | FPR95↓ | ID-ACC↑ |
|---|---|---|---|---|---|
| | Baseline | 92.72 | 93.48 | 38.30 | 93.67 |
| | Early Stopping w. ACC | 92.75 | 93.54 | 37.84 | 93.69 |
| | Early Stopping w. OOD | 93.04 | 93.78 | 36.56 | 93.18 |
| | Weight Decay 0.1 | 86.78 | 88.04 | 65.08 | 88.53 |
| | Weight Decay 0.01 | 90.86 | 91.77 | 47.64 | 92.07 |
| **CIFAR-10** | Weight Decay 0.001 | 90.68 | 90.90 | 47.38 | 94.26 |
| | Drop Rate 0.3 | 90.52 | 91.79 | 51.23 | 90.05 |
| | Drop Rate 0.4 | 84.29 | 86.43 | 68.17 | 82.55 |
| | Drop Rate 0.5 | 83.29 | 85.14 | 68.17 | 75.31 |
| | **UM** (ours) | 93.58 | 94.14 | 33.66 | 92.76 |
| | **UMAP** (ours) | 93.17 | 93.87 | 36.11 | 93.69 |

Table 6: Comparison among overfitting methods and ODIN with WRN-40-4 (%). ↑ indicates higher values are better, and ↓ indicates lower values are better.

| $D_{in}$ | Method | AUROC↑ | AUPR↑ | FPR95↓ | ID-ACC↑ |
|---|---|---|---|---|---|
| | Baseline | 86.24 | 85.90 | 60.13 | 93.86 |
| | Early Stopping w. ACC | 83.80 | 83.30 | 65.13 | 93.99 |
| | Early Stopping w. OOD | 89.92 | 90.94 | 49.87 | 91.47 |
| | Weight Decay 0.1 | 84.38 | 84.75 | 65.75 | 89.88 |
| | Weight Decay 0.01 | 88.08 | 88.45 | 55.16 | 93.16 |
| **CIFAR-10** | Weight Decay 0.001 | 86.34 | 86.38 | 57.42 | 94.91 |
| | Drop Rate 0.3 | 87.53 | 87.25 | 56.12 | 94.22 |
| | Drop Rate 0.4 | 88.24 | 88.41 | 54.62 | 94.20 |
| | Drop Rate 0.5 | 89.13 | 89.99 | 53.07 | 93.91 |
| | **UM** (ours) | 89.61 | 91.13 | 50.97 | 92.68 |
| | **UMAP** (ours) | 90.43 | 91.73 | 46.96 | 93.86 |

the detection performance of the model is drop significantly. Based on the training and test curves in our observation, the memorization in overfitting is expected to happen later than the final stage in which the test performance would drop. Since we have already used some strategies to prevent overfitting, it does not exist. Intuitively, the "atypical samples" identified in our work are relative to the OOD detection task. The memorization of "atypical samples" indicates that the model may not be able to draw the general information of the ID distribution through further learning on those atypical samples through the original classification task Since we mainly provide the empirical observation and understanding of the proposed algorithm in this work, further analysis from other views or theoretically would be an interesting and a major part of future work.

Table 7: Comparison among overfitting methods and Energy with WRN-40-4 (%). ↑ indicates higher values are better, and ↓ indicates lower values are better.

| $D_{in}$ | Method | AUROC↑ | AUPR↑ | FPR95↓ | ID-ACC↑ |
|---|---|---|---|---|---|
| | Baseline | 87.69 | 88.16 | 58.47 | 93.86 |
| | Early Stopping w. ACC | 88.07 | 88.65 | 67.61 | 93.99 |
| | Early Stopping w. OOD | 90.92 | 91.94 | 46.63 | 91.47 |
| | Weight Decay 0.1 | 86.97 | 88.51 | 63.54 | 89.88 |
| | Weight Decay 0.01 | 89.77 | 89.82 | 50.23 | 93.16 |
| **CIFAR-10** | Weight Decay 0.001 | 89.25 | 89.84 | 50.95 | 93.91 |
| | Drop Rate 0.3 | 89.74 | 90.07 | 52.16 | 93.22 |
| | Drop Rate 0.4 | 89.94 | 90.53 | 51.13 | 94.20 |
| | Drop Rate 0.5 | 90.09 | 91.04 | 52.76 | 93.91 |
| | **UM** (ours) | 91.74 | 92.67 | 40.40 | 92.68 |
| | **UMAP** (ours) | 88.84 | 89.31 | 50.23 | 93.86 |

## D    DETAILED REALIZATION OF THE PROPOSED ALGORITHMS

In this section, we provide the detailed realization of our proposed Unleashing Mask (UM) (i.e., in Algorithm 1) and Unleashing Mask Adopt Pruning (UMAP) (i.e., in Algorithm 2).

To estimate the loss constrain $\zeta$ (i.e.,$\widehat{\ell}_{CE}(m_\delta \odot f^*)$ in Eq 3 with the fixed given model $f^*$) for forgetting, we need to randomly knock out parts of weights according to the given mask ratio $\delta$. To be specific, we sample a score from a Gaussian distribution for every weight. Then we initialize a unit matrix for every layer of the model concerning the size of the layer. We formulate the mask $m_\delta$ according to the sampled scores. Find the threshold for each layer that is smaller than the score of the given mask ratio in that layer (termed as quantile). Then set all the ones, whose corresponding scores are more significant than the layers' thresholds, to zeros. In our algorithms, the fine-tuning epochs $k$ is the epochs we fine-tune after we get the well-trained model.

We dot-multiply every layer's weights with the formulated binary matrix as if we delete some parts of the weights. We input a batch of training samples to the masked model and treat the mean value of the outputs' cross-entropy loss as the loss constraint. After all of these have been done, we begin to fine-tune the model's weights with the loss constraint applied to the cross-entropy loss.

For UMAP, the only difference from UM is that, instead of fine-tuning the weights, we generate a popup score for every weight, and force the gradients to pass through the scores. In every iteration, we need to formulate a binary mask according to the given prune rate $p$. This is just what we do when estimating the loss constraint. For more details, it can refer to (Ramanujan et al., 2020). In Table 8, we summarize the overall comparison results of UM and UMAP to show their effectiveness.

## E    ADDITIONAL SETUPS OF THE EXPERIMENTS

In this section, we describe more details about the experimental setups for our exploration.

**Model setups.**    For DenseNet-101, we fix the growth rate and reduce rate to 12 and 0.5 respectively with the bottleneck block included in the backbone. We also explore the proposed UM on WideResNet (Zagoruyko & Komodakis, 2016) with 40 depth and 4 widen factor, which is termed as WRN-40-4. The batch size for both ID and OOD testing samples is 64, and the batch size of auxiliary samples is 2000. The $\lambda$ in Eq.(9) is 0.5 to keep the OE loss comparable to the CE loss. As for the strategy of sampling outliers, we randomly retrieve 50000 samples from ImageNet-1k (Deng et al., 2009) for OE and Energy (w. $D_{aux}$) and 50000 samples using Thompson sampling (Ming et al., 2022) for POEM.

**Learning rate schedules.**    We use 4 different learning rate schedules to demonstrate the existence of the once-covered OOD detection capability. For cosine annealing, we follow the common setups in Loshchilov & Hutter (2017); for linear schedule, the learning rate remains the same in the first one-third epochs, decreases linearly to the tenth of the initial rate in the middle one-third epochs, and decrease linearly to 1% of the initial rate in the last one-third epochs; for the multiple decay

---

**Algorithm 1** Unleashing Mask (UM)

---

**Input:** well-trained model : $\theta$, mask ratio: $\delta \in [0, 1]$, fine-tuning epochs of UM: $k$, training samples: $x \sim D_{\text{in}}^{\text{s}}$ ;
**Output:** fine-tuned model $\hat{\theta}$;

1:   Initialize a popup score for every weight
2: **for** $w \in \theta$ **do**
3:     $s_w \sim N(\mu, \sigma^2)$
4: **end for**
5:   Generate mask by the popup scores
6: **for** $l \in \theta_{\text{layers}}$ **do**
7:     $m_\delta^l = s^l > \text{quantile}(s^l, \delta)$
8: **end for**
9:   Estimate loss constrain
10: $\zeta = \mathbb{E}_{x \sim D_{\text{in}}^{\text{s}}}(\hat{\mathcal{L}}_{\text{CE}}(x, m_\delta \odot \theta))$
11:   Unleashing Mask: fine-tuning
12: **for** $t \in (1, \ldots, k)$ **do**
13:     $\theta^{(t+1)} = \theta^{(t)} - \eta \frac{\partial(|\mathcal{L}_{\text{CE}}(x,\theta)-\zeta|+\zeta)}{\partial\theta}$
14: **end for**

---

**Algorithm 2** Unleashing Mask Adopt Pruning (UMAP)

---

**Input:** well-trained model : $\theta$, mask ratio: $\delta \in [0, 1]$, fine-tuning epochs of UM: $k$, training samples: $x \sim D_{\text{in}}^{\text{s}}$, prune rate: $p$ ;
**Output:** learnt binary mask $\hat{m}_p$;

1:   Initialize a popup score for every weight
2: **for** $w \in \theta$ **do**
3:     $s_w \sim N(\mu, \sigma^2)$
4: **end for**
5:   Generate mask by the popup scores
6: **for** $l \in \theta_{\text{layers}}$ **do**
7:     $m_\delta^l = s^l > \text{quantile}(s^l, \delta)$
8: **end for**
9:   Estimate loss constrain
10: $\zeta = \mathbb{E}_{x \sim D_{\text{in}}^{\text{s}}}(\hat{\mathcal{L}}_{\text{CE}}(x, m_\delta \odot \theta))$
11:   Unleashing Mask Adopt Pruning: fine-tuning
12: $s^{(1)} \sim N(\mu, \sigma^2)$
13: **for** $t \in (1, \ldots, k)$ **do**
14:     **for** $l \in \theta_{\text{layers}}$ **do**
15:       $\hat{m}_p^l = s^{l(t)} > \text{quantile}(s^{l(t)}, p)$
16:     **end for**
17:     $s^{(t+1)} = s^{(t)} - \eta \frac{\partial(|\mathcal{L}_{\text{CE}}(x,\hat{m}_p \odot \theta)-\zeta|+\zeta)}{\partial\theta}$
18: **end for**

---

schedule, the learning rate decreases $10\%$ of the initial rate (0.01) every $10\%$ epochs (10 epochs); for the multiple step schedule, the learning rate decreases to $10\%$ of the current rate every 30 epochs. All those learning rate schedules for our experiments are intuitively illustrated in Figure 4(a).

## F    ADDITIONAL EXPERIMENT RESULTS AND ABLATION STUDY

In this section, we provide more experiment results. We first show the fine-grained results on CIFAR-10 and CIFAR-100, then conduct the experiments under a different model structure (i.e., WRN-40-4), and finally apply an additional ablation study on the proposed UM and UMAP. The mean and variance of all metrics (ID-ACC, AUROC, AUPR, FPR95) are reported based on multiple independent trials.

Table 8: Completed Results (%). Comparison with competitive OOD detection baselines. ↑ indicates higher values are better, and ↓ indicates lower values are better.

| $D_{in}$ | Method | AUROC↑ | AUPR↑ | FPR95↓ | ID-ACC↑ | w./w.o $D_{aux}$ |
|---|---|---|---|---|---|---|
| **CIFAR-10** | MSP(Hendrycks & Gimpel, 2017a) | $89.90 \pm 0.30$ | $91.48 \pm 0.43$ | $60.08 \pm 0.76$ | $\mathbf{94.01 \pm 0.08}$ | |
| | ODIN(Liang et al., 2018) | $91.46 \pm 0.56$ | $91.67 \pm 0.58$ | $42.31 \pm 1.38$ | $\mathbf{94.01 \pm 0.08}$ | |
| | Mahalanobis(Lee et al., 2018b) | $75.10 \pm 1.04$ | $72.32 \pm 1.92$ | $61.35 \pm 1.25$ | $\mathbf{94.01 \pm 0.08}$ | |
| | Energy(Liu et al., 2020) | $92.07 \pm 0.22$ | $92.72 \pm 0.39$ | $42.69 \pm 1.31$ | $\mathbf{94.01 \pm 0.08}$ | |
| | **Energy+UM** (ours) | $93.73 \pm 0.36$ | $94.27 \pm 0.60$ | $33.29 \pm 1.70$ | $92.80 \pm 0.47$ | |
| | **Energy+UMAP** (ours) | $\mathbf{93.97 \pm 0.11}$ | $\mathbf{94.38 \pm 0.06}$ | $\mathbf{30.71 \pm 1.94}$ | $\mathbf{94.01 \pm 0.08}$ | |
| | OE(Hendrycks et al., 2019b) | $97.07 \pm 0.01$ | $97.31 \pm 0.05$ | $13.80 \pm 0.28$ | $92.59 \pm 0.32$ | ✓ |
| | Energy (w. $D_{aux}$)(Liu et al., 2020) | $94.58 \pm 0.64$ | $94.69 \pm 0.65$ | $18.79 \pm 2.31$ | $80.91 \pm 3.13$ | ✓ |
| | POEM(Ming et al., 2022) | $94.37 \pm 0.07$ | $94.51 \pm 0.06$ | $18.50 \pm 0.33$ | $77.24 \pm 2.22$ | ✓ |
| | **OE+UM** (ours) | $\mathbf{97.60 \pm 0.03}$ | $\mathbf{97.87 \pm 0.02}$ | $\mathbf{11.22 \pm 0.16}$ | $\mathbf{93.66 \pm 0.12}$ | ✓ |
| | **Energy+UM** (ours) | $93.02 \pm 0.42$ | $92.36 \pm 0.38$ | $24.41 \pm 1.65$ | $71.97 \pm 0.92$ | ✓ |
| | **POEM+UM** (ours) | $93.04 \pm 0.02$ | $92.99 \pm 0.02$ | $23.52 \pm 0.16$ | $67.41 \pm 0.27$ | ✓ |
| | **OE+UMAP** (ours) | $97.48 \pm 0.01$ | $97.74 \pm 0.00$ | $12.21 \pm 0.09$ | $93.44 \pm 0.21$ | ✓ |
| | **Energy+UMAP** (ours) | $95.63 \pm 1.15$ | $95.92 \pm 1.17$ | $17.51 \pm 2.59$ | $88.12 \pm 4.22$ | ✓ |
| | **POEM+UMAP** (ours) | $94.18 \pm 2.98$ | $94.15 \pm 3.46$ | $20.55 \pm 8.70$ | $76.62 \pm 17.95$ | ✓ |
| **CIFAR-100** | MSP(Hendrycks & Gimpel, 2017a) | $74.06 \pm 0.69$ | $75.37 \pm 0.73$ | $83.14 \pm 0.87$ | $\mathbf{74.86 \pm 0.21}$ | |
| | ODIN(Liang et al., 2018) | $76.18 \pm 0.14$ | $76.49 \pm 0.20$ | $78.93 \pm 0.31$ | $\mathbf{74.86 \pm 0.21}$ | |
| | Mahalanobis(Lee et al., 2018b) | $63.90 \pm 1.91$ | $64.31 \pm 0.91$ | $78.79 \pm 0.50$ | $\mathbf{74.86 \pm 0.21}$ | |
| | Energy(Liu et al., 2020) | $\mathbf{76.29 \pm 0.24}$ | $\mathbf{77.06 \pm 0.55}$ | $78.46 \pm 0.06$ | $\mathbf{74.86 \pm 0.21}$ | |
| | **Energy+UM** (ours) | $76.22 \pm 0.42$ | $76.39 \pm 1.03$ | $74.05 \pm 0.55$ | $64.55 \pm 0.24$ | |
| | **Energy+UMAP** (ours) | $75.57 \pm 0.59$ | $75.66 \pm 0.07$ | $\mathbf{72.21 \pm 1.46}$ | $\mathbf{74.86 \pm 0.21}$ | |
| | OE(Hendrycks et al., 2019b) | $90.55 \pm 0.87$ | $90.34 \pm 0.94$ | $34.73 \pm 3.85$ | $73.59 \pm 0.30$ | ✓ |
| | Energy (w. $D_{aux}$)(Liu et al., 2020) | $88.92 \pm 0.57$ | $89.13 \pm 0.56$ | $37.90 \pm 2.59$ | $57.85 \pm 2.65$ | ✓ |
| | POEM(Ming et al., 2022) | $88.95 \pm 0.54$ | $88.94 \pm 0.31$ | $38.10 \pm 1.30$ | $56.18 \pm 1.92$ | ✓ |
| | **OE+UM** (ours) | $91.04 \pm 0.11$ | $91.13 \pm 0.24$ | $34.71 \pm 0.81$ | $\mathbf{75.15 \pm 0.18}$ | ✓ |
| | **Energy+UM** (ours) | $90.39 \pm 0.40$ | $90.14 \pm 0.45$ | $32.65 \pm 3.13$ | $71.95 \pm 0.23$ | ✓ |
| | **POEM+UM** (ours) | $\mathbf{91.18 \pm 0.35}$ | $\mathbf{91.45 \pm 0.27}$ | $\mathbf{30.78 \pm 1.76}$ | $70.17 \pm 0.01$ | ✓ |
| | **OE+UMAP** (ours) | $91.10 \pm 0.16$ | $90.99 \pm 0.23$ | $33.62 \pm 0.26$ | $74.76 \pm 0.11$ | ✓ |
| | **Energy+UMAP** (ours) | $90.52 \pm 0.26$ | $90.46 \pm 0.50$ | $32.17 \pm 0.30$ | $72.76 \pm 0.18$ | ✓ |
| | **POEM+UMAP** (ours) | $91.10 \pm 0.29$ | $91.41 \pm 0.28$ | $31.02 \pm 1.70$ | $71.05 \pm 0.04$ | ✓ |

**Empirical verification on typical/atypical data.** In the following Tables 9, 10, 11, 12, and 13, we further conduct the experiments to identify the negative effect of learning on those atypical samples by comparing with a counterpart that learning only with the typical samples. The results confirm that the degeneration on detection performance is more likely to come from learning atypical samples.

In Table 9, we provide the main results for verification on typical/atypical samples. Intuitively, we intend to separate the training dataset into a typical set and an atypical set and train respectively on these two sets to see whether it is learning atypical samples that causes OOD performance to decrease during the latter part of the training phase. We force training samples through the model (DenseNet-101) of the 60th epoch and get the CE loss for separation. We provide the ACC of the generated sets on the model of the 60th epoch (ACC in the tables). The extremely low ACCs of the atypical sets show that the model of the 60th epoch can hardly predict the samples, which meets our definition of the atypical sample. We then fine-tune the model of the 60th epoch with the generated dataset and report the OOD performance. The results show learning from only those atypical data fail to gain better detection performance than its counterpart, i.e., learning from only those typical data. Learning on those atypical samples fails to draw the suitable features for the OOD detection task, though it still can improve the original task performance. The experiments provide a conceptual verification of our conjecture which links our observation and the proposed method.

**Results of fine-tuning for less epochs.** UM adopts finetuning on the proposed objective for forgetting has shown the advantages of being cost-effective compared with train-from-scratch. For the tuning epochs, we show in Figures 9 and 10 that fine-tuning using UM can converge within about 20 epochs, indicating that we can apply our UM/UMAP for far less than 100 epochs (compared with train-from-scratch) to restore the better detection performance of the original well-trained model. It is intuitively reasonable that finetuning with the newly designed objective would benefit from the well-trained model, allowing a faster convergence since the two phases consider the same task with the same training data. As for the major experiments conducted in our work, finetuning adopts 100 epochs for better exploring and understanding its learning dynamics for research purposes, this configuration is indicated in the training details of Section 4.1.

Table 9: Fine-tuning on typical/atypical samples with different model structures (%). ↑ indicates higher values are better, and ↓ indicates lower values are better.

| $D_{in}$ | Dataset Size | Structure | Atypical/Typical | AUROC↑ | AUPR↑ | FPR95↓ |
|---|---|---|---|---|---|---|
| **CIFAR-10** | 200 | DenseNet-101 | Atypical | 81.45 | 82.40 | 62.10 |
| | | | Typical | **82.86** | **84.38** | **60.01** |
| | | WRN-40-4 | Atypical | 85.13 | 86.57 | 66.41 |
| | | | Typical | **86.26** | **86.89** | **59.93** |
| **CIFAR-100** | 1000 | DenseNet-101 | Atypical | 71.96 | 73.16 | 85.57 |
| | | | Typical | **74.79** | **75.83** | **80.97** |
| | | WRN-40-4 | Atypical | 66.64 | 67.41 | 86.92 |
| | | | Typical | **71.95** | **72.02** | **80.00** |

Table 10: Fine-tuning on typical/atypical CIFAR-10 samples with DenseNet-101 (%). ↑ indicates higher values are better, and ↓ indicates lower values are better.

| $D_{in}$ | Dataset Size | Atypical/Typical | ACC | AUROC↑ | AUPR↑ | FPR95↓ |
|---|---|---|---|---|---|---|
| **CIFAR-10** | 200 | Atypical | 3.50 | 81.45 | 82.40 | 62.10 |
| | | Typical | 100.00 | **82.86** | **83.48** | **60.01** |
| | 350 | Atypical | 11.14 | 85.90 | 86.01 | 55.10 |
| | | Typical | 100.00 | **85.90** | **86.16** | **52.81** |
| | 500 | Atypical | 16.80 | 84.94 | 85.33 | 59.27 |
| | | Typical | 100.00 | **85.53** | **86.10** | **58.74** |

Table 11: Fine-tuning on typical/atypical CIFAR-10 samples with WRN-40-4 (%). ↑ indicates higher values are better, and ↓ indicates lower values are better.

| $D_{in}$ | Dataset Size | Atypical/Typical | ACC | AUROC↑ | AUPR↑ | FPR95↓ |
|---|---|---|---|---|---|---|
| **CIFAR-10** | 200 | Atypical | 3.50 | 85.13 | 86.57 | 66.41 |
| | | Typical | 100.00 | **86.26** | **86.89** | **59.93** |
| | 350 | Atypical | 11.14 | 82.92 | 84.24 | 68.57 |
| | | Typical | 100.00 | **85.82** | **87.84** | **65.54** |
| | 500 | Atypical | 16.80 | 82.88 | 83.22 | 66.75 |
| | | Typical | 100.00 | **87.38** | **87.93** | **52.27** |

Table 12: Fine-tuning on typical/atypical CIFAR-100 samples with DenseNet-101 (%). ↑ indicates higher values are better, and ↓ indicates lower values are better.

| $D_{in}$ | Dataset Size | Atypical/Typical | ACC | AUROC↑ | AUPR↑ | FPR95↓ |
|---|---|---|---|---|---|---|
| **CIFAR-100** | 500 | Atypical | 1.00 | 72.69 | 73.28 | 80.71 |
| | | Typical | 100.00 | **74.07** | **75.20** | **80.19** |
| | 800 | Atypical | 2.88 | 69.74 | 71.15 | 85.46 |
| | | Typical | 100.00 | **72.49** | **73.17** | **81.97** |
| | 1000 | Atypical | 3.50 | 71.96 | 73.16 | 85.57 |
| | | Typical | 100.00 | **74.79** | **75.83** | **80.97** |

We also provide an extra comparison to show the relative efficiency of our proposed UM/UMAP in the following Table 14 and Table 15. The results show that UM and UMAP can efficiently restore detection performance compared with the baseline. It is intuitively reasonable that fine-tuning would

Table 13: Fine-tuning on typical/atypical CIFAR-100 samples with WRN-40-4 (%). ↑ indicates higher values are better, and ↓ indicates lower values are better.

| $D_{in}$ | Dataset Size | Atypical/Typical | ACC | AUROC↑ | AUPR↑ | FPR95↓ |
|---|---|---|---|---|---|---|
| **CIFAR-100** | 500 | Atypical | 1.00 | 66.03 | 66.17 | 89.56 |
| | | Typical | 100.00 | **68.60** | **69.93** | **86.53** |
| | 800 | Atypical | 2.88 | 67.59 | 68.66 | 85.61 |
| | | Typical | 100.00 | **70.25** | **68.95** | **79.66** |
| | 1000 | Atypical | 3.50 | 66.64 | 67.41 | 86.92 |
| | | Typical | 100.00 | **71.95** | **72.02** | **80.00** |

benefit from the well-trained model, allowing a faster convergence as the two phases consider the same task and training data. Considering the significance of the OOD awareness for those safety-critical areas, it is worthwhile to further excavate the OOD detection capability of the deployed well-trained model using our UM and UMAP.

Table 14: Fine-tuning for 20 epochs with DenseNet-101 (%). ↑ indicates higher values are better, and ↓ indicates lower values are better.

| $D_{in}$ | Epoch | Method | AUROC↑ | AUPR↑ | FPR95↓ | ID-ACC↑ |
|---|---|---|---|---|---|---|
| **CIFAR-10** | 100 | MSP | 89.90 | 91.48 | 60.08 | 94.01 |
| | | ODIN | 91.46 | 91.67 | 42.31 | 94.01 |
| | | Energy | 92.07 | 92.72 | 42.69 | 94.01 |
| | | Energy + UM | 93.73 | 94.27 | 33.29 | 92.80 |
| | | Energy + UMAP | 93.97 | 94.38 | 30.71 | 94.01 |
| | 20 | MSP + UM | 90.31 | 91.99 | 53.61 | 91.70 |
| | | ODIN +UM | 94.08 | 94.67 | 31.01 | 91.70 |
| | | Energy + UM | 93.60 | 94.32 | 33.03 | 91.70 |
| | | MSP + UMAP | 88.70 | 90.39 | 57.69 | 94.01 |
| | | ODIN + UMAP | 92.88 | 93.33 | 35.19 | 94.01 |
| | | Energy + UMAP | 92.88 | 93.39 | 35.60 | 94.01 |

Table 15: Fine-tuning for 20 epochs with WRN-40-4 (%). ↑ indicates higher values are better, and ↓ indicates lower values are better.

| $D_{in}$ | Epoch | Method | AUROC↑ | AUPR↑ | FPR95↓ | ID-ACC↑ |
|---|---|---|---|---|---|---|
| **CIFAR-10** | 100 | MSP | 87.12 | 87.84 | 68.29 | 93.86 |
| | | ODIN | 83.29 | 82.74 | 65.68 | 93.86 |
| | | Energy | 87.69 | 88.16 | 58.47 | 93.86 |
| | | Energy + UM | 91.74 | 92.67 | 40.40 | 92.68 |
| | | Energy + UMAP | 88.84 | 89.31 | 50.23 | 93.86 |
| | 20 | MSP + UM | 89.86 | 91.32 | 51.62 | 91.96 |
| | | ODIN +UM | 91.97 | 92.58 | 41.78 | 91.96 |
| | | Energy + UM | 92.95 | 93.64 | 36.21 | 91.96 |
| | | MSP + UMAP | 88.77 | 90.61 | 61.60 | 93.86 |
| | | ODIN + UMAP | 90.85 | 91.89 | 45.70 | 93.86 |
| | | Energy + UMAP | 91.66 | 92.49 | 42.94 | 93.86 |

**Fine-grained results on OOD data.** In order to further figure out the effectiveness of the proposed UM and UMAP on different OOD datasets, we further report the fine-grained results of our experiments on CIFAR-10 and CIFAR-100 with 6 OOD datasets (CIFAR-10/CIFAR-100, textures, Places365, SUN, LSUN, iNaturalist). The results on the 6 OOD datasets show the general effectiveness of the proposed UM as well as UMAP. In Table 16, **OE + UM** can outperform all the OOD baselines, and further improve the OOD performance even though the original detection performance

Table 16: Fine-grained Results (%) of DenseNet-101 on CIFAR-10. Comparison on different OOD benchmark datasets respectively. ↑ indicates higher values are better, and ↓ indicates lower values are better.

| ID dataset | Method | OOD dataset | | | | | |
|---|---|---|---|---|---|---|---|
| | | CIFAR-100 | | Textures | | Places365 | |
| | | FPR95↓ | AUROC↑ | FPR95↓ | AUROC↑ | FPR95↓ | AUROC↑ |
| | **Energy + UMAP** | $54.95 \pm 2.61$ | $87.72 \pm 1.05$ | $33.59 \pm 1.32$ | $92.67 \pm 0.23$ | $32.80 \pm 4.14$ | $93.57 \pm 1.12$ |
| | OE | $59.29 \pm 1.30$ | $88.51 \pm 0.22$ | $2.89 \pm 0.30$ | $99.16 \pm 0.05$ | $11.14 \pm 1.11$ | $97.50 \pm 0.19$ |
| | Energy (w. $D_{aux}$) | $79.88 \pm 2.47$ | $74.99 \pm 2.40$ | $4.27 \pm 0.57$ | $98.80 \pm 0.19$ | $14.22 \pm 3.99$ | $97.07 \pm 0.72$ |
| | POEM | $82.30 \pm 1.57$ | $72.74 \pm 1.42$ | $1.91 \pm 0.41$ | $99.40 \pm 0.10$ | $11.24 \pm 2.70$ | $96.67 \pm 0.48$ |
| | **OE + UM** (ours) | $\mathbf{55.74 \pm 1.47}$ | $\mathbf{89.53 \pm 0.18}$ | $\mathbf{1.42 \pm 0.15}$ | $\mathbf{99.49 \pm 0.04}$ | $\mathbf{7.77 \pm 0.69}$ | $\mathbf{98.15 \pm 0.08}$ |
| | **Energy (w. $D_{aux}$)+ UM** (ours) | $84.52 \pm 0.01$ | $70.09 \pm 0.47$ | $8.30 \pm 0.88$ | $97.76 \pm 0.05$ | $20.27 \pm 1.30$ | $96.06 \pm 0.31$ |
| | **POEM + UM** (ours) | $84.87 \pm 1.56$ | $68.97 \pm 0.39$ | $4.73 \pm 0.52$ | $98.88 \pm 0.13$ | $19.83 \pm 0.34$ | $96.35 \pm 0.09$ |
| | **OE + UMAP** (ours) | $59.05 \pm 1.41$ | $89.14 \pm 0.14$ | $1.86 \pm 0.07$ | $99.35 \pm 0.00$ | $8.21 \pm 0.12$ | $98.07 \pm 0.03$ |
| | **Energy (w. $D_{aux}$) + UMAP** (ours) | $75.18 \pm 4.96$ | $80.93 \pm 4.49$ | $2.24 \pm 1.34$ | $99.25 \pm 0.29$ | $9.30 \pm 2.12$ | $97.90 \pm 0.40$ |
| CIFAR-10 | **POEM + UMAP** (ours) | $79.33 \pm 4.14$ | $76.89 \pm 4.86$ | $2.10 \pm 1.37$ | $99.34 \pm 0.30$ | $9.94 \pm 6.92$ | $98.01 \pm 1.11$ |
| | Method | SUN | | LSUN | | iNaturalist | |
| | | FPR95↓ | AUROC↑ | FPR95↓ | AUROC↑ | FPR95↓ | AUROC↑ |
| | **Energy + UMAP** | $29.05 \pm 2.78$ | $94.41 \pm 0.73$ | $2.31 \pm 0.88$ | $99.42 \pm 0.04$ | $47.22 \pm 14.03$ | $92.63 \pm 2.14$ |
| | OE | $8.38 \pm 0.71$ | $98.00 \pm 0.14$ | $5.90 \pm 1.43$ | $98.60 \pm 0.21$ | $5.09 \pm 0.64$ | $98.76 \pm 0.11$ |
| | Energy (w. $D_{aux}$) | $10.30 \pm 3.82$ | $97.77 \pm 0.64$ | $12.80 \pm 4.67$ | $96.08 \pm 1.38$ | $6.93 \pm 1.86$ | $98.40 \pm 0.32$ |
| | POEM | $8.39 \pm 2.42$ | $98.16 \pm 0.43$ | $9.69 \pm 1.89$ | $97.25 \pm 0.58$ | $3.78 \pm 0.90$ | $98.99 \pm 0.17$ |
| | **OE + UM** (ours) | $\mathbf{5.51 \pm 0.44}$ | $\mathbf{98.55 \pm 0.07}$ | $\mathbf{3.51 \pm 0.43}$ | $\mathbf{98.93 \pm 0.09}$ | $\mathbf{2.87 \pm 0.49}$ | $\mathbf{99.14 \pm 0.09}$ |
| | **Energy (w. $D_{aux}$)+ UM** (ours) | $16.13 \pm 1.86$ | $96.84 \pm 0.30$ | $23.27 \pm 2.40$ | $92.11 \pm 0.94$ | $11.20 \pm 2.35$ | $97.54 \pm 0.42$ |
| | **POEM + UM** (ours) | $16.16 \pm 0.57$ | $97.01 \pm 0.08$ | $25.69 \pm 0.15$ | $93.38 \pm 0.27$ | $9.30 \pm 1.60$ | $98.05 \pm 0.24$ |
| | **OE + UMAP** (ours) | $6.16 \pm 0.02$ | $98.49 \pm 0.01$ | $4.53 \pm 0.16$ | $98.86 \pm 0.06$ | $3.40 \pm 0.74$ | $98.96 \pm 0.09$ |
| | **Energy (w. $D_{aux}$) + UMAP** (ours) | $6.67 \pm 1.50$ | $98.40 \pm 0.32$ | $23.50 \pm 5.61$ | $94.78 \pm 2.04$ | $3.77 \pm 2.14$ | $98.93 \pm 0.46$ |
| | **POEM + UMAP** (ours) | $7.00 \pm 5.80$ | $98.46 \pm 0.96$ | $21.17 \pm 12.84$ | $94.74 \pm 3.67$ | $3.63 \pm 2.78$ | $98.99 \pm 0.54$ |

Table 17: Fine-grained Results (%) of DenseNet-101 on CIFAR-100. Comparison on different OOD benchmark datasets respectively. ↑ indicates higher values are better, and ↓ indicates lower values are better.

| ID dataset | Method | OOD dataset | | | | | |
|---|---|---|---|---|---|---|---|
| | | CIFAR-10 | | Textures | | Places365 | |
| | | FPR95↓ | AUROC↑ | FPR95↓ | AUROC↑ | FPR95↓ | AUROC↑ |
| | MSP | $83.53 \pm 0.33$ | $75.11 \pm 0.27$ | $86.90 \pm 0.18$ | $71.45 \pm 0.40$ | $85.83 \pm 0.48$ | $70.54 \pm 0.42$ |
| | ODIN | $85.29 \pm 0.17$ | $73.31 \pm 0.24$ | $86.45 \pm 1.27$ | $71.91 \pm 0.27$ | $84.35 \pm 0.64$ | $73.58 \pm 0.51$ |
| | Mahalanobis | $98.25 \pm 0.05$ | $49.60 \pm 1.51$ | $\mathbf{33.06 \pm 3.76}$ | $\mathbf{90.19 \pm 1.21}$ | $95.20 \pm 0.49$ | $53.69 \pm 1.55$ |
| | Energy | $\mathbf{82.16 \pm 0.59}$ | $\mathbf{75.31 \pm 0.27}$ | $90.20 \pm 0.30$ | $68.98 \pm 0.34$ | $82.39 \pm 0.97$ | $73.78 \pm 0.66$ |
| | **Energy+UM** (ours) | $89.62 \pm 0.07$ | $66.12 \pm 0.93$ | $86.99 \pm 1.22$ | $65.39 \pm 1.44$ | $\mathbf{77.30 \pm 2.08}$ | $\mathbf{76.06 \pm 1.05}$ |
| | OE | $90.97 \pm 0.46$ | $69.02 \pm 0.39$ | $14.36 \pm 0.25$ | $95.92 \pm 0.12$ | $40.19 \pm 6.97$ | $90.70 \pm 2.01$ |
| | Energy (w. $D_{aux}$) | $96.14 \pm 0.06$ | $57.52 \pm 0.88$ | $9.02 \pm 0.06$ | $97.16 \pm 0.26$ | $35.18 \pm 4.73$ | $93.29 \pm 1.14$ |
| | POEM | $96.19 \pm 0.16$ | $55.82 \pm 1.05$ | $7.63 \pm 1.40$ | $97.69 \pm 0.06$ | $32.67 \pm 3.73$ | $93.94 \pm 0.68$ |
| | **OE + UM** (ours) | $89.61 \pm 0.08$ | $\mathbf{71.24 \pm 0.08}$ | $16.78 \pm 0.25$ | $95.60 \pm 0.08$ | $39.77 \pm 0.34$ | $91.07 \pm 0.13$ |
| | **Energy (w. $D_{aux}$)+ UM** (ours) | $95.38 \pm 0.45$ | $63.41 \pm 0.14$ | $6.41 \pm 0.83$ | $97.77 \pm 0.33$ | $30.96 \pm 3.61$ | $92.85 \pm 0.69$ |
| | **POEM + UM** (ours) | $95.78 \pm 0.14$ | $60.23 \pm 0.70$ | $\mathbf{5.17 \pm 0.18}$ | $\mathbf{98.53 \pm 0.03}$ | $\mathbf{23.90 \pm 0.84}$ | $\mathbf{95.45 \pm 0.11}$ |
| | **OE + UMAP** (ours) | $\mathbf{90.72 \pm 0.35}$ | $69.76 \pm 0.25$ | $15.32 \pm 0.23$ | $95.72 \pm 0.01$ | $36.42 \pm 1.91$ | $92.08 \pm 0.49$ |
| | **Energy (w. $D_{aux}$) + UMAP** (ours) | $95.39 \pm 0.10$ | $63.26 \pm 0.18$ | $6.52 \pm 0.44$ | $97.83 \pm 0.18$ | $31.18 \pm 0.43$ | $93.13 \pm 0.41$ |
| | **POEM + UMAP** (ours) | $95.69 \pm 0.17$ | $61.62 \pm 0.24$ | $5.23 \pm 0.58$ | $98.52 \pm 0.01$ | $26.06 \pm 1.16$ | $94.91 \pm 0.25$ |
| CIFAR-100 | Method | SUN | | LSUN | | iNaturalist | |
| | | FPR95↓ | AUROC↑ | FPR95↓ | AUROC↑ | FPR95↓ | AUROC↑ |
| | MSP | $88.75 \pm 0.23$ | $66.75 \pm 0.25$ | $67.83 \pm 1.37$ | $82.94 \pm 0.32$ | $85.00 \pm 0.73$ | $76.62 \pm 0.25$ |
| | ODIN | $88.49 \pm 0.99$ | $69.64 \pm 0.61$ | $34.80 \pm 2.55$ | $93.92 \pm 0.75$ | $81.67 \pm 2.77$ | $78.36 \pm 1.57$ |
| | Mahalanobis | $95.53 \pm 0.37$ | $54.37 \pm 1.35$ | $89.31 \pm 4.83$ | $43.19 \pm 16.36$ | $93.63 \pm 1.19$ | $49.60 \pm 1.51$ |
| | Energy | $97.17 \pm 0.92$ | $69.04 \pm 0.83$ | $35.09 \pm 3.17$ | $93.49 \pm 0.87$ | $85.70 \pm 2.14$ | $75.82 \pm 1.72$ |
| | **Energy+UM** (ours) | $\mathbf{81.96 \pm 2.26}$ | $\mathbf{71.47 \pm 1.88}$ | $\mathbf{22.54 \pm 5.93}$ | $\mathbf{94.98 \pm 1.75}$ | $\mathbf{74.28 \pm 3.72}$ | $\mathbf{80.72 \pm 3.75}$ |
| | OE | $44.47 \pm 9.10$ | $90.70 \pm 2.01$ | $\mathbf{5.75 \pm 1.18}$ | $98.57 \pm 0.13$ | $25.51 \pm 4.12$ | $94.46 \pm 0.88$ |
| | Energy (w. $D_{aux}$) | $32.69 \pm 5.69$ | $93.63 \pm 1.48$ | $55.75 \pm 4.31$ | $87.96 \pm 1.03$ | $17.34 \pm 4.54$ | $96.50 \pm 0.81$ |
| | POEM | $30.45 \pm 5.11$ | $94.26 \pm 0.90$ | $46.68 \pm 3.59$ | $90.30 \pm 2.17$ | $16.50 \pm 2.09$ | $96.63 \pm 0.23$ |
| | **OE + UM** (ours) | $44.23 \pm 0.20$ | $90.28 \pm 0.03$ | $5.80 \pm 0.33$ | $98.63 \pm 0.03$ | $26.72 \pm 1.95$ | $94.51 \pm 0.43$ |
| | **Energy (w. $D_{aux}$)+ UM** (ours) | $28.98 \pm 3.15$ | $93.18 \pm 0.69$ | $37.56 \pm 4.81$ | $91.98 \pm 0.83$ | $10.83 \pm 2.06$ | $97.09 \pm 0.69$ |
| | **POEM + UM** (ours) | $\mathbf{21.34 \pm 1.07}$ | $\mathbf{95.76 \pm 0.22}$ | $33.74 \pm 6.22$ | $94.43 \pm 1.18$ | $\mathbf{8.85 \pm 0.23}$ | $\mathbf{97.93 \pm 0.03}$ |
| | **OE + UMAP** (ours) | $39.58 \pm 2.02$ | $91.37 \pm 0.52$ | $5.77 \pm 0.71$ | $\mathbf{98.64 \pm 0.12}$ | $23.33 \pm 1.24$ | $95.08 \pm 0.22$ |
| | **Energy (w. $D_{aux}$) + UMAP** (ours) | $29.65 \pm 1.06$ | $93.38 \pm 0.19$ | $35.94 \pm 0.75$ | $92.08 \pm 0.39$ | $13.96 \pm 2.48$ | $96.87 \pm 0.15$ |
| | **POEM + UMAP** (ours) | $23.73 \pm 0.71$ | $95.25 \pm 0.06$ | $33.09 \pm 5.94$ | $93.57 \pm 0.88$ | $9.76 \pm 1.09$ | $97.77 \pm 0.24$ |

is already well. By equipping with our proposed UM and UMAP, the baselines can outperform their counterparts on most of the OOD datasets. For instance, the FPR95 can decrease from 1.91 to 1.42. In Table 17, we also take a closer check about results on CIFAR-100 with 6 OOD datasets. Our proposed method can almost improve all competitive baselines (either the scoring functions or the finetuning with auxiliary outliers) on the 6 OOD datasets. In both w. $D_{aux}$ and w.o. $D_{aux}$ scenarios, Unleashing Mask can significantly excavate the intrinsic OOD detection capability of the model. In addition to unleashing the excellent OOD performance, UMAP can also maintain the high ID-ACC by learning a binary mask instead of tuning the well-trained original parameters directly.

Table 18: Results (%) of WRN-40-4. Comparison with competitive OOD detection baselines. We respectively train WRN-40-4 on CIFAR-10 and CIFAR-100. For those methods involving outliers, we retrieve 5000 samples from ImageNet-1k. ↑ indicates higher values are better, and ↓ indicates lower values are better.

| $D_{in}$ | Method | AUROC↑ | AUPR↑ | FPR95↓ | ID-ACC↑ |
|---|---|---|---|---|---|
| **CIFAR-10** | MSP(Hendrycks & Gimpel, 2017a) | $87.12 \pm 0.25$ | $87.84 \pm 0.30$ | $68.29 \pm 0.96$ | $\mathbf{93.86 \pm 0.19}$ |
| | ODIN(Liang et al., 2018) | $83.29 \pm 0.72$ | $82.74 \pm 0.79$ | $65.68 \pm 0.77$ | $\mathbf{93.86 \pm 0.19}$ |
| | Mahalanobis(Lee et al., 2018b) | $77.57 \pm 0.28$ | $76.11 \pm 0.10$ | $61.18 \pm 0.10$ | $\mathbf{93.86 \pm 0.19}$ |
| | Energy(Liu et al., 2020) | $87.69 \pm 0.54$ | $88.16 \pm 0.69$ | $58.47 \pm 1.94$ | $\mathbf{93.86 \pm 0.19}$ |
| | **Energy+UM** (ours) | $\mathbf{91.74 \pm 0.43}$ | $\mathbf{92.67 \pm 0.52}$ | $\mathbf{40.40 \pm 1.32}$ | $92.68 \pm 0.23$ |
| | **Energy+UMAP** (ours) | $88.84 \pm 1.02$ | $89.31 \pm 1.44$ | $50.23 \pm 2.25$ | $93.86 \pm 0.19$ |
| **CIFAR-100** | MSP(Hendrycks & Gimpel, 2017a) | $72.34 \pm 0.63$ | $72.69 \pm 0.44$ | $85.40 \pm 0.59$ | $\mathbf{75.01 \pm 0.07}$ |
| | ODIN(Liang et al., 2018) | $68.78 \pm 0.67$ | $66.92 \pm 0.72$ | $85.28 \pm 0.64$ | $\mathbf{75.01 \pm 0.07}$ |
| | Mahalanobis(Lee et al., 2018b) | $68.20 \pm 0.99$ | $68.30 \pm 1.15$ | $76.46 \pm 2.02$ | $\mathbf{75.01 \pm 0.07}$ |
| | Energy(Liu et al., 2020) | $74.00 \pm 0.41$ | $73.02 \pm 0.47$ | $81.37 \pm 0.08$ | $\mathbf{75.01 \pm 0.07}$ |
| | **Energy+UM** (ours) | $76.07 \pm 0.04$ | $76.94 \pm 0.06$ | $74.29 \pm 1.66$ | $59.08 \pm 2.75$ |
| | **Energy+UMAP** (ours) | $\mathbf{77.35 \pm 0.78}$ | $\mathbf{77.43 \pm 0.91}$ | $\mathbf{68.20 \pm 0.06}$ | $75.01 \pm 0.07$ |

Table 19: Fine-grained Results (%) of WRN-40-4 on CIFAR-10. Comparison on different OOD benchmark datasets. ↑ indicates higher values are better, and ↓ indicates lower values are better.

| ID dataset | Method | OOD dataset | | | | | |
|---|---|---|---|---|---|---|---|
| | | CIFAR-100 | | Textures | | Places365 | |
| | | FPR95↓ | AUROC↑ | FPR95↓ | AUROC↑ | FPR95↓ | AUROC↑ |
| | MSP | $70.96 \pm 0.70$ | $86.08 \pm 0.08$ | $68.81 \pm 1.29$ | $86.53 \pm 0.83$ | $68.31 \pm 0.25$ | $86.71 \pm 0.13$ |
| | ODIN | $64.97 \pm 0.08$ | $83.36 \pm 0.11$ | $66.86 \pm 2.24$ | $81.34 \pm 0.81$ | $66.49 \pm 1.16$ | $83.47 \pm 0.93$ |
| | Mahalanobis | $79.84 \pm 0.55$ | $70.33 \pm 0.24$ | $\mathbf{22.56 \pm 0.08}$ | $\mathbf{94.07 \pm 0.04}$ | $85.09 \pm 0.59$ | $67.90 \pm 0.37$ |
| | Energy | $61.09 \pm 0.58$ | $86.66 \pm 0.04$ | $64.29 \pm 1.72$ | $85.56 \pm 0.53$ | $55.32 \pm 0.13$ | $88.29 \pm 0.26$ |
| | **Energy+UM** (ours) | $\mathbf{57.21 \pm 1.41}$ | $\mathbf{87.56 \pm 0.15}$ | $46.49 \pm 1.03$ | $89.74 \pm 0.45$ | $\mathbf{40.68 \pm 4.46}$ | $\mathbf{92.51 \pm 0.97}$ |
| | **Energy+UMAP** (ours) | $65.45 \pm 1.10$ | $84.65 \pm 0.95$ | $59.14 \pm 1.64$ | $85.27 \pm 1.74$ | $48.16 \pm 1.89$ | $90.43 \pm 0.47$ |
| **CIFAR-10** | Method | SUN | | LSUN | | iNaturalist | |
| | | FPR95↓ | AUROC↑ | FPR95↓ | AUROC↑ | FPR95↓ | AUROC↑ |
| | MSP | $68.62 \pm 0.50$ | $86.95 \pm 0.23$ | $52.97 \pm 3.07$ | $92.41 \pm 0.18$ | $76.05 \pm 0.01$ | $83.44 \pm 0.36$ |
| | ODIN | $65.47 \pm 0.78$ | $83.79 \pm 1.10$ | $31.89 \pm 3.44$ | $94.34 \pm 0.89$ | $79.28 \pm 0.18$ | $79.80 \pm 0.35$ |
| | Mahalanobis | $82.92 \pm 0.28$ | $70.52 \pm 0.47$ | $64.31 \pm 0.57$ | $67.75 \pm 0.55$ | $81.50 \pm 2.91$ | $74.97 \pm 2.91$ |
| | Energy | $54.88 \pm 0.18$ | $88.67 \pm 0.30$ | $24.99 \pm 1.38$ | $95.98 \pm 0.37$ | $75.89 \pm 0.85$ | $82.40 \pm 0.22$ |
| | **Energy+UM** (ours) | $\mathbf{38.92 \pm 3.46}$ | $\mathbf{92.98 \pm 0.95}$ | $\mathbf{8.38 \pm 0.77}$ | $\mathbf{98.18 \pm 0.16}$ | $\mathbf{66.02 \pm 6.70}$ | $\mathbf{85.22 \pm 3.42}$ |
| | **Energy+UMAP** (ours) | $45.94 \pm 2.64$ | $91.27 \pm 0.56$ | $14.10 \pm 0.04$ | $97.46 \pm 0.14$ | $74.69 \pm 0.15$ | $81.13 \pm 1.12$ |

**Experiment on different model structure.** Following 4.2, we additionally conduct critical experiments on the WRN-40-4 (Lin et al., 2021) backbone to demonstrate the effectiveness of the proposed UM and UMAP. In Figure 7, we can find during the model training phase on ID data, there also exists the once-covered OOD detection capability can be explored in later development. In Table 18, we show the comparison of multiple OOD detection baselines, evaluating the OOD performance on the 7 OOD datasets mentioned in Section 4.1. The results again demonstrate that our proposed method indeed excavates the intrinsic detection capability and improves the performance.

As for the fine-grained results of WRN-40-4, we report results on 6 OOD datasets respectively. When trained on CIFAR-10, UM can outstrip all the scoring function baselines on 5 OOD datasets except Textures on which Mahalanobis performs better while UMAP still has excellent OOD performance ranking only second to UM. When trained on CIFAR-100, UM and UMAP can also outperform the baselines on most OOD datasets. The fine-grained results of WRN-40-4 further demonstrate the effectiveness of the proposed UM/UMAP on other architectures.

**Additional results about the general existence of once-covered OOD detection capability.** In Section 4.3, we display the once-covered OOD detection capability on CIFAR-10 using SVHN as the OOD dataset. Here, we additionally verify the previously observed trend during training when training DenseNet-101 on CIFAR-100 using iNaturalist as an OOD dataset. In Figure 5, we trace the three evaluation metrics during training on CIFAR-100 using 4 different learning rate schedules. Consistent with the original experiment, we still use iNaturalist as the OOD dataset. It can be seen for all the three metrics that exists a middle stage where the model has the better OOD detection capability (For FPR95, it is smaller (better) in the middle stage; for AUROC and AUPR, they are

Table 20: Fine-grained Results (%) of WRN-40-4 on CIFAR-100. Comparison on different OOD benchmark datasets. ↑ indicates higher values are better, and ↓ indicates lower values are better.

| ID dataset | Method | OOD dataset | | | | | |
|---|---|---|---|---|---|---|---|
| | | CIFAR-10 | | Textures | | Places365 | |
| | | FPR95↓ | AUROC↑ | FPR95↓ | AUROC↑ | FPR95↓ | AUROC↑ |
| | MSP | $83.83 \pm 0.29$ | $75.50 \pm 0.21$ | $86.15 \pm 0.23$ | $72.36 \pm 0.40$ | $86.72 \pm 0.29$ | $69.60 \pm 0.27$ |
| | ODIN | $83.70 \pm 0.30$ | $74.32 \pm 0.03$ | $81.57 \pm 1.74$ | $71.67 \pm 0.28$ | $88.07 \pm 0.11$ | $64.83 \pm 1.36$ |
| | Mahalanobis | $96.89 \pm 0.11$ | $68.78 \pm 0.67$ | $\mathbf{31.02 \pm 2.04}$ | $\mathbf{91.85 \pm 0.91}$ | $93.34 \pm 0.44$ | $61.28 \pm 1.62$ |
| | Energy | $\mathbf{81.32 \pm 0.47}$ | $\mathbf{77.49 \pm 0.26}$ | $86.38 \pm 0.49$ | $73.50 \pm 0.45$ | $84.45 \pm 0.38$ | $69.82 \pm 0.63$ |
| | **Energy+UM** (ours) | $89.23 \pm 1.51$ | $63.85 \pm 1.73$ | $78.90 \pm 0.07$ | $72.58 \pm 1.33$ | $\mathbf{80.46 \pm 1.99}$ | $70.49 \pm 1.01$ |
| | **Energy+UMAP** (ours) | $94.11 \pm 0.72$ | $60.77 \pm 0.96$ | $66.94 \pm 4.49$ | $75.82 \pm 5.33$ | $82.59 \pm 0.27$ | $\mathbf{71.92 \pm 3.58}$ |
| CIFAR-100 | Method | SUN | | LSUN | | iNaturalist | |
| | | FPR95↓ | AUROC↑ | FPR95↓ | AUROC↑ | FPR95↓ | AUROC↑ |
| | MSP | $88.88 \pm 0.83$ | $65.22 \pm 0.85$ | $78.56 \pm 0.66$ | $79.10 \pm 0.53$ | $86.72 \pm 0.29$ | $73,75 \pm 0.56$ |
| | ODIN | $91.00 \pm 0.10$ | $59.06 \pm 1.81$ | $70.14 \pm 2.42$ | $84.03 \pm 0.86$ | $87.86 \pm 1.13$ | $64.52 \pm 1.51$ |
| | Mahalanobis | $94.22 \pm 0.01$ | $60.09 \pm 1.56$ | $89.73 \pm 2.87$ | $40.81 \pm 3.07$ | $87.25 \pm 3.28$ | $74.98 \pm 2.85$ |
| | Energy | $88.35 \pm 0.52$ | $64.04 \pm 0.76$ | $59.84 \pm 0.06$ | $87.91 \pm 0.53$ | $88.91 \pm 0.78$ | $67.81 \pm 0.91$ |
| | **Energy+UM** (ours) | $84.04 \pm 0.09$ | $67.19 \pm 0.14$ | $33.87 \pm 1.21$ | $92.29 \pm 0.02$ | $76.91 \pm 6.07$ | $79.28 \pm 4.17$ |
| | **Energy+UMAP** (ours) | $\mathbf{80.53 \pm 1.31}$ | $\mathbf{72.68 \pm 4.50}$ | $\mathbf{27.79 \pm 2.19}$ | $\mathbf{93.39 \pm 0.57}$ | $\mathbf{55.53 \pm 4.61}$ | $\mathbf{85.65 \pm 0.83}$ |

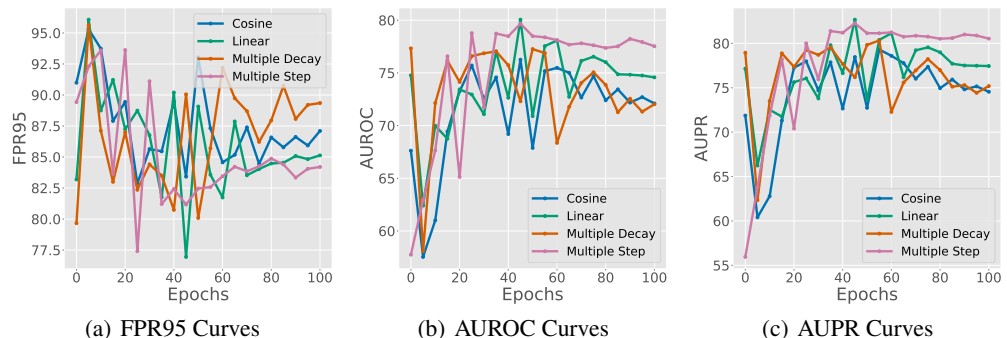

(a) FPR95 Curves      (b) AUROC Curves      (c) AUPR Curves

Figure 5: Ablation studies on three metrics with 4 different learning rate schedules. The model is DenseNet-101 trained on CIFAR-100 with iNaturalist as the OOD dataset. (a) change of FPR95 throughout the pruning phase when training on CIFAR-100; (b) change of AUROC throughout the pruning phase when training on CIFAR-100; (c) change of AUPR throughout the pruning phase when training on CIFAR-100. It demonstrates a better middle stage exists according to the three metrics.

higher (better) in the middle stage). Besides that, we also look into the change of OOD performance on other architecture (e.g., WRN-40-4) in Figure 6 and Figure 7. In Figure 6, we display the curves of three metrics of WRN-40-4 when trained on CIFAR-10 with SVHN and Textures as OOD datasets. The trend that the OOD performance first goes better and then converges to worse OOD performance can be reflected. In Figure 7, we continually provide curves of the three metrics of WRN-40-4 during training on CIFAR-100 with iNaturalist, Places365, and SUN as OOD datasets. A clear better middle stage can still be excavated in this scenario.

**UMAP: adopting pruning on UM.** We conduct various experiments to see whether pruning has an impact on Unleashing Mask itself. To be specific, we expect the pruning to learn a mask on the given model while not impairing the excellent OOD performance that UM brings. In Figure 8, it presents that pruning from a wide range (e.g. $p \in [0.3, 0.9]$) can well maintain the effectiveness of UM while possessing a terrific convergence trend. For simplicity, we use prune to indicate the original pruning approach and UMAP indicate UM with pruning on the mask with our newly designed forgetting objective in Figure 8. In Figure 8(a), the solid lines represent the proposed UMAP and the dashed lines represent only pruning the well-trained model at prune rates $0.2$, $0.5$, and $0.8$. While the model's OOD performance can't be improved (not better than the baseline) through only pruning, using our proposed forgetting objective for the loss constrain can significantly bring out better OOD performance at a wide range of mask rates (e.g. $p \in [0.5, 0.8]$). In Figure 8(b), we intuitively reflect the effect of the estimated loss constraint by the initialized mask which redirects the gradients when the loss reaches the value, while the loss will just approach $0$ when pruning only. In Figure 8(c), we

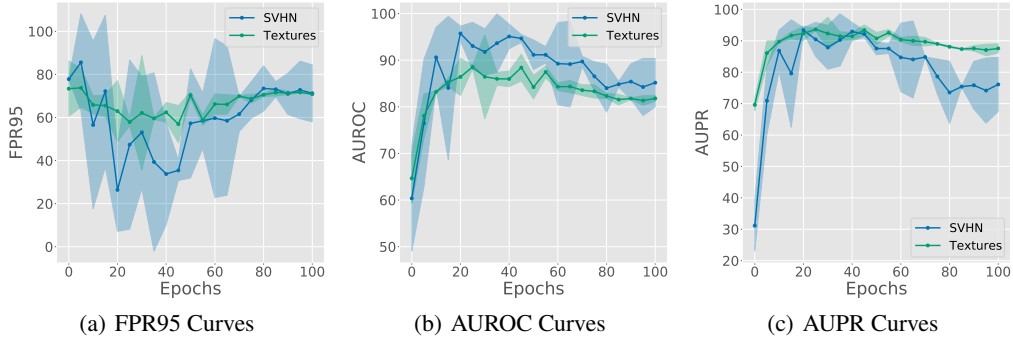

Figure 6: Ablation studies on three metrics of WRN-40-4 with CIFAR-10 as ID dataset, SVHN, and Textures as OOD datasets. (a) change of FPR95 throughout the pruning phase when training on CIFAR-10; (b) change of AUROC throughout the pruning phase when training on CIFAR-10; (c) change of AUPR throughout the pruning phase when training on CIFAR-10. It demonstrates a better middle stage exists according to the three metrics.

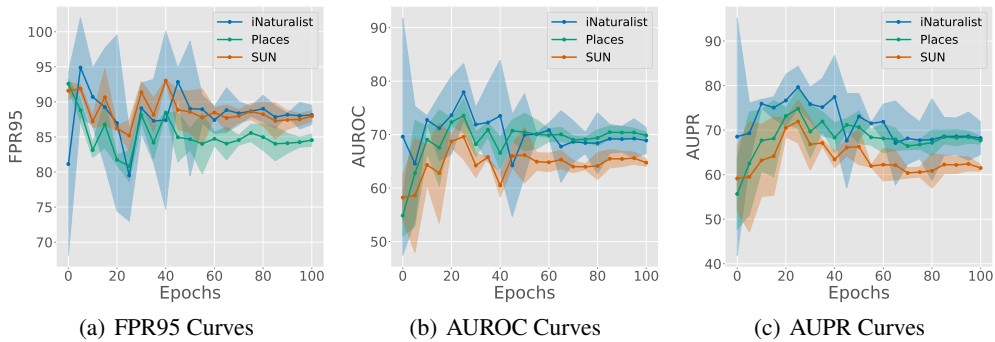

Figure 7: Ablation studies on three metrics of WRN-40-4 with CIFAR-100 as ID dataset, iNaturalist, Places365, and SUN as OOD datasets. (a) change of FPR95 throughout the pruning phase when training on CIFAR-100; (b) change of AUROC throughout the pruning phase when training on CIFAR-100; (c) change of AUPR throughout the pruning phase when training on CIFAR-100. It demonstrates a better middle stage exists according to the three metrics.

can see that ID-ACC for both UMAP and Prune can converge to approximately the same high level ($92 \sim 94$), though we can simply remove the learned mask to recover the original ID-ACC.

**The effectiveness of UM.** In Figure 9, we present the FPR95, AUROC, and AUPR curves during training to show the comparison of the original training and our proposed UM on ID data. We observe that training using UM can consistently outperform than the vanilla model training, either for the final stage or the middle stage with the best OOD detection performance indicated by the FPR95 curve. In Figure 10, we also adopt different mask rates for the initialized loss constraint estimation for forgetting the atypical samples. The results show that a wide range of mask ratios (i.e., from 96% to 99%) to estimate the loss constraint used in Eq.3 can gain better OOD detection performance than the baseline. It shows the mask ratio would be robust to hyper-parameter selection under a certain value. The principle intuition behind this is our revealed important observation as indicated in Figures 1(a), 2(b), and 2(c). With the guidance of the general mechanism, empirically choosing the hyper-parameter using the validation set is supportable and valuable for excavating better OOD detection capability of the model as conducted by previous literature (Hendrycks et al., 2019b; Liu et al., 2020; Sun et al., 2021).

In our experiments, we empirically determine the value of our proposed UM and UMAP by examining the training loss on the masked output. For CIFAR-10 as ID datasets, the value of mask rate is 97.5%

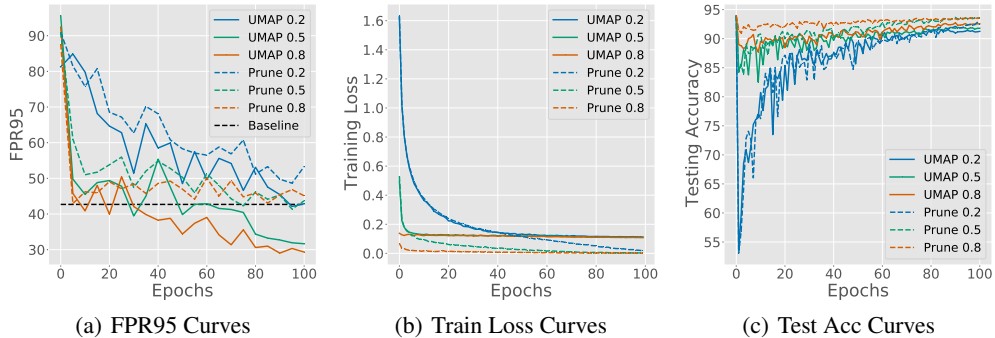

Figure 8: Ablation studies on Prune Rate of UMAP. (a) change of OOD performance throughout the pruning phase; (b) training loss converges to estimated loss constraint properly; (c) though ID-ACC is not taken into consideration for UMAP, it still raise high after training for 100 epochs.

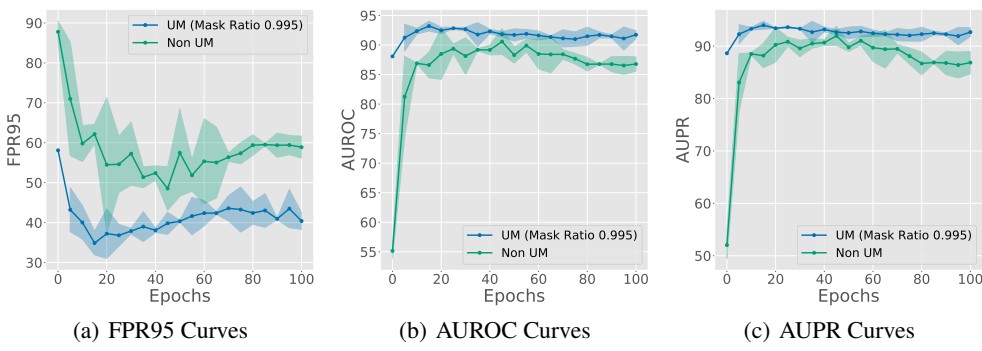

Figure 9: Ablation studies to reflect the effectiveness of UM. The mask ratio of UM is $99.5\%$. (a) change of FPR95 throughout the training phase on CIFAR-10; (b) change of AUROC throughout the training phase on CIFAR-10; (c) change of AUPR throughout the training phase on CIFAR-10.

and the estimated loss constraint for forgetting is $0.10$ for our tuning until the convergence; For CIFAR-100, the value of mask rate is $97\%$ and the estimated loss constraint for forgetting is $1.20$ for our tuning until the convergence.

To choose the parameters of the estimated loss constraint, we use the TinyImageNet (Tavanaei, 2020) dataset as the validation set, which is not seen during training and is not considered in our evaluation of OOD detection performance. Since the core intuition behind our method is to restore the OOD detection performance starting from the well-trained model stage, forgetting a relatively small portion (empirically found around 97% mask ratio) of atypical samples can be beneficial. To find the optimal parameter for tuning, more advanced searching techniques like AutoML or validation design based on the important observation in our work may be further employed in the future.

**Fine-grained comparison of the model weights.** We display the weights of the original model, pruned model, and the UMAP model respectively in Figure 11. The histograms show that the adopted pruning algorithm tends to choose weights far from 0 for the first convolution layer, shown in Figure 11(a). However, for almost all layers (from the 2nd to the 98th), the pruning chooses weights with no respect to the value of weights, shown in Figure 11(b). For the "head" of the model (the fully connected layer), the pruning algorithm itself still keeps its behavior on the first layer, while UMAP forces the prune algorithm to choose weights near 0, shown in Figure 11(c), indicating that forgetting learned atypical samples doesn't necessarily correspond to larger weights or smaller weights.

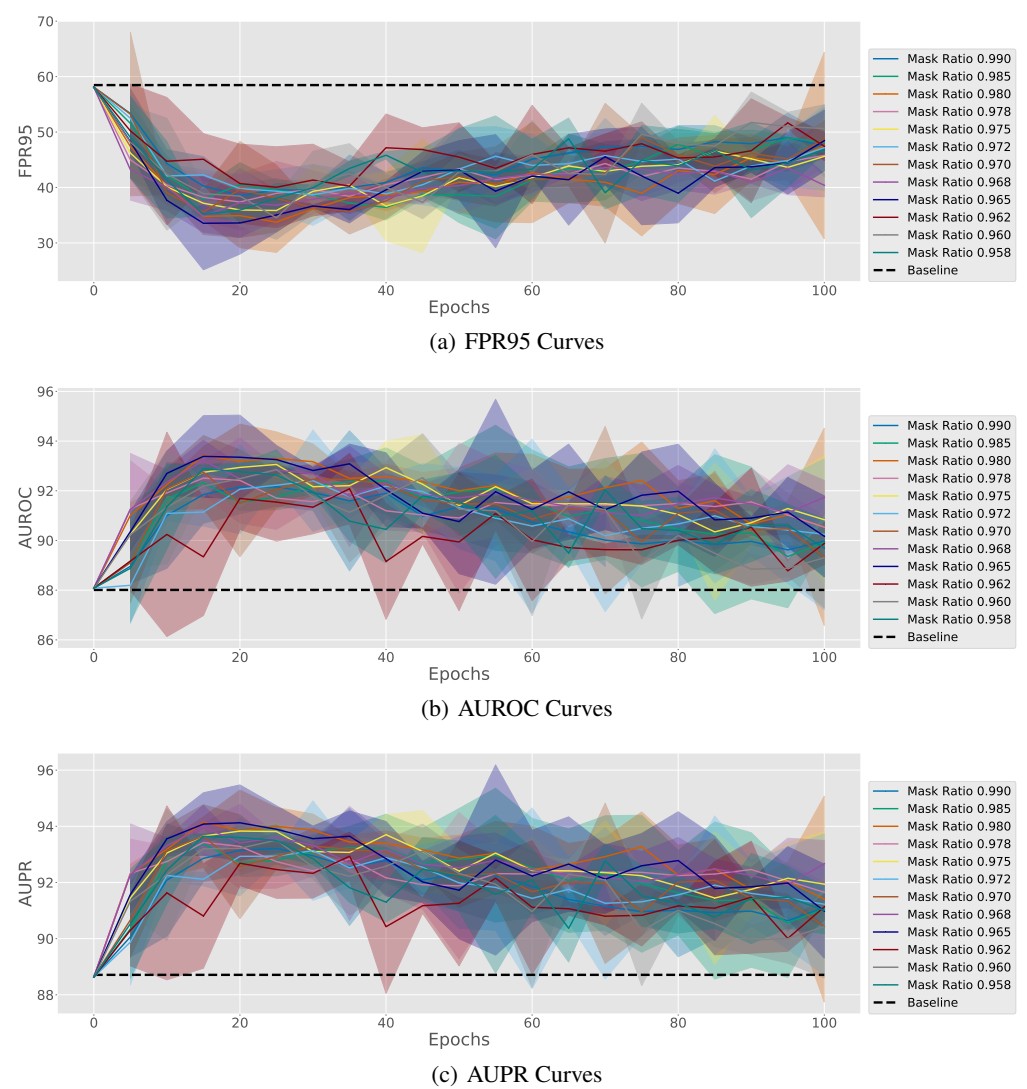

Figure 10: Ablation studies of WRN-40-4 on various Mask Ratios. The mask rate is from $95.8\%$ to $99.0\%$. (a) change of FPR95 throughout the training on CIFAR-10; (b) change of AUROC throughout the training on CIFAR-10; (c) change of AUPR throughout the training on CIFAR-10.

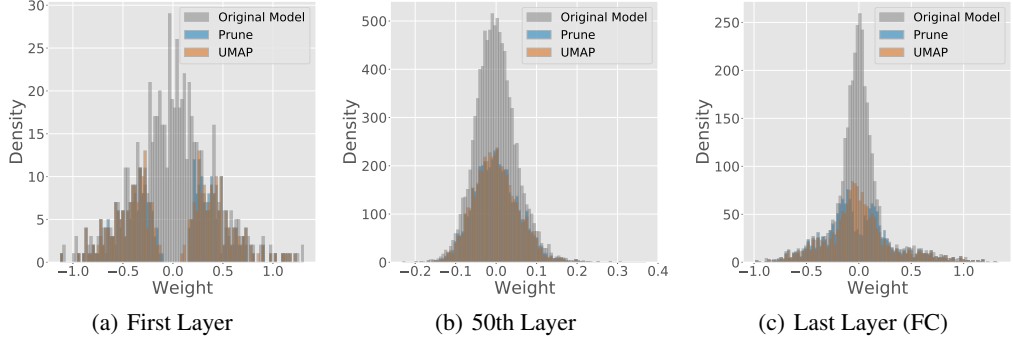

Figure 11: Histograms of different layers for the original model, pruned model, and UMAP model. The model is DenseNet-101 with a prune rate of $50\%$. (a) the histogram of the first convolution layer; (b) the histogram of the 50th convolution layer; (c) the histogram of the last (fully connected) layer.

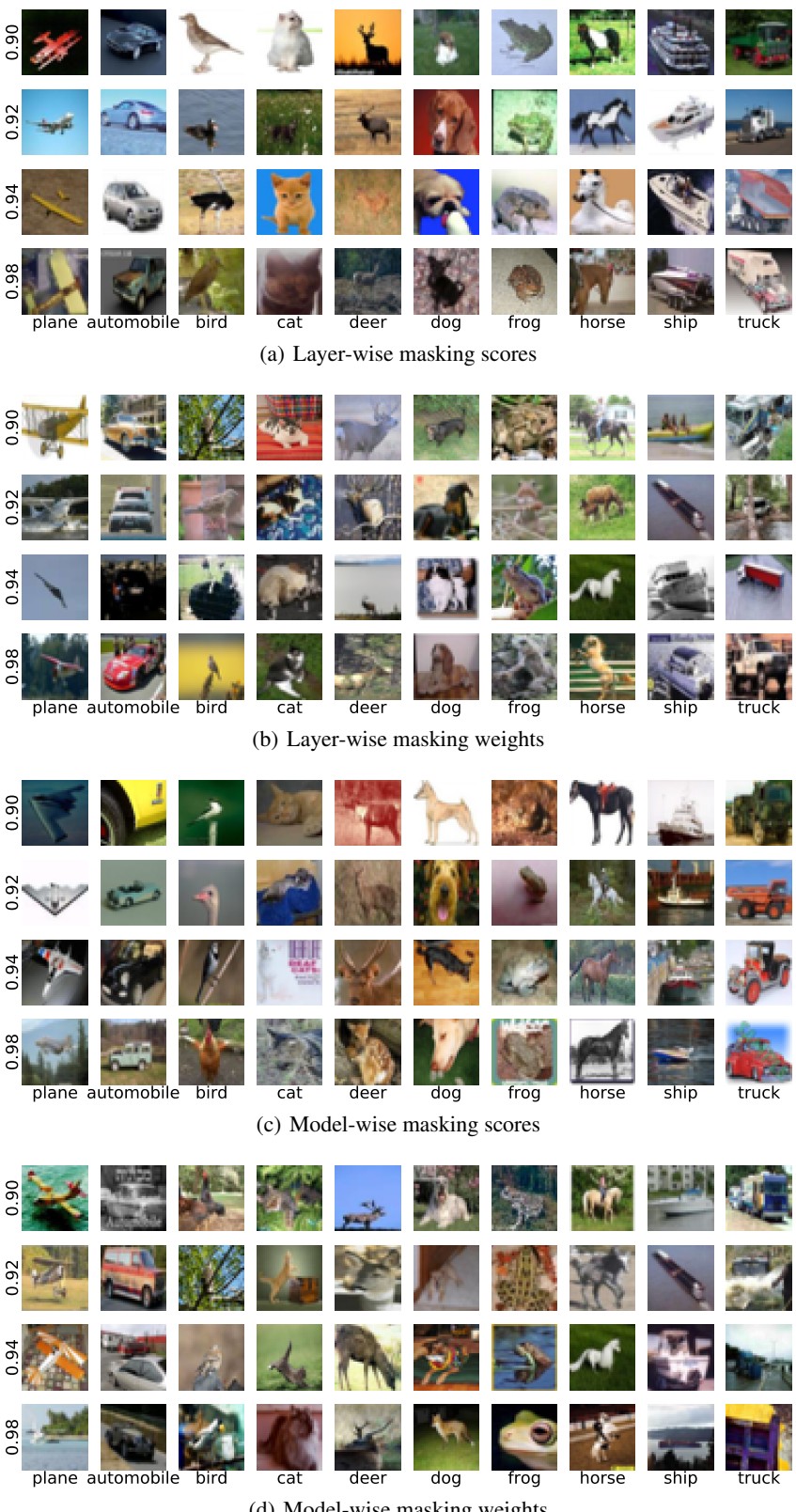

(a) Layer-wise masking scores

(b) Layer-wise masking weights

(c) Model-wise masking scores

(d) Model-wise masking weights

Figure 12: Examples of misclassified samples after masking the original well-trained model. The scores are estimated according to uniform distribution. (a) layer-wise masking scores; (b) layer-wise masking weights directly; (c) model-wise masking scores; (d) model-wise masking weights directly.

