# OpenReview forum: "Unleashing Mask: Explore the Intrinsic Out-of-distribution Detection Capability"
_ICLR.cc/2023/Conference — Submitted to ICLR 2023_

### Official Review · Reviewer_3ovB · 2022-10-13

**Confidence:** 3
**Correctness:** 3
**Technical Novelty And Significance:** 3
**Empirical Novelty And Significance:** 3
**Recommendation:** 6

**Clarity, Quality, Novelty And Reproducibility:**

I think while there may lack certain important discussion, the overall clarity and quality of this paper are above the passing line, and enough details have been provided in the paper for reproducibility.

**Strength And Weaknesses:**

Strength:
1. I find the observation that there exists a historical training stage where the model has a higher OOD detection performance than the final well-trained one very interesting and intuitive. I am pretty supervised that no one ever made such an observation.
2. I believe that throughout the main paper and the appendix, a quite comprehensive set of experiments has been conducted.

Weaknesses (or confusions that may need further clarification):
1. I prefer if the authors could discuss the relationship between this paper and the concept of overfitting. Specifically, from my perspective, the main inspiration of this paper from its own observation is that after a certain number of epochs of training, the model starts to memorize rather than learn knowledge. However, this concept is quite similar to the idea of overfitting. Hence, my two questions will be: (1) Does this mean that the large number of methods that are designed to alleviate overfitting can also be used to facilitate OOD detection, and has the authors tried this? (2) If (1) is not the case, could the authors possibly explain more on what is the difference between the memorization of "atypical samples" here and the memorization in overfitting?

2. I may also prefer if the authors could discuss more on the relationship between their observation and their proposed method. Specifically, when the authors simply say "We further attribute it to the memorization behavior on atypical samples.", I feel confused about this. In other words, why the reason behind the observation that a historical training stage can have a better OOD detection result than the final stage is necessarily the memorization behavior on atypical samples? I hope to see a little bit more discussion on this.

3. Personally, I think the description of UMAP may be too brief in the main paper. I understand that there is a page limit there for the main paper, but I still have the feeling that UMAP may need to be discussed slightly more since a large number of experiments are based on UMAP, which leaves me a feeling that it is quite important but less explained.

4. Two minor comments are: (1) Is it a typo in equation 3 that the loss aims to minimize over the mask instead of the model parameter? (2) I think that especially in Table 1, the author should not only bold their "94.01 ± 0.08" and "74.86 ± 0.21", but also the same results of the previous methods.

**Summary Of The Paper:**

In summary, this paper first observes that there exists a historical training stage where the model has a higher OOD detection performance than the final well-trained one. After that, this paper aims to backtrack the model to that stage. To achieve this, this paper proposes a method called UM to forget those "atypical samples" that are sensitive to the changes in model parameters.

**Summary Of The Review:**

Overall, I think this paper gives an exciting observation and the proposed method sounds reasonable. However, with certain important discussions lacking in the current version, while I tend to accept this paper, I still expect the authors to answer my questions well in the rebuttal phase.

---

> ### Author Response · Authors · 2022-11-15
> **Response to Reviewer 3ovB (5/5)**
>
>
> **Table 8: Fine-tuning on typical/atypical CIFAR-100 samples with DenseNet-101**
>
> | D$_{in}$  | Dataset size | ACC  | Atypical/Typical | AUROC$\uparrow$ | AUPR$\uparrow$ | FPR95$\downarrow$  |
> |-----------|--------------|------|------------------|-----------------|----------------|--------------------|
> | CIFAR-100 | 500          | 1.00 | Atypical                | 72.69           | 73.28          | 80.71              |
> | CIFAR-100 | 500          | 100.00| Typical                | **74.07**           | **75.20**          | **80.19**              |
> | CIFAR-100 | 800          | 2.88 | Atypical                | 69.74           | 71.15          | 85.46              |
> | CIFAR-100 | 800          | 100.00| Typical                | **72.49**           | **73.17**          | **81.97**              |
> | CIFAR-100 | 1000         | 3.50 | Atypical                | 71.96           | 73.16          | 85.57              |
> | CIFAR-100 | 1000         | 100.00| Typical                | **74.79**           | **75.83**          | **80.97**              |
>
>
>
> **Table 9: Fine-tuning on typical/atypical CIFAR-100 samples with WRN-40-4**
>
> | D$_{in}$  | Dataset size | ACC  | Atypical/Typical | AUROC$\uparrow$ | AUPR$\uparrow$ | FPR95$\downarrow$  |
> |-----------|--------------|------|------------------|-----------------|----------------|--------------------|
> | CIFAR-100 | 500          | 1.00 | Atypical                | 66.03           | 66.17          | 89.56              |
> | CIFAR-100 | 500          | 100.00| Typical                | **68.60**            | **69.93**          | **86.53**              |
> | CIFAR-100 | 800          | 2.88 | Atypical                | 67.59           | 68.66          | 85.61              |
> | CIFAR-100 | 800          | 100.00| Typical                | **70.25**           | **68.95**          | **79.66**              |
> | CIFAR-100 | 1000         | 3.50 | Atypical                | 66.64           | 67.41          | 86.92              |
> | CIFAR-100 | 1000         | 100.00| Typical                | **71.95**           | **72.02**          | **80.00**              |
>
> (A brief description of the above experiments for reference.) Intuitively, we intend to separate the training dataset into a typical set and an atypical set and train respectively on these two sets to see whether it is learning atypical samples that causes OOD performance to decrease during the latter part of the training phase. We force training samples through the model (DenseNet-101) of the 60th epoch and get the CE loss for separation. We provide the ACC of the generated sets on the model of the 60th epoch (ACC in the above tables 6-9). The extremely low ACCs of the atypical sets show that the model of the 60th epoch can hardly predict the samples, which meets our definition of the atypical sample. We then fine-tune the model of the 60th epoch with the generated dataset report the OOD performance. As suggested by the results, training on atypical samples can undermine the OOD performance, compared to those of typical samples.
>
>
> > **Q3:** Personally, I think the description of UMAP may be too brief in the main paper. I understand that there is a page limit there for the main paper, but I still have the feeling that UMAP may need to be discussed slightly more since a large number of experiments are based on UMAP, which leaves me a feeling that it is quite important but less explained.
>
> **A3:** Thanks for your constructive suggestions. We will add a special paragraph behind the "objective for forgetting" to introduce the proposed UMAP. Different from the objective of UM (i.e., Eq.(3)), we minimize the loss over the mask to achieve the target of forgetting those atypical samples. UMAP provides the users a chance to utilize an extra mask to restore the detection capacity but not affect the model parameter for the inference on original tasks.
>
>
> > **Q4:** Is it a typo in equation 3 that the loss aims to minimize over the mask instead of the model parameter?
>
> **A4:** Thanks for pointing out this. We revise the second term of Eq.(3), i.e.,$\widehat{\ell}_\text{CE}(m_\delta\odot f)$ to be $\widehat{\ell}_\text{CE}(m_\delta\odot f^*)$, where $m_\delta \odot f^*$ denotes the masked output of the fixed pretrained model parameter that is used to estimate the loss constraint for the forgetting objective. In our realization, it would be a constant value during the fine-tuning process. Equation 3 for our UM still minimizes the loss value over the model parameter.
>
>
> > **Q5:** I think that especially in Table 1, the author should not only bold their "94.01 ± 0.08" and "74.86 ± 0.21", but also the same results of the previous methods.
>
> **A5:** Thanks for the suggestion, we have bold the same results of the previous methods in Table 1 as well as other related tables that have the same issue.

---

> ### Author Response · Authors · 2022-11-15
> **Response to Reviewer 3ovB (4/5)**
>
>
> > **Q2:** I may also prefer if the authors could discuss more on the relationship between their observation and their proposed method. Specifically, when the authors simply say "We further attribute it to the memorization behavior on atypical samples.", I feel confused about this. In other words, why the reason behind the observation that a historical training stage can have a better OOD detection result than the final stage is necessarily the memorization behavior on atypical samples? I hope to see a little bit more discussion on this.
>
> **A2:** First, for the empirical observation of the OOD detection performance during the training phase, we further analyze its underlying learning behavior and present it in Figure 2. Focusing on the general degeneration of detection performance at the later stage (i.e., Epoch 60-100), we found the critical difference before/after the turning point is that the model continually pursues minimizing the training loss to approaching zero, while the loss value mainly comes from those atypical samples visualized in Figure 2\(f). Thus, our conjecture attributes it to the "memorization behavior" of those atypical samples especially in the later training stage to pursue the approximately zero training loss.
>
> Second, to better distinguish the effect of learning on those samples, we conduct extra experiments to compare the OOD detection performance using two parts of data that were selected by their typicality related to our observation. In the following tables (Tables 5-9), the results show learning from only those atypical data fail to gain better detection performance than its counterpart, i.e., learning from only those typical data. It provides a conceptual verification of our conjecture which links our observation and the proposed method.
>
> **Table 5: Fine-tuning on typical/atypical samples with different model structures**
>
> | D$_{in}$  | Dataset size | Structure | Atypical/Typical | AUROC$\uparrow$ | AUPR$\uparrow$ | FPR95$\downarrow$ |
> |-----------|--------------|----------|------------------|-------|-------|-------|
> | CIFAR-10  | 200          | DenseNet | Atypical                | 81.45 | 82.40 | 62.10 |
> | CIFAR-10  | 200          | DenseNet | Typical                | **82.86** | **83.48** | **60.01** |
> | CIFAR-10  | 200          | WRN-40-4 | Atypical                | 85.13 | 86.57 | 66.41 |
> | CIFAR-10  | 200          | WRN-40-4 | Typical                | **86.26** | **86.89** | **59.93** |
> | CIFAR-100 | 1000         | DenseNet | Atypical                | 71.96 | 73.16 | 85.57 |
> | CIFAR-100 | 1000         | DenseNet | Typical                | **74.79** | **75.83** | **80.97** |
> | CIFAR-100 | 1000         | WRN-40-4 | Atypical                | 66.64 | 67.41 | 86.92 |
> | CIFAR-100 | 1000         | WRN-40-4 | Typical                | **71.95** | **72.02** | **80.00** |
>
> **Table 6: Fine-tuning on typical/atypical CIFAR-10 samples with DenseNet-101**
>
> | D$_{in}$ | Dataset size | ACC  | Atypical/Typical | AUROC$\uparrow$ | AUPR$\uparrow$ | FPR95$\downarrow$  |
> |----------|--------------|------|------------------|-----------------|----------------|--------------------|
> | CIFAR-10 | 200          | 3.50 | Atypical                | 81.45           | 82.40          | 62.10              |
> | CIFAR-10 | 200          | 100.00| Typical                | **82.86**           | **83.48**          | **60.01**              |
> | CIFAR-10 | 350          | 11.14| Atypical                | 85.90           | 86.01          | 55.10              |
> | CIFAR-10 | 350          | 100.00| Typical                | **85.90**           | **86.16**          | **52.81**              |
> | CIFAR-10 | 500          | 16.80| Atypical                | 84.94           | 85.33          | 59.27              |
> | CIFAR-10 | 500          | 100.00| Typical                | **85.53**           | **86.10**          | **58.74**              |
>
> **Table 7: Fine-tuning on typical/atypical CIFAR-10 samples with WRN-40-4**
>
> | D$_{in}$ | Dataset size | ACC  | Atypical/Typical | AUROC$\uparrow$ | AUPR$\uparrow$ | FPR95$\downarrow$  |
> |----------|--------------|------|------------------|-----------------|----------------|--------------------|
> | CIFAR-10 | 200          | 3.50 | Atypical                | 85.13           | 86.57          | 66.41              |
> | CIFAR-10 | 200          | 100.00| Typical                | **86.26**           | **86.89**          | **59.93**              |
> | CIFAR-10 | 350          | 11.14| Atypical                | 82.92           | 84.24          | 68.57              |
> | CIFAR-10 | 350          | 100.00| Typical                | **85.82**           | **87.84**          | **65.54**              |
> | CIFAR-10 | 500          | 16.80| Atypical                | 82.88           | 83.22          | 66.75              |
> | CIFAR-10 | 500          | 100.00| Typical                | **87.38**           | **87.93**          | **52.27**              |

---

> ### Author Response · Authors · 2022-11-15
> **Response to Reviewer 3ovB (3/5)**
>
>
> **Table 4: Comparison among overfitting methods and Energy with WRN-40-4**
>
> | D$_{in}$ | Method                | AUROC$\uparrow$ | AUPR$\uparrow$ | FPR95$\downarrow$ | ID-ACC$\uparrow$  |
> |----------|-----------------------|-----------------|----------------|-------------------|-------------------|
> | CIFAR-10 | Baseline              | 87.69           | 88.16          | 58.47             | 93.86             |
> | CIFAR-10 | Early Stopping w. ACC | 88.07           | 88.65          | 67.61             | 93.99             |
> | CIFAR-10 | Early Stopping w. OOD | 90.92           | 91.94          | 46.63             | 91.47             |
> | CIFAR-10 | Weight Decay 0.1      | 86.97           | 88.51          | 63.54             | 89.88             |
> | CIFAR-10 | Weight Decay 0.01     | 89.77           | 89.82          | 50.23             | 93.16             |
> | CIFAR-10 | Weight Decay 0.001    | 89.25           | 89.84          | 50.95             | 93.91             |
> | CIFAR-10 | Drop Rate 0.3         | 89.74           | 90.07          | 52.16             | 93.22             |
> | CIFAR-10 | Drop Rate 0.4         | 89.94           | 90.53          | 51.13             | 94.20             |
> | CIFAR-10 | Drop Rate 0.5         | 90.09           | 91.04          | 52.76             | 93.91             |
> | CIFAR-10 | UM                    | 91.74           | 92.67          | 40.40             | 92.68             |
> | CIFAR-10 | UMAP                  | 88.84           | 89.31          | 50.23             | 93.86             |
>
> According to our extra experiments, most conventional methods proposed to prevent conventional overfitting show limited benefits on gaining better OOD detection performance. Based on our important observation, the effective criterion, i.e., early stopping, also need to change its validation target to be the OOD data. However, most of them suffer from higher sacrifice on the performance of the original task and maybe not compatible and practical in the current general setting, i.e., starting from a well-trained model.
>
> > **Q1.2:** If (Q1.1) is not the case, could the authors possibly explain more on what is the difference between the memorization of "atypical samples" here and the memorization in overfitting?
>
> **A1.2:** Conceptually, we would like to discuss more to draw some distinguishable points from the memorization of "atypical samples" in our work and the memorization of overfitting.
>
> Considering the concept discrepancy as we discuss in Q1, one conclusive message is that "memorization of the atypical samples" are not "memorization in overfitting". Those atypical samples are empirically beneficial in improving the performance on the original classification task as shown in Figure 2. However, this part of knowledge is not very necessary and even harmful to the OOD detection task as the detection performance of the model is drop significantly. Based on the training and test curves in our observation, the memorization in overfitting is expected to happen later than the final stage in which the test performance would drop. Since we have already used some strategies to prevent overfitting, it does not exist. Intuitively, the "atypical samples" identified in our work are relative to the OOD detection task. The memorization of "atypical samples" indicates that the model may not be able to draw the general information of the ID distribution through further learning on those atypical samples through the original classification task.
>
> We would accordingly revise the claim in our main text to explain this part of the concept. Since we mainly provide the empirical observation and understanding of the proposed algorithm in this work, further analysis from other views or theoretically would be an interesting and a major part of future extension.

---

> ### Author Response · Authors · 2022-11-15
> **Response to Reviewer 3ovB (2/5)**
>
> > **Q1.1:** Does this mean that the large number of methods that are designed to alleviate overfitting can also be used to facilitate OOD detection, and has the authors tried this?
>
> **A1.1:** We conduct the extra comparisons between our UM and UMAP with those methods for reducing overfitting. The results are summarized in the following tables (Tables 1-4).
>
> **Table 1: Comparison among overfitting methods and ODIN with DenseNet-101**
>
> | D$_{in}$ | Method    | AUROC$\uparrow$ | AUPR$\uparrow$ | FPR95$\downarrow$ | ID-ACC$\uparrow$ |
> |----------|-----------------------|-------|-------|-------|---------|
> | CIFAR-10 | Baseline              | 91.67 | 91.89 | 40.74 | 93.67   |
> | CIFAR-10 | Early Stopping w. ACC | 92.13 | 92.46 | 38.86 | 93.69   |
> | CIFAR-10 | Early Stopping w. OOD | 92.95 | 93.26 | 35.23 | 93.18   |
> | CIFAR-10 | Weight Decay 0.1      | 86.64 | 86.67 | 60.07 | 88.53   |
> | CIFAR-10 | Weight Decay 0.01     | 90.76 | 91.25 | 44.20 | 92.07   |
> | CIFAR-10 | Weight Decay 0.001    | 88.93 | 88.25 | 48.95 | 94.26   |
> | CIFAR-10 | Drop Rate 0.3         | 91.14 | 92.21 | 46.58 | 90.05   |
> | CIFAR-10 | Drop Rate 0.4         | 84.94 | 86.62 | 62.52 | 82.55   |
> | CIFAR-10 | Drop Rate 0.5         | 83.75 | 85.17 | 62.17 | 75.31   |
> | CIFAR-10 | UM                    | 92.45 | 93.06 | 37.13 | 92.76   |
> | CIFAR-10 | UMAP                  | 91.92 | 92.88 | 37.69 | 93.69   |
>
> **Table 2: Comparison among overfitting methods and Energy with DenseNet-101**
>
> | D$_{in}$ | Method    | AUROC$\uparrow$ | AUPR$\uparrow$ | FPR95$\downarrow$ | ID-ACC$\uparrow$ |
> |----------|-----------------------|-------|-------|-------|---------|
> | CIFAR-10 | Baseline              | 92.72 | 93.48 | 38.30 | 93.67   |
> | CIFAR-10 | Early Stopping w. ACC | 92.75 | 93.54 | 37.84 | 93.69   |
> | CIFAR-10 | Early Stopping w. OOD | 93.04 | 93.78 | 36.56 | 93.18   |
> | CIFAR-10 | Weight Decay 0.1      | 86.78 | 88.04 | 65.08 | 88.53   |
> | CIFAR-10 | Weight Decay 0.01     | 90.86 | 91.77 | 47.64 | 92.07   |
> | CIFAR-10 | Weight Decay 0.001    | 90.68 | 90.90 | 47.38 | 94.26   |
> | CIFAR-10 | Drop Rate 0.3         | 90.52 | 91.79 | 51.23 | 90.05   |
> | CIFAR-10 | Drop Rate 0.4         | 84.39 | 86.43 | 68.17 | 82.55   |
> | CIFAR-10 | Drop Rate 0.5         | 83.29 | 85.14 | 68.75 | 75.31   |
> | CIFAR-10 | UM                    | 93.58 | 94.14 | 33.66 | 92.76   |
> | CIFAR-10 | UMAP                  | 93.17 | 93.87 | 36.11 | 93.69   |
>
> **Table 3: Comparison among overfitting methods and ODIN with WRN-40-4**
>
> | D$_{in}$ | Method                | AUROC$\uparrow$ | AUPR$\uparrow$ | FPR95$\downarrow$ | ID-ACC$\uparrow$  |
> |----------|-----------------------|-----------------|----------------|-------------------|-------------------|
> | CIFAR-10 | Baseline              | 86.24           | 85.90          | 60.13             | 93.86             |
> | CIFAR-10 | Early Stopping w. ACC | 83.80           | 83.30          | 65.13             | 93.99             |
> | CIFAR-10 | Early Stopping w. OOD | 89.92           | 90.94          | 49.87             | 91.47             |
> | CIFAR-10 | Weight Decay 0.1      | 84.38           | 84.75          | 65.75             | 89.88             |
> | CIFAR-10 | Weight Decay 0.01     | 88.08           | 88.45          | 55.16             | 93.16             |
> | CIFAR-10 | Weight Decay 0.001    | 86.34           | 86.38          | 57.42             | 94.91             |
> | CIFAR-10 | Drop Rate 0.3         | 87.53           | 87.25          | 56.12             | 94.22             |
> | CIFAR-10 | Drop Rate 0.4         | 88.24           | 88.41          | 54.62             | 94.20             |
> | CIFAR-10 | Drop Rate 0.5         | 89.13           | 89.99          | 53.07             | 93.91             |
> | CIFAR-10 | UM                    | 89.61           | 91.13          | 50.97             | 92.68             |
> | CIFAR-10 | UMAP                  | 90.43           | 91.73          | 46.96             | 93.86             |

---

> ### Author Response · Authors · 2022-11-15
> **Response to Reviewer 3ovB (1/5)**
>
>
> Thank you for your time devoted to reviewing this paper and your constructive suggestions. Here are our detailed replies to your questions.
>
> > **Q1**: I prefer if the authors could discuss the relationship between this paper and the concept of overfitting. Specifically, from my perspective, the main inspiration of this paper from its own observation is that after a certain number of epochs of training, the model starts to memorize rather than learn knowledge. However, this concept is quite similar to the idea of overfitting.
>
> **A1:** We would like to discuss the relationship with the conventional overfitting from the perspective of their unique concepts and enlighten some distinguishable points of our observation with empirical evidence.
>
> First of all, we would refer to the concept of the conventional overfitting [1,2], i.e., the model "overfit" the training data but fail to generalize and perform well on the test data that is unseen during training. The common empirical reflection of overfitting is that the training error is decreasing while the test error is increasing at the same time, which enlarges the generalization gap of the model. It has been empirically confirmed not the case in our observation as observed in Figure 2(a) and 2(b). To be specific, for the original classification task, there is no conventional overfitting observed as the test performance is still improved at the later training stage, which is a general pursuit of the model development phase on the original tasks.
>
> Then, when we consider the OOD detection performance of the well-pretrained model, our unique observation is about the inconsistency between gaining better OOD detection capability and pursuing better performance on the original classification task for the in-distribution (ID) data. It is worth noting that here the training task is not the binary classification of OOD detection, but the classification task on ID data. It is out of the rigorous concept of the conventional overfitting and has received limited focus and discussion in the previous literature about OOD detection to the best of our knowledge. Considering the practical scenario that exists target-level discrepancy, our revealed observation may encourage us to revisit the detection capability of the well-trained model.
>
> Third, through empirical observation, those strategies designed for preventing the conventional overfitting may need to change the target to the OOD detection based on the important observation. In our experiments, for all the baseline models including that used in Figure 1, we have adopted those strategies [3,4] (e.g., drop-out, weight decay) to reduce overfitting. It is found to be not enough to restore the OOD detection performance. For another shared issue, on the CIFAR-100 dataset, our UM restore the OOD detection capability of the well-trained model with a significant sacrifice on "ID-ACC". Using those strategies for reducing overfitting in the model development phase maybe not be acceptable to the users that it achieves such a lower performance on the original task. In contrast, our proposed UMAP can be a more practical and flexible way to restore detection performance.
>
> To further investigate and compare the performance using different strategies for reducing overfitting, we provide extra experimental results and discussion on the response later.
>
> We will add all the comprehensive discussion and extra empirical exploration in our updated version.
>
> [1] Goodfellow, Ian, Yoshua Bengio, and Aaron Courville. Deep learning. MIT press, 2016.
>
> [2] Belkin, Mikhail, et al. "Reconciling modern machine-learning practice and the classical bias–variance trade-off." Proceedings of the National Academy of Sciences 116.32 (2019): 15849- 15854.
>
> [3] Srivastava, Nitish, et al. "Dropout: a simple way to prevent neural networks from overfitting." The journal of machine learning research 15.1 (2014): 1929-1958.
>
> [4] Hastie, Trevor, et al. The elements of statistical learning: data mining, inference, and prediction. Vol. 2. New York: springer, 2009.

---

> ### Comment · Reviewer_3ovB · 2022-11-17
> **Response to Authors**
>
> Thanks to the authors for their further explanation. After reading the responses from the authors, I think they largely address my concern and I decide to keep my positive rating.

---

> > ### Author Response · Authors · 2022-11-17
> > **Thank you for reading our response**
> >
> > Dear Reviewer 3ovB,
> >
> > Thank you for reading our response! we are glad to hear that our clarification addressed your concerns.
> >
> > Best,
> >
> > Authors

---

### Official Review · Reviewer_miMq · 2022-10-14

**Confidence:** 4
**Clarity, Quality, Novelty And Reproducibility:** See Strength And Weaknesses.
**Correctness:** 3
**Technical Novelty And Significance:** 3
**Empirical Novelty And Significance:** 3
**Recommendation:** 8

**Strength And Weaknesses:**


Strength:

1. The paper is well-written and easy to follow in most sections.

2. The motivation is clear and the proposed methods make sense in addressing the related problems.

3. The related works are extensive and well-organized

4. The experimental section is comprehensive and covers the majority of the necessary empirical study.


Weakness (not necessary, but questions/suggestions):

1. Why can the atypical samples be mined by the layer-wise mask? I was lost in reading section 3.2. More explanations and details can be elaborated on there.

2. Are the masks applied at all layers or just some typical layers?

3. How does Equation 3 stabilize around the stage that can forget those atypical samples? Seems like the absolute value term makes sure the masked loss is close to the original loss. The second term is minimizing the masked loss. Where is the "forgetting" mechanism happening?

4. Why use the layer-wised mask? Is it the optimal choice for identifying the atypical samples? Can we use a unit/weight mask or use high CE error to identify atypical samples?

5. Can we use other strategies like the early stopping criterion to prevent overfitting? The comparison would be interesting to see.

6. Another interesting experiment to do is to separate the training dataset into two types (typical samples / atypical samples) and train two models respectively on them. It would be good to show that atypical samples are indeed the cause of OOD performance getting worse.



**Summary Of The Paper:**

This paper is motivated by the observation that a fully-converged model does not necessarily give the best OOD detection performance. The overfitting is likely due to the memorization of the atypical samples in the training data. The author proposes the Unleashing Mask used in the training stage to identify these atypical samples and let the network forgets them. The experimental results show that the proposed methods are effective in both post hoc methods and OE-based methods.


**Summary Of The Review:**

This paper reveals an important observation that well-converged networks do not necessarily lead to a good OOD detection performance. The cause is likely the over-memorization of the atypical samples. The paper proposes an algorithm to identify and forget them, which makes sense. Overall this paper is a good study of the OOD detection problem. So I vote for acceptance.

---

> ### Author Response · Authors · 2022-11-15
> **Response to Reviewer miMq (4/4)**
>
>
> **Table 7: Fine-tuning on typical/atypical CIFAR-10 samples with WRN-40-4**
>
> | D$_{in}$ | Dataset size | ACC  | Atypical/Typical | AUROC$\uparrow$ | AUPR$\uparrow$ | FPR95$\downarrow$  |
> |----------|--------------|------|------------------|-----------------|----------------|--------------------|
> | CIFAR-10 | 200          | 3.50 | Atypical                | 85.13           | 86.57          | 66.41              |
> | CIFAR-10 | 200          | 100.00| Typical                | **86.26**           | **86.89**          | **59.93**              |
> | CIFAR-10 | 350          | 11.14| Atypical                | 82.92           | 84.24          | 68.57              |
> | CIFAR-10 | 350          | 100.00| Typical                | **85.82**           | **87.84**          | **65.54**              |
> | CIFAR-10 | 500          | 16.80| Atypical                | 82.88           | 83.22          | 66.75              |
> | CIFAR-10 | 500          | 100.00| Typical                | **87.38**           | **87.93**          | **52.27**              |
>
> **Table 8: Fine-tuning on typical/atypical CIFAR-100 samples with DenseNet-101**
>
> | D$_{in}$  | Dataset size | ACC  | Atypical/Typical | AUROC$\uparrow$ | AUPR$\uparrow$ | FPR95$\downarrow$  |
> |-----------|--------------|------|------------------|-----------------|----------------|--------------------|
> | CIFAR-100 | 500          | 1.00 | Atypical                | 72.69           | 73.28          | 80.71              |
> | CIFAR-100 | 500          | 100.00| Typical                | **74.07**           | **75.20**          | **80.19**              |
> | CIFAR-100 | 800          | 2.88 | Atypical                | 69.74           | 71.15          | 85.46              |
> | CIFAR-100 | 800          | 100.00| Typical                | **72.49**           | **73.17**          | **81.97**              |
> | CIFAR-100 | 1000         | 3.50 | Atypical                | 71.96           | 73.16          | 85.57              |
> | CIFAR-100 | 1000         | 100.00| Typical                | **74.79**           | **75.83**          | **80.97**              |
>
> **Table 9: Fine-tuning on typical/atypical CIFAR-100 samples with WRN-40-4**
>
> | D$_{in}$  | Dataset size | ACC  | Atypical/Typical | AUROC$\uparrow$ | AUPR$\uparrow$ | FPR95$\downarrow$  |
> |-----------|--------------|------|------------------|-----------------|----------------|--------------------|
> | CIFAR-100 | 500          | 1.00 | Atypical                | 66.03           | 66.17          | 89.56              |
> | CIFAR-100 | 500          | 100.00| Typical                | **68.60**            | **69.93**          | **86.53**              |
> | CIFAR-100 | 800          | 2.88 | Atypical                | 67.59           | 68.66          | 85.61              |
> | CIFAR-100 | 800          | 100.00| Typical                | **70.25**           | **68.95**          | **79.66**              |
> | CIFAR-100 | 1000         | 3.50 | Atypical                | 66.64           | 67.41          | 86.92              |
> | CIFAR-100 | 1000         | 100.00| Typical                | **71.95**           | **72.02**          | **80.00**              |
>
> (A brief description of the above experiments for reference.) Intuitively, we intend to separate the training dataset into a typical set and an atypical set and train respectively on these two sets to see whether it is learning atypical samples that causes OOD performance to decrease during the latter part of the training phase. We force training samples through the model (DenseNet-101) of the 60th epoch and get the CE loss for separation. We provide the ACC of the generated sets on the model of the 60th epoch (ACC in the above tables 6-9). The extremely low ACCs of the atypical sets show that the model of the 60th epoch can hardly predict the samples, which meets our definition of the atypical sample. We then fine-tune the model of the 60th epoch with the generated dataset to report the OOD performance. As suggested by the results, training on atypical samples can undermine the OOD performance, compared to those of typical samples.
>
> We will add the whole experimental results in our updated version, and also the detailed discussion.

---

> ### Author Response · Authors · 2022-11-15
> **Response to Reviewer miMq (3/4)**
>
> **Table 4: Comparison among overfitting methods and Energy with WRN-40-4**
>
> | D$_{in}$ | Method                | AUROC$\uparrow$ | AUPR$\uparrow$ | FPR95$\downarrow$ | ID-ACC$\uparrow$  |
> |----------|-----------------------|-----------------|----------------|-------------------|-------------------|
> | CIFAR-10 | Baseline              | 87.69           | 88.16          | 58.47             | 93.86             |
> | CIFAR-10 | Early Stopping w. ACC | 88.07           | 88.65          | 67.61             | 93.99             |
> | CIFAR-10 | Early Stopping w. OOD | 90.92           | 91.94          | 46.63             | 91.47             |
> | CIFAR-10 | Weight Decay 0.1      | 86.97           | 88.51          | 63.54             | 89.88             |
> | CIFAR-10 | Weight Decay 0.01     | 89.77           | 89.82          | 50.23             | 93.16             |
> | CIFAR-10 | Weight Decay 0.001    | 89.25           | 89.84          | 50.95             | 93.91             |
> | CIFAR-10 | Drop Rate 0.3         | 89.74           | 90.07          | 52.16             | 93.22             |
> | CIFAR-10 | Drop Rate 0.4         | 89.94           | 90.53          | 51.13             | 94.20             |
> | CIFAR-10 | Drop Rate 0.5         | 90.09           | 91.04          | 52.76             | 93.91             |
> | CIFAR-10 | UM                    | 91.74           | 92.67          | 40.40             | 92.68             |
> | CIFAR-10 | UMAP                  | 88.84           | 89.31          | 50.23             | 93.86             |
>
>
> > **Q5:** Another interesting experiment to do is to separate the training dataset into two types (typical samples / atypical samples) and train two models respectively on them. It would be good to show that atypical samples are indeed the cause of OOD performance getting worse.
>
> **A5:** Thanks for your suggestion. We add the extra experiments following the valuable setting. To be specific, we further select two parts of data according to their typicality and compare the learning only on one kind of data. The results are summarized in the following tables 5-9, which shows that only learning with those atypical data can not achieve better detection performance compared with its counterpart, i.e., learning with only the relative typical data.
>
> **Table 5: Fine-tuning on typical/atypical samples with different model structures**
>
> | D$_{in}$  | Dataset size | Structure | Atypical/Typical | AUROC$\uparrow$ | AUPR$\uparrow$ | FPR95$\downarrow$ |
> |-----------|--------------|----------|------------------|-------|-------|-------|
> | CIFAR-10  | 200          | DenseNet | Atypical                | 81.45 | 82.40 | 62.10 |
> | CIFAR-10  | 200          | DenseNet | Typical                | **82.86** | **83.48** | **60.01** |
> | CIFAR-10  | 200          | WRN-40-4 | Atypical                | 85.13 | 86.57 | 66.41 |
> | CIFAR-10  | 200          | WRN-40-4 | Typical                | **86.26** | **86.89** | **59.93** |
> | CIFAR-100 | 1000         | DenseNet | Atypical                | 71.96 | 73.16 | 85.57 |
> | CIFAR-100 | 1000         | DenseNet | Typical                | **74.79** | **75.83** | **80.97** |
> | CIFAR-100 | 1000         | WRN-40-4 | Atypical                | 66.64 | 67.41 | 86.92 |
> | CIFAR-100 | 1000         | WRN-40-4 | Typical                | **71.95** | **72.02** | **80.00** |
>
> **Table 6: Fine-tuning on typical/atypical CIFAR-10 samples with DenseNet-101**
>
> | D$_{in}$ | Dataset size | ACC  | Atypical/Typical | AUROC$\uparrow$ | AUPR$\uparrow$ | FPR95$\downarrow$  |
> |----------|--------------|------|------------------|-----------------|----------------|--------------------|
> | CIFAR-10 | 200          | 3.50 | Atypical                | 81.45           | 82.40          | 62.10              |
> | CIFAR-10 | 200          | 100.00| Typical                | **82.86**           | **83.48**          | **60.01**              |
> | CIFAR-10 | 350          | 11.14| Atypical                | 85.90           | 86.01          | 55.10              |
> | CIFAR-10 | 350          | 100.00| Typical                | **85.90**           | **86.16**          | **52.81**              |
> | CIFAR-10 | 500          | 16.80| Atypical                | 84.94           | 85.33          | 59.27              |
> | CIFAR-10 | 500          | 100.00| Typical                | **85.53**           | **86.10**          | **58.74**              |

---

> ### Author Response · Authors · 2022-11-15
> **Response to Reviewer miMq (2/4)**
>
>
> > **Q4:** Can we use other strategies like the early stopping criterion to prevent overfitting? The comparison would be interesting to see.
>
> **A4:** Thanks for your question. Here we report the extra experimental results compared with those methods for preventing overfitting (in Tables 1,2,3 and 4), like drop-out, regularization, and also early stopping with the validation dataset. The results show that most of the conventional strategies for preventing overfitting can gain limited benefit for OOD detection. The effective criterion, i.e., early stopping, also needs to change its validation target to be the OOD detection performance. It results from the different underlying principles of overfitting and the memorization on atypical samples in OOD detection, since the conceptual definition of overfitting is different from the latter. We will add the experimental resutls and corresponding discussion in our updated version.
>
>
> **Table 1: Comparison among overfitting methods and ODIN with DenseNet-101**
>
> | D$_{in}$ | Method    | AUROC$\uparrow$ | AUPR$\uparrow$ | FPR95$\downarrow$ | ID-ACC$\uparrow$ |
> |----------|-----------------------|-------|-------|-------|---------|
> | CIFAR-10 | Baseline              | 91.67 | 91.89 | 40.74 | 93.67   |
> | CIFAR-10 | Early Stopping w. ACC | 92.13 | 92.46 | 38.86 | 93.69   |
> | CIFAR-10 | Early Stopping w. OOD | 92.95 | 93.26 | 35.23 | 93.18   |
> | CIFAR-10 | Weight Decay 0.1      | 86.64 | 86.67 | 60.07 | 88.53   |
> | CIFAR-10 | Weight Decay 0.01     | 90.76 | 91.25 | 44.20 | 92.07   |
> | CIFAR-10 | Weight Decay 0.001    | 88.93 | 88.25 | 48.95 | 94.26   |
> | CIFAR-10 | Drop Rate 0.3         | 91.14 | 92.21 | 46.58 | 90.05   |
> | CIFAR-10 | Drop Rate 0.4         | 84.94 | 86.62 | 62.52 | 82.55   |
> | CIFAR-10 | Drop Rate 0.5         | 83.75 | 85.17 | 62.17 | 75.31   |
> | CIFAR-10 | UM                    | 92.45 | 93.06 | 37.13 | 92.76   |
> | CIFAR-10 | UMAP                  | 91.92 | 92.88 | 37.69 | 93.69   |
>
> **Table 2: Comparison among overfitting methods and Energy with DenseNet-101**
>
> | D$_{in}$ | Method    | AUROC$\uparrow$ | AUPR$\uparrow$ | FPR95$\downarrow$ | ID-ACC$\uparrow$ |
> |----------|-----------------------|-------|-------|-------|---------|
> | CIFAR-10 | Baseline              | 92.72 | 93.48 | 38.30 | 93.67   |
> | CIFAR-10 | Early Stopping w. ACC | 92.75 | 93.54 | 37.84 | 93.69   |
> | CIFAR-10 | Early Stopping w. OOD | 93.04 | 93.78 | 36.56 | 93.18   |
> | CIFAR-10 | Weight Decay 0.1      | 86.78 | 88.04 | 65.08 | 88.53   |
> | CIFAR-10 | Weight Decay 0.01     | 90.86 | 91.77 | 47.64 | 92.07   |
> | CIFAR-10 | Weight Decay 0.001    | 90.68 | 90.90 | 47.38 | 94.26   |
> | CIFAR-10 | Drop Rate 0.3         | 90.52 | 91.79 | 51.23 | 90.05   |
> | CIFAR-10 | Drop Rate 0.4         | 84.39 | 86.43 | 68.17 | 82.55   |
> | CIFAR-10 | Drop Rate 0.5         | 83.29 | 85.14 | 68.75 | 75.31   |
> | CIFAR-10 | UM                    | 93.58 | 94.14 | 33.66 | 92.76   |
> | CIFAR-10 | UMAP                  | 93.17 | 93.87 | 36.11 | 93.69   |
>
> **Table 3: Comparison among overfitting methods and ODIN with WRN-40-4**
>
> | D$_{in}$ | Method                | AUROC$\uparrow$ | AUPR$\uparrow$ | FPR95$\downarrow$ | ID-ACC$\uparrow$  |
> |----------|-----------------------|-----------------|----------------|-------------------|-------------------|
> | CIFAR-10 | Baseline              | 86.24           | 85.90          | 60.13             | 93.86             |
> | CIFAR-10 | Early Stopping w. ACC | 83.80           | 83.30          | 65.13             | 93.99             |
> | CIFAR-10 | Early Stopping w. OOD | 89.92           | 90.94          | 49.87             | 91.47             |
> | CIFAR-10 | Weight Decay 0.1      | 84.38           | 84.75          | 65.75             | 89.88             |
> | CIFAR-10 | Weight Decay 0.01     | 88.08           | 88.45          | 55.16             | 93.16             |
> | CIFAR-10 | Weight Decay 0.001    | 86.34           | 86.38          | 57.42             | 94.91             |
> | CIFAR-10 | Drop Rate 0.3         | 87.53           | 87.25          | 56.12             | 94.22             |
> | CIFAR-10 | Drop Rate 0.4         | 88.24           | 88.41          | 54.62             | 94.20             |
> | CIFAR-10 | Drop Rate 0.5         | 89.13           | 89.99          | 53.07             | 93.91             |
> | CIFAR-10 | UM                    | 89.61           | 91.13          | 50.97             | 92.68             |
> | CIFAR-10 | UMAP                  | 90.43           | 91.73          | 46.96             | 93.86             |

---

> ### Author Response · Authors · 2022-11-15
> **Response to Reviewer miMq (1/4)**
>
>
> Thank you for your time devoted to reviewing this paper and your constructive suggestions. Here are our detailed replies to your questions.
>
> > **Q1:** Why can the atypical samples be mined by the layer-wise mask? I was lost in reading section 3.2. More explanations and details can be elaborated on there. Why use the layer-wised mask? Is it the optimal choice for identifying the atypical samples? Can we use a unit/weight mask or use high CE error to identify atypical samples?
>
> **A1:** Thanks for your question and we will merge the explanation below into our updated version to make it clearer.
>
> First, for identifying those atypical samples using a layer-wise mask with the well-pre-trained model, the core intuition behind is constructing the parameter-level discrepancy to mine the atypical samples. It is inspired by and based on the evidence drawn from previous literature about learning behaviors [1,2] of deep neural networks (DNNs) and sparse representation [3,4,5]. To be specific, the atypical samples tend to be learned by the DNNs later than those typical samples [1], and are relatively more sensitive to the changes of the model parameter as the model does not generalize well on that. By the layer-wise mask, the constructed discrepancy can make the model misclassify the atypical samples and estimate loss constraint for the forgetting objective, as visualized in Figure 3\(c).
>
> Second, introducing the layer-wise mask has several advantages for achieving the staged target of mining atypical samples in our proposed method, while we would also admit that the layer-wise mask is not an irreplaceable option or maybe not optimal. On the one hand, considering that the model has been trained to approach the zero error on training data, utilizing the layer-wise mask is an integrated strategy to 1) figure out the atypical samples and 2) obtain the loss value computed by the masked output that misclassifies them. The loss constraint is later used in the forgetting objective to fine-tune the model. On the other hand, the layer-wise mask is also compatible with the proposed UMAP to generate a flexible mask for restoring the detection capability of the original model.
>
> Third, we also adopt the unit/weight mask and visualize the misclassified samples following your constructive conjecture. We think they can also be used to mine the atypical samples and can be extended or improved to be a more flexible choice. Further investigating the specific effect of different methods that construct the parameter-level discrepancy would be an interesting sub-topic in future work. For the value of CE loss, although the atypical samples tend to have high CE loss value, they are already memorized and correctly classified as indicated by the zero training error. Only using the high CE error can not provide the loss estimation when the model does not correctly classify those samples.
>
> [1] Arpit, Devansh, et al. "A closer look at memorization in deep networks." International conference on machine learning. PMLR, 2017.
>
> [2] Goodfellow, Ian, Yoshua Bengio, and Aaron Courville. Deep learning. MIT press, 2016.
>
> [3] Frankle, Jonathan, and Michael Carbin. "The Lottery Ticket Hypothesis: Finding Sparse, Trainable Neural Networks." International Conference on Learning Representations. 2018.
>
> [4] Goodfellow, Ian J., et al. "Challenges in representation learning: A report on three machine learning contests." International conference on neural information processing. Springer, Berlin, Heidelberg, 2013.
>
> [5] Barham, Paul, et al. "Pathways: Asynchronous distributed dataflow for ML." Proceedings of Machine Learning and Systems 4 (2022): 430-449.
>
>
> > **Q2:** Are the masks applied at all layers or just some typical layers?
>
> The masks are applied to all layers including the convolutional layers and the fully connected layer, which is consistent with the mask generation in the conventional pruning pipeline.
>
>
> > **Q3:** How does Equation 3 stabilize around the stage that can forget those atypical samples? Seems like the absolute value term makes sure the masked loss is close to the original loss. The second term is minimizing the masked loss. Where is the "forgetting" mechanism happening?
>
> **A3:** The current equation 3 has a typo in the second loss term. We would revise $\widehat{\ell}_\text{CE}(m_\delta\odot f)$ to be $\widehat{\ell}_\text{CE}(m_\delta\odot f^*)$, where $m_\delta \odot f^*$ denotes the masked output of the fixed pretrained model parameter, which are used to estimate the loss constraint for the forgetting objective. The overall objective of forgetting encourages the model to be optimized to finally stabilize around the estimated loss value, and to approach the stage that can forget the atypical samples and restore the detection capability.

---

> ### Author Response · Authors · 2022-11-22
> **Looking forward to your response or further discussion in Discussion Stage 2**
>
> Dear Reviewer miMq,
>
> Thanks again for your time and valuable comments! In Discussion Stage 1, we have carefully considered your initial advice/questions and provided the individual responses (e.g., A1-A5 in the [details](https://openreview.net/forum?id=K2OixmPDou3&noteId=igZIOW_YFJ)) with the revised submission (and a [shortened summary](https://openreview.net/forum?id=K2OixmPDou3&noteId=Cy5l4oeMb6n)) based on your constructive suggestions.
>
> In Discussion Stage 2, would you mind confirming whether you are satisfied with our responses and revision? We are more than glad to have further discussion with you if there exists any other suggestion. Thanks!
>
> Best,
>
> Authors of Paper1069

---

### Official Review · Reviewer_PAPe · 2022-10-25

**Confidence:** 3
**Correctness:** 2
**Technical Novelty And Significance:** 2
**Empirical Novelty And Significance:** 3
**Recommendation:** 5

**Clarity, Quality, Novelty And Reproducibility:**

The paper is written about average -- better than some papers but overall there is a lack of clarity and a need for editorial work.

It's not clear what is meant in many places.

While it's helpful to add 10 pages of Appendix, it also felt a bit slapped together. The paper should be able to stand as-is within the paper limits.

Some example edits and specific comments:

Abstract: "to reveal the once-covered detection capability" --> unclear what is meant by "covered". Perhaps "that restores the OOD discriminative capabilities of the model"?

p. 1: "how to find" --> "how can one find" or "how can we find"

p. 1: "Especially, it is a general existence across various setups," --> unclear. Maybe "This is generally true across different models"?

Fig. 1: "FPR95" should be defined here rather than page 7 or  forcing the reader to refer to another citation. "false positive rate (FPR95) of OOD examples when true positive rate of in-distribution examples is at 95%"

Fig. 1:  "for multiple times" --> " multiple times"

Fig. 1: "More exploration can refer to Figure 2." --> "Figure 2 contains further exploration."

p. 2: "expected stage." --> unclear what this refers to.

p. 2: "results accordingly demonstrate their rationality.", perhaps consider:  "results accordingly demonstrate their effectiveness" because experimental results cannot show rationality.

Figure 3: "samples that sensitive" --> "samples that are sensitive"

Figure 3: "Then fine-tune" --> "Then we fine-tune"?

p. 6 "CE loss" --> at least the first time, please use "cross-entropy (CE) loss"

p. 7 "existential" --> "existing"

Figure 2: The caption should indicate the primary training task and dataset.


**Strength And Weaknesses:**

The paper presents experiments training on CIFAR10 and CIFAR100, then measuring classification accuracy as well as OOD false positive rates. The proposed method appears to work well across many OOD sets. However, I do have a few questions.

While I can understand the basic idea linking the memorization of atypical ID examples to a decrease in OOD discrimination ability as a motivation for UM, I am not sure that this is solidly supported by the evidence of the paper.

Further, it isn't clear to me exactly how the parameters of the key equation (3) are selected. According to the paper: "For
CIFAR-10 as ID datasets, the value of mask rate is 97.5% and the estimated loss constraint for
forgetting is 0.1 for our tuning until the convergence; For CIFAR-100, the value of mask rate is
97% and the estimated loss constraint for forgetting is 1.2 for our tuning until the convergence."
Would it be possible to run a baseline with parameters selected to effectively turn off masking (i.e., mask rate 100%)?

Would it be possible to run CIFAR100 for OOD and add to Table 2?


**Summary Of The Paper:**

The paper proposes Unleashing Mask (UM) with the goal of maintaining a model's capability to distinguish between out-of-distribution (OOD) and in-distribution (ID) training examples.

The paper posits that as classification models train and the training loss decreases, there is a point where simple measures such as ODIN effectively distinguish between ID and OOD, but then as training loss trends to zero, the model is forced to learn atypical and difficult, in-domain class examples which degrades. While this atypical example memorization does improve model accuracy at the primary task, it also harms the model's performance for OOD.

To address this, the UM method specifies a binary layer-wise mask that serves to force forgetting of atypical examples.

**Summary Of The Review:**

The paper presents the idea of using forgetting of atypical examples via UM to preserve a models' ability to distinguish OOD.
While experiments support the overall proposal, it seems there are some additional probing or analysis that could be performed to more deeply try to understand whether the improvement is coming from the model or from some other factors. The paper has some issues with clarity and would benefit from a heavy edit.

---

> ### Author Response · Authors · 2022-11-15
> **Response to Reviewer PAPe (3/3)**
>
>
> > **Q2:** how the parameters of the key equation (3) are selected. According to the paper: "For CIFAR-10 as ID datasets, the value of mask rate is 97.5% and the estimated loss constraint for forgetting is 0.1 for our tuning until the convergence; For CIFAR-100, the value of mask rate is 97% and the estimated loss constraint for forgetting is 1.2 for our tuning until the convergence."
>
> **A2:** To choose the parameters of the estimated loss constraint, we use the TinyImageNet dataset as the validation set, which is not seen during training and is not considered in our evaluation of OOD detection performance.
>
> To investigate the effect of the parameter used in the experiments, we also provide the ablation study and present the results corresponding to the different parameters used in key Eq.(3) in Figure 10 using WideResNet-40-4. The results show that a wide range of mask ratios (i.e., from 96% to 99%) to estimate the loss constraint used in Eq.(3) can gain better OOD detection performance than the baseline. It shows the mask ratio would be robust to hyper-parameter selection under a certain value.
>
> Since the core intuition behind our method is to restore the OOD detection performance starting from the well-trained model stage, forgetting a relatively small portion (empirically found around 97% mask ratio) of atypical samples can be beneficial. To find the optimal parameter for tuning, more advanced searching techniques like AutoML or validation design based on the important observation in our work may be further employed in the future.
>
>
> > **Q3:** Would it be possible to run a baseline with parameters selected to effectively turn off masking (i.e., mask rate 100%)?
>
> **A3:** If we understand correctly, turning off the masking from UM would degenerate to the original baseline only trained on the original classification task, and turning off the masking from UMAP would degenerate to our proposed UM to optimize the original model parameter.
>
> In the UM, we propose to finetune the model with a constrained loss objective to forget the atypical samples to restore the detection capability, where we directly optimize the original model parameter without the mask. The mask in UM is introduced to construct the parameter-level discrepancy for estimating the loss constraint for forgetting. We would revise the second term of Eq.(3), i.e.,$\widehat{\ell}_\text{CE}(m_\delta\odot f)$ to be $\widehat{\ell}_\text{CE}(m_\delta\odot f^*)$, where $m_\delta \odot f^*$ denotes the masked output of the fixed given model parameter. Keeping the second part of Eq.(3) unchanged, in our proposed UMAP, we will learn a mask based on the original model parameter to achieve the forgetting target, where the $\ell_\text{CE}(f)$ would be replaced to $\ell_\text{CE}(m \odot f)$ in Eq.(3). We will re-clarify the above in our updated version to make it clear.
>
>
> > **Q4:** Would it be possible to run CIFAR100 for OOD and add to Table 2?
>
> **A4:** The complete fine-grained results of the experiments on CIFAR-100 is provided in Table 5 of Appendix C. In addition, we also provide similar results using another model structure in Table 8. We will note it in the experimental section of our main text and make it clearer and more concise with our comprehensive empirical results.
>
>
> > **Q5:** While it’s helpful to add 10 pages of Appendix, it also felt a bit slapped together. The paper should be able to stand as-is within the paper limits. Some example edits and specific comments ...
>
> **A5:** Thanks for your suggestion on improving the presentation quality of our work, we have carefully addressed the clarity issues and re-organize our main text to be clearer and condensed.

---

> ### Author Response · Authors · 2022-11-15
> **Response to Reviewer PAPe (2/3)**
>
>
> **Table 4: Fine-tuning on typical/atypical CIFAR-100 samples with DenseNet-101**
>
> | D$_{in}$  | Dataset size | ACC  | Atypical/Typical | AUROC$\uparrow$ | AUPR$\uparrow$ | FPR95$\downarrow$  |
> |-----------|--------------|------|------------------|-----------------|----------------|--------------------|
> | CIFAR-100 | 500          | 1.00 | Atypical                | 72.69           | 73.28          | 80.71              |
> | CIFAR-100 | 500          | 100.00| Typical                | **74.07**           | **75.20**          | **80.19**              |
> | CIFAR-100 | 800          | 2.88 | Atypical                | 69.74           | 71.15          | 85.46              |
> | CIFAR-100 | 800          | 100.00| Typical                | **72.49**           | **73.17**          | **81.97**              |
> | CIFAR-100 | 1000         | 3.50 | Atypical                | 71.96           | 73.16          | 85.57              |
> | CIFAR-100 | 1000         | 100.00| Typical                | **74.79**           | **75.83**          | **80.97**              |
>
> **Table 5: Fine-tuning on typical/atypical CIFAR-100 samples with WRN-40-4**
>
> | D$_{in}$  | Dataset size | ACC  | Atypical/Typical | AUROC$\uparrow$ | AUPR$\uparrow$ | FPR95$\downarrow$  |
> |-----------|--------------|------|------------------|-----------------|----------------|--------------------|
> | CIFAR-100 | 500          | 1.00 | Atypical                | 66.03           | 66.17          | 89.56              |
> | CIFAR-100 | 500          | 100.00| Typical                | **68.60**            | **69.93**          | **86.53**              |
> | CIFAR-100 | 800          | 2.88 | Atypical                | 67.59           | 68.66          | 85.61              |
> | CIFAR-100 | 800          | 100.00| Typical                | **70.25**           | **68.95**          | **79.66**              |
> | CIFAR-100 | 1000         | 3.50 | Atypical                | 66.64           | 67.41          | 86.92              |
> | CIFAR-100 | 1000         | 100.00| Typical                | **71.95**           | **72.02**          | **80.00**              |
>
> By the condition controlled in the experiments and our performance comparison with those baselines considered in our work, the improvement is mainly from its adjusted learning behavior instead of the model structure since we didn’t change the model structure using our mask in the UM. The empirical results show that learning on those atypical samples fails to draw the suitable features for the OOD detection task also it still can improve the original task performance. We will add the experimental results as well as further discussion to our updated version.
>
> (A brief description of the above experiments for reference.) Intuitively, we intend to separate the training dataset into a typical set and an atypical set and train respectively on these two sets to see whether it is learning atypical samples that causes OOD performance to decrease during the latter part of the training phase. We force training samples through the model (DenseNet-101) of the 60th epoch and get the CE loss for separation. We provide the ACC of the generated sets on the model of the 60th epoch (ACC in the above Tables 2-5). The extremely low ACCs of the atypical sets show that the model of the 60th epoch can hardly predict the samples, which meets our definition of the atypical sample. We then fine-tune the model of the 60th epoch with the generated dataset report the OOD performance. As suggested by the results, training on atypical samples can undermine the OOD performance, compared to those of typical samples.

---

> ### Author Response · Authors · 2022-11-15
> **Response to Reviewer PAPe (1/3)**
>
> Thank you for your time devoted to reviewing this paper and your constructive suggestions. Here are our detailed replies to your questions.
>
> > **Q1:** While I can understand the basic idea linking the memorization of atypical ID examples to a decrease in OOD discrimination ability as a motivation for UM, I am not sure that this is solidly supported by the evidence of the paper. While experiments support the overall proposal, it seems there are some additional probing or analysis that could be performed to more deeply try to understand whether the improvement is coming from the model or from some other factors.
>
> **A1:** Thanks for your constructive comments. In the following tables (Tables 1-5), we further conduct the experiments to identify the negative effect of learning on those atypical samples by comparing with a counterpart that learning only with the typical samples. The results confirm that the degeneration on OOD detection performance is more likely to come from learning atypical samples.
>
> **Table 1: Fine-tuning on typical/atypical samples with different model structures**
>
> | D$_{in}$  | Dataset size | Structure | Atypical/Typical | AUROC$\uparrow$ | AUPR$\uparrow$ | FPR95$\downarrow$ |
> |-----------|--------------|----------|------------------|-------|-------|-------|
> | CIFAR-10  | 200          | DenseNet | Atypical                | 81.45 | 82.40 | 62.10 |
> | CIFAR-10  | 200          | DenseNet | Typical                | **82.86** | **83.48** | **60.01** |
> | CIFAR-10  | 200          | WRN-40-4 | Atypical                | 85.13 | 86.57 | 66.41 |
> | CIFAR-10  | 200          | WRN-40-4 | Typical                | **86.26** | **86.89** | **59.93** |
> | CIFAR-100 | 1000         | DenseNet | Atypical                | 71.96 | 73.16 | 85.57 |
> | CIFAR-100 | 1000         | DenseNet | Typical                | **74.79** | **75.83** | **80.97** |
> | CIFAR-100 | 1000         | WRN-40-4 | Atypical                | 66.64 | 67.41 | 86.92 |
> | CIFAR-100 | 1000         | WRN-40-4 | Typical                | **71.95** | **72.02** | **80.00** |
>
> **Table 2: Fine-tuning on typical/atypical CIFAR-10 samples with DenseNet-101**
>
> | D$_{in}$ | Dataset size | ACC  | Atypical/Typical | AUROC$\uparrow$ | AUPR$\uparrow$ | FPR95$\downarrow$  |
> |----------|--------------|------|------------------|-----------------|----------------|--------------------|
> | CIFAR-10 | 200          | 3.50 | Atypical                | 81.45           | 82.40          | 62.10              |
> | CIFAR-10 | 200          | 100.00| Typical                | **82.86**           | **83.48**          | **60.01**              |
> | CIFAR-10 | 350          | 11.14| Atypical                | 85.90           | 86.01          | 55.10              |
> | CIFAR-10 | 350          | 100.00| Typical                | **85.90**           | **86.16**          | **52.81**              |
> | CIFAR-10 | 500          | 16.80| Atypical                | 84.94           | 85.33          | 59.27              |
> | CIFAR-10 | 500          | 100.00| Typical                | **85.53**           | **86.10**          | **58.74**              |
>
> **Table 3: Fine-tuning on typical/atypical CIFAR-10 samples with WRN-40-4**
>
> | D$_{in}$ | Dataset size | ACC  | Atypical/Typical | AUROC$\uparrow$ | AUPR$\uparrow$ | FPR95$\downarrow$  |
> |----------|--------------|------|------------------|-----------------|----------------|--------------------|
> | CIFAR-10 | 200          | 3.50 | Atypical                | 85.13           | 86.57          | 66.41              |
> | CIFAR-10 | 200          | 100.00| Typical                | **86.26**           | **86.89**          | **59.93**              |
> | CIFAR-10 | 350          | 11.14| Atypical                | 82.92           | 84.24          | 68.57              |
> | CIFAR-10 | 350          | 100.00| Typical                | **85.82**           | **87.84**          | **65.54**              |
> | CIFAR-10 | 500          | 16.80| Atypical                | 82.88           | 83.22          | 66.75              |
> | CIFAR-10 | 500          | 100.00| Typical                | **87.38**           | **87.93**          | **52.27**              |

---

> ### Author Response · Authors · 2022-11-17
> **Would you mind confirming if you have further questions? Thanks!**
>
> Dear Reviewer PAPe,
>
> We appreciate your efforts and time in reviewing our paper! We have carefully considered your concerns/advice and provided as much refinement and experiments to address them regarding the submission. Would you mind checking our response, and confirming if you have further questions?
>
> Best,
>
> Authors

---

> ### Author Response · Authors · 2022-11-18
> **Would you mind checking our response? Welcome for more discussions.**
>
> Dear Reviewer PAPe,
>
> Thanks again for your time and efforts in reviewing our paper. As the end of Discussion Stage 1 is approaching and the window for paper revision is closing, here is a summary of our previous response and update:
>
> - Added supportive experiments to tamp the association between our motivation and the proposed methods (see Appendix F).
> - Expanded discussion about the hyper-parameter mask rate (see Appendix F).
> - Clarified the arrangement of experiments of CIFAR-100 (see Section 4.2).
> - Carefully revised the clarity issues (see revision).
>
> **We humbly expect you could check our detailed responses with our updated version, and confirm whether our response has addressed your concerns. More discussions are always welcome. Please let us know if there are any further questions or suggestions that we could clarify or improve.**
>
> Best,
>
> Authors

---

> ### Author Response · Authors · 2022-11-22
> **Need further clarification? Looking forward to your response or further discussions in Discussion Stage 2**
>
> Dear Reviewer PAPe,
>
> Thanks again for your time and valuable comments! In Discussion Stage 1, we have carefully considered your initial advice/questions and provided the individual responses (e.g., A1-A5 in the [details](https://openreview.net/forum?id=K2OixmPDou3&noteId=5KrglufT1YA)) with the revised submission (and a [shortened summary](https://openreview.net/forum?id=K2OixmPDou3&noteId=Cy5l4oeMb6n)) based on your constructive suggestions.
>
>
> - Overall, we provided a comprehensive exploration of the research problem: unleashing the intrinsic Out-Of-Distribution discriminative capability,
>     - We revealed an interesting and important phenomenon:
>         - Verified the observation from various perspectives, e.g., different ID/OOD datasets, different model structures, different training conditions (e.g., learning rate), different OOD baselines **(e.g., Figures 1/2\(c)-(d)/4(a)-(b)/5/6/7/9, Tables 4/5/6/7)**;
>     - We further delved into the observation and figured out the potential attribution
>         - Identified the memorization on atypical samples **(e.g., Figures 2(a)-(b)(e)-(f)/3(b)-\(c)/12)**;
>         - Provided empirical verification of negative effects from training on atypical samples **(e.g., Tables 9/10/11/12/13)**;
>     - Motivated our proposed UM and UMAP:
>         - Broad performance comparison with competitive baselines and advanced method **(e.g., Tables 1/2/8/16/17/18/19/20)**;
>         - Ablation study on our objective for forgetting, method efficiency, parameter-robustness (e.g., mask rate, prune rate), and practicality **(e.g., Figures 4\(c)-(e)/8/10, Tables 14/15)**;
>         - Other issues about clarification and investigation (e.g., the difference from overfitting)**(e.g., Figures 11, Tables 4/5/6/7)**.
>
> **In Discussion Stage 2**, we are actively available for further clarification and discussion with you in the openreview system if there are any unclear parts or concerns/questions. **We would appreciate it if you could confirm whether you are satisfied with our response, and we will do our best to address any further questions/suggestions. Thanks!**
>
> Best,
>
> Authors of Paper1069

---

> ### Author Response · Authors · 2022-11-24
> **Looking forward to your responses or further suggestions/comments!**
>
> Dear Reviewer PAPe,
>
> We appreciate your efforts and time in reviewing our paper! We have carefully considered and addressed your initial concerns regarding our submission. Would you mind checking our response, and confirming if you have further questions? If any, we will address them to improve our submission.
>
> Best,
>
> Authors of Paper1069

---

> ### Author Response · Authors · 2022-11-25
> **Looking forward to your responses or further suggestions/comments!**
>
> Dear Reviewer PAPe,
>
> We appreciate your efforts and time in reviewing our paper! We have carefully considered your concerns/advice and provided as much refinement and experiments to address them regarding the submission. Would you mind checking our response, and confirming if you have further questions? If any, we will address them to improve our submission.
>
> Best,
>
> Authors of Paper1069

---

> ### Author Response · Authors · 2022-11-26
> **Looking forward to your responses or further suggestions/comments!**
>
> Dear Reviewer PAPe,
>
> We appreciate your efforts and time in reviewing our paper! We have carefully considered your concerns/advice and provided as much refinement and experiments to address them regarding the submission. Would you mind checking our response, and confirming if you have further questions? If any, we will address them to improve our submission.
>
> Best,
>
> Authors of Paper1069

---

> ### Author Response · Authors · 2022-11-27
> **Looking forward to your responses or further suggestions/comments!**
>
> Dear Reviewer PAPe,
>
> We appreciate your efforts and time in reviewing our paper! We have carefully considered your concerns/advice and provided as much refinement and experiments to address them regarding the submission. Would you mind checking our response, and confirming if you have further questions? If any, we will address them to improve our submission.
>
> Best,
>
> Authors of Paper1069

---

> ### Author Response · Authors · 2022-11-30
> **Looking forward to your responses or further suggestions/comments!**
>
> Dear Reviewer PAPe,
>
> We appreciate your efforts and time in reviewing our paper! We have carefully considered your concerns/advice and provided as much refinement and experiments to address them regarding the submission.
>
> We understand you might be quite busy. However, as the discussion deadline is approaching, would you mind checking our response and confirming whether you have any further questions?
>
> Best,
>
> Authors of Paper1069

---

> ### Author Response · Authors · 2022-12-02
> **Would you mind confirming if you have further questions? Thanks!**
>
> Dear Reviewer PAPe,
>
> We appreciate your efforts and time in reviewing our paper! We have carefully considered your initial concerns/advice and provided as much refinement and experiments to address them regarding the submission.
>
> We understand you might be quite busy. However, as the discussion deadline is approaching and we have not received any feedback from you yet since the discussion phase began, would you mind checking our response (e.g., A1-A5 in the [details](https://openreview.net/forum?id=K2OixmPDou3&noteId=5KrglufT1YA), [clarification summary](https://openreview.net/forum?id=K2OixmPDou3&noteId=WY2GkJchCz), and [revision summary](https://openreview.net/forum?id=K2OixmPDou3&noteId=Cy5l4oeMb6n)) and confirming whether you have any further questions?
>
> If you have any more questions, we are happy to discuss them in the openreview system and will do our best to address them!
>
> Best,
>
> Authors of Paper1069

---

> ### Author Response · Authors · 2022-12-04
> **As the discussion phase is near the end, would you mind confirming if you have further questions?**
>
> Dear Reviewer PAPe,
>
> We appreciate your efforts and time in reviewing our paper! We have carefully considered your initial concerns/advice and provided as much refinement and experiments to address them regarding the submission.
>
> We understand you might be quite busy. However, as the discussion deadline is approaching and we have not received any feedback from you yet since the discussion phase began, would you mind checking our response (e.g., A1-A5 in the [[details]](https://openreview.net/forum?id=K2OixmPDou3&noteId=5KrglufT1YA), [[clarification summary]](https://openreview.net/forum?id=K2OixmPDou3&noteId=WY2GkJchCz), and [[revision summary]](https://openreview.net/forum?id=K2OixmPDou3&noteId=Cy5l4oeMb6n)) and confirming whether you have any further questions?
>
> If you have any more questions, we are happy to discuss them in the openreview system and will do our best to address them!
>
> Best,
>
> Authors of Paper1069

---

> ### Author Response · Authors · 2022-12-06
> **As the discussion phase is near the end, would you mind confirming if you have further questions?**
>
> Dear Reviewer PAPe,
>
> We appreciate your efforts and time in reviewing our paper! We have carefully considered your concerns/advice and provided as much refinement and experiments to address them regarding the submission.
>
> We understand you might be quite busy. However, as the discussion deadline is approaching and we have not received any feedback from you yet since the discussion phase began, would you mind checking our response (e.g., A1-A5 in the [[details]](https://openreview.net/forum?id=K2OixmPDou3&noteId=5KrglufT1YA), [[clarification summary]](https://openreview.net/forum?id=K2OixmPDou3&noteId=WY2GkJchCz), and [[revision summary]](https://openreview.net/forum?id=K2OixmPDou3&noteId=Cy5l4oeMb6n)) and confirming whether you have any further questions?
>
> If you have any more questions, we are happy to discuss at the openreview system and will do our best to address them!
>
> Best,
>
> Authors of Paper1069

---

> ### Author Response · Authors · 2022-12-07
> **To Reviewer PAPe:  Would you mind checking our response? Thanks!**
>
> Dear Reviewer PAPe,
>
> We appreciate your efforts and time in reviewing our paper! We have carefully considered your concerns/advice and provided as much refinement and experiments to address them regarding the submission.
>
> We understand you might be quite busy. However, as the discussion deadline is approaching and we have not received any feedback from you yet since the discussion phase began, would you mind checking our response (e.g., A1-A5 in the [[details]](https://openreview.net/forum?id=K2OixmPDou3&noteId=5KrglufT1YA), [[clarification summary]](https://openreview.net/forum?id=K2OixmPDou3&noteId=WY2GkJchCz), and [[revision summary]](https://openreview.net/forum?id=K2OixmPDou3&noteId=Cy5l4oeMb6n)) and confirming whether you have any further questions?
>
> If you have any more questions, we are happy to discuss at the openreview system and will do our best to address them!
>
> Best,
>
> Authors of Paper1069

---

### Official Review · Reviewer_Ga94 · 2022-10-30

**Confidence:** 4
**Correctness:** 3
**Technical Novelty And Significance:** 2
**Empirical Novelty And Significance:** 1
**Recommendation:** 3

**Clarity, Quality, Novelty And Reproducibility:**

Clarity: Parts of the paper lacks explanation and thus needs clarification (see questions).

Quality: The paper lacks appropriate theoretical analysis or detailed/thorough empirical support.

Novelty: Fine-tuning a pretrained noisy model for another hundreds of epochs, rather than robustly training a model in scratch not to include noise while achieving performance, seems unreasonable.

Reproducibility: Because authors noted that they will upload their codes and datasets during discussion phase for reviewing purposes in APPENDIX, so the reproducibility cannot be checked at the moment.

[1] KDD 2021, Robust Learning by Self-Transition for Handling Noisy Labels.

**Strength And Weaknesses:**

Strength: Adaptability to previous robust loss functions.

Detailed information on baselines in appendix A is helpful for readers who needs baseline details in a page.

Weaknesses
1. Expensive: Because UM denoises a pretrained model via fine-tuning about 100 (I believe so based on several empirical results in the table although not specified for fine-tuning; authors need to clarify it) epochs, I don't think it worths such expensive fine-tuning process.
2. Weak logical support: In addition to Weaknesses 1, if we can access to already deployed model and its training data, it worths applying UM or UMAP to a model when it's at best condition (at epoch 60 according to the authors) and help it learn only in-distribution data. Why should we start as fine-tuning?
3. Labor-intensive: mask rate should be empirically found by UM users. Besides, we need to choose first the OOD baseline, second UM or UMAP to obtain the best result because no case consistently performs the best.
4. Lack of empirical support: We need to evaluate the impact of UM or UMAP, but results such as Table 1 and 2 lacks their results (I understand the authors' intention to emphasize the extendability of UM and UMAP.)
5. No theoretical analysis: To support the authors' claim on UM, even a typical theoretical analysis is required which is absent.

Questions
1. Why UM model itself doesn't have (i.e., FC + UM or FC + UMAP)?
2. What is fine tuning epochs $k$ and loss constraint $zeta$?
3. Why only Fig 4(c) has 200 epochs? Besides, is 200 fine-tuning stage as a whole?
4. What are OOD recall before/after fine-tuning by UM/UMAP?
5. Some acronym words such as FPR95 needs explanation for readers without sufficient expertise or knowledge in this field.

**Summary Of The Paper:**

This paper is to restore the out-of-distribution sample detection ability given a well-trained model by fine-tuning with the proposed method Unleashing Mask (UM). Based on empirical observations, authors show that a model is trained to learn how to classify in-distribution samples while distinguishing out-of-distribution samples in early phase. Then, at epoch 60 in the authors' example, the model keeps learning in-distribution samples while losing its ability to distinguish out-of-distribution samples. To recover the model's ability to identify out-of-distribution samples, authors suggest UM that identifies atypical samples and helps the model forget them.

**Summary Of The Review:**

Based on the weaknesses and overall quality assessment, I would recommend reject of this paper. I welcome the enlightenment of the authors on my ignorance and misunderstanding if there is any.

---

> ### Author Response · Authors · 2022-11-15
> **Response to Reviewer Ga94 (4/4)**
>
> > **Q9:** What are OOD recall before/after fine-tuning by UM/UMAP?
>
> **A9:** Following the evaluation metrics of the related literature [1,2,3,4], the recall value is not adopted singly as it can not be computed without a specific threshold for the OOD detection task based on the scoring functions. In OOD detection, the general metric that is adopted is FPR95 (as explained in our evaluation metrics in Section 4.1) to evaluate the detection performance, which has the threshold at 95% True Positive Rate (TPR, or called "recall"). FPR95 indicates the probability for a negative sample to be misclassified as positive when the TPR is at 95%. The results in Table 1 or other tables in our appendix show that we decrease the false positive rate after fine-tuning with our UM/UMAP.
>
> [1] Hendrycks, Dan, Mantas Mazeika, and Thomas Dietterich. "Deep Anomaly Detection with Outlier Exposure." International Conference on Learning Representations. 2018.
>
> [2] Liang, Shiyu, Yixuan Li, and R. Srikant. "Enhancing The Reliability of Out-of-distribution Image Detection in Neural Networks." International Conference on Learning Representations. 2018.
>
> [3] Liu, Weitang, et al. "Energy-based out-of-distribution detection." Advances in Neural Information Processing Systems 33 (2020): 21464-21475.
>
> [4] Huang, Rui, Andrew Geng, and Yixuan Li. "On the importance of gradients for detecting distributional shifts in the wild." Advances in Neural Information Processing Systems 34 (2021): 677-689.
>
>
> > **Q10:** Some acronym words such as FPR95 needs explanation for readers without sufficient expertise or knowledge in this field.
>
> **A10:** Thanks for pointing this out, for FPR95 and other commonly used criteria in OOD detection tasks, we explained in detail in Section 4.1. False Positive Rate at 95% (FPR95) True Positive Rate (TPR) indicates the probability for a negative sample to be misclassified as positive when the TPR is at 95%. For those in the previous main text, we will accordingly add the explanation the first time it exists in our updated version. For the detailed formulations of FPR and TPR, we will also add them to our appendix for reference.
>
>
> > **Q11:** Novelty: Fine-tuning a pretrained noisy model for another hundreds of epochs, rather than robustly training a model in scratch not to include noise while achieving performance, seems unreasonable.
>
> **A11:** As for the novelty concern, we would like to re-clarify some major points and explain more to avoid some potential misunderstanding.
>
> First, there is no "noise" samples are considered or claimed in our setting, as all of the training data for the original classification task are in-distribution (ID) data with the correct label. We follow the conventional setting as the previous literature about scoring functions design or outlier exposure.
>
> In our work, we provide a new perspective to explore the OOD detection performance via backtracking the model training phase on the original task. It is different from most previous works that start with the well-trained model on ID data. Empirically, we reveal a general existence of the potential OOD detection capability of the well-trained model and discover the inconsistency between gaining better OOD detection capability with pursuing better classification performance on ID data. By further comparison of the training dynamics, we identify learning on those samples with relatively atypical semantic features to pursuing lower training loss (for better performance in the original task) is harmful to OOD detection.
>
> Accordingly, we propose UM and UMAP to restore the intrinsic OOD detection capability of the given model by our newly designed objective for forgetting. It is empirically verified to be generally effective by our extensive experiments and also shows the advantages to be more efficient as indicated by our previous response.
>
> To the best of our knowledge, the important observation about the target-oriented discrepancy is not considered and received limited discussion in the previous literature on OOD detection. Without it, we would have no underlying principle to tuning or training the expected model for excavating the OOD detection capability.
>
> > **Q12:** Reproducibility: Because authors noted that they will upload their codes and datasets during discussion phase for reviewing purposes in APPENDIX, so the reproducibility cannot be checked at the moment.
>
> **A12:** Thank you for pointing out this. To avoid concerns about the reproducibility and the detailed setups in this work, we will open our source code in the anonymous repository.

---

> ### Author Response · Authors · 2022-11-15
> **Response to Reviewer Ga94 (3/4)**
>
>
> > **Q4:** Lack of empirical support: We need to evaluate the impact of UM or UMAP, but results such as Table 1 and 2 lacks their results (I understand the authors’ intention to emphasize the extendability of UM and UMAP.)
>
> Actually, we have conducted complete experiments about UM and UMAP under various settings to evaluate the general effectiveness of UM and UMAP. As indicated in Section 4.2 and Appendix B, please kindly refer to Tables 3-8 that we leave in the appendix due to the page limit.
>
> Extensive experiments show UMAP can achieve comparable or even better performance than UM. Considering the motivation, we propose UMAP to learn a mask instead of directly tuning the original model with our objective of forgetting. It would not affect the inference of the original model to maintain the ID-ACC. For the impact of UM and UMAP, UM is more serving a research proposal that verifies our method can restore the OOD detection capability by forgetting those atypical samples, and UMAP may be a more compatible and practical choice for usage.
>
>
> > **Q5:** No theoretical analysis: To support the authors’ claim on UM, even a typical theoretical analysis is required which is absent.
>
> **A5:** To our best knowledge, our work provides a new conceptual perspective to revisit the given model for the OOD detection task. We mainly focus on empirically discovering and understanding the important inconsistency of better OOD detection capability with original classification performance, which is not revealed and is limited discussed before. Accordingly, we propose UM/UMAP to restore it, and conduct extensive experiments to verify our conjecture and provide a comprehensive understanding of both the observation and methods. Similar to the previous work [1,2,3], as currently there is no appropriate theoretical framework in OOD detection to analyze those empirical innovations, we would like to leave it to future extension.
>
> [1] Hendrycks, Dan, Mantas Mazeika, and Thomas Dietterich. "Deep Anomaly Detection with Outlier Exposure." International Conference on Learning Representations. 2018.
>
> [2] Hendrycks, Dan, Kimin Lee, and Mantas Mazeika. "Using pre-training can improve model robustness and uncertainty." International Conference on Machine Learning. PMLR, 2019.
>
> [3] Liu, Weitang, et al. "Energy-based out-of-distribution detection." Advances in Neural Information Processing Systems 33 (2020): 21464-21475.
>
> [4] Huang, Rui, Andrew Geng, and Yixuan Li. "On the importance of gradients for detecting distributional shifts in the wild." Advances in Neural Information Processing Systems 34 (2021): 677-689.
>
>
> > **Q6:** Why UM model itself doesn’t have (i.e., FC + UM or FC + UMAP)?
>
> **A6:** From the perspective of motivation, our proposed UM and UMAP are naturally based on the important observation of the well-trained given model and are seeking to restore the OOD detection capability of the original model with the objective of forgetting. Hence, if we understand correctly, for the UM model, adding the UM tuning or UMAP again may have the same effects with the same underlying mechanism, which can degenerate into the general version of our UM and UMAP. As for the "FC", there may be a misunderstanding here that our masks are applied to all layers not only the fully connected layer.
>
>
> > **Q7:** What is fine tuning epochs $k$ and loss constraint $\zeta$?
>
> **A7:** The fine-tuning epochs $k$ is the epochs we fine-tune after we get the well-trained model; The loss constraint $\zeta$ is the estimated value by our introduced mask (it equals to the second term, i.e.,$\widehat{\ell}_\text{CE}(m_\delta\odot f^*)$ in Eq.(3) with the fixed given model $f^*$). We will revise the current typo in Eq.(3) and link it to our realization to make them clearer.
>
>
> > **Q8:** Why only Fig 4\(c) has 200 epochs? Besides, is 200 fine-tuning stage as a whole?
>
> **A8:** As indicated in Figure 4\(c) (i.e., "index" under the x-axis) and the description in Section 4.3, the x-axis represents the index of samples within a mini-batch instead of the epoch. We generate Figure 4\(c) by examining the CE loss of a training mini-batch (200 in this figure) respectively with and without our introduced mask, which is to show the effect of constructing the parameter-level discrepancy by our introduced mask.

---

> ### Author Response · Authors · 2022-11-15
> **Response to Reviewer Ga94 (2/4)**
>
> > **Q2:** Weak logical support: If we can access to already deployed model and its training data, it worths applying UM or UMAP to a model when it’s at best condition (at epoch 60 according to the authors) and help it learn only in-distribution data. Why should we start as fine-tuning?
>
> **A2:** First, we would like to clarify that all the training samples including those "atypical samples" are in-distribution (ID) data according to the definition. There is no OOD data whose label space is disjoint with the training samples considered in the training process.
>
> In this work, we focus on the general setting considered in previous literature (like scoring function design [1,2] or outlier exposure [3,4]) of OOD detection, in which we have the well-trained given model on the original task and the training data. Different from the previous methods that leverage the detection capability of the given models, we reconsider whether it is the appropriate model for OOD detection. Based on the important observation revealed in our work (i.e., the inconsistency between gaining better OOD detection capability and pursuing better performance on in-distribution data), we propose UM and UMAP to adjust the model to be more appropriate and compatible for OOD detection.
>
> It indicates that we keep the same starting point as previous works, and it is more general and backward compatible to those train-from-scratch cases or needs the middle checkpoints (that may not be always accessible in practice, like some storage-constrained limitations). That is to say, our UM can also be applied or extended to those situations. From the perspective of practicability and usability, our UM and UMAP also show the advantages on finetuning and pruning for efficiently restoring the OOD discriminative capability and maintaining performance on the original tasks.
>
> [1] Hendrycks, Dan, and Kevin Gimpel. "A Baseline for Detecting Misclassified and Out-of-Distribution Examples in Neural Networks." (2016).
>
> [2] Liu, Weitang, et al. "Energy-based out-of-distribution detection." Advances in Neural Information Processing Systems 33 (2020): 21464-21475.
>
> [3] Hendrycks, Dan, Mantas Mazeika, and Thomas Dietterich. "Deep Anomaly Detection with Outlier Exposure." International Conference on Learning Representations. 2018.
>
> [4] Ming, Yifei, Ying Fan, and Yixuan Li. "Poem: Out-of-distribution detection with posterior sampling." International Conference on Machine Learning. PMLR, 2022.
>
>
> > **Q3:** Labor-intensive: mask rate should be empirically found by UM users. Besides, we need to choose first the OOD baseline, second UM or UMAP to obtain the best result because no case consistently performs the best.
>
> **A3:** In Table 1, we present the part of experimental results that show the compatibility of UM and UMAP with either the fundamental scoring functions or outlier exposure with auxiliary data. In general, UM can restore the OOD detection capability based on the revealed inconsistency (between gaining OOD discriminative capability and pursuing classification performance on ID data). However, it may undermine the ID-ACC. Different from UM directly tuning on the original model, our proposed UM adopt pruning (UMAP) to achieve the same goal with the objective of forgetting. Extensive experiments (please kindly refer to Tables 4-8) show that UMAP can gain comparable or better OOD detection performance and not affect the ID-ACC of the original model, indicating that UMAP is generally a more compatible choice for usage.
>
> To visualize the effect of the parameter used in the experiments, we also provide the ablation study and present the results corresponding to different parameters used in key Eq.(3) in Figure 10 using WideResNet. The results show that a wide range of mask ratios (i.e., from 96% to 99%) to estimate the loss constraint used in Eq.(3) can gain better OOD detection performance than the baseline. It shows the mask ratio would be robust to hyper-parameter selection under a certain value. The principle intuition behind this is our revealed important observation as indicated in Figures 1(a) and 2(a)(b). With the guidance of the general mechanism, empirically choosing the hyper-parameter using the validation set is supportable and valuable for excavating better OOD detection capability of the model as conducted by previous literature [1,2,3].
>
> [1] Hendrycks, Dan, Mantas Mazeika, and Thomas Dietterich. "Deep Anomaly Detection with Outlier Exposure." International Conference on Learning Representations. 2018.
>
> [2] Liu, Weitang, et al. "Energy-based out-of-distribution detection." Advances in Neural Information Processing Systems 33 (2020): 21464-21475.
>
> [3] Sun, Yiyou, Chuan Guo, and Yixuan Li. "React: Out-of-distribution detection with rectified activations." Advances in Neural Information Processing Systems 34 (2021): 144-157.

---

> ### Author Response · Authors · 2022-11-15
> **Response to Reviewer Ga94 (1/4)**
>
>
> Thank you for your time devoted to reviewing this paper and your constructive suggestions. Here are our detailed replies to your questions.
>
> > **Q1:** Expensive: Because UM denoises a pretrained model via fine-tuning about 100 (I believe so based on several empirical results in the table although not specified for fine-tuning; authors need to clarify it) epochs, I don’t think it worths such expensive fine-tuning process.
>
> **A1:** Our UM adopts finetuning on the proposed objective for forgetting has shown the advantages of being cost-effective compared with train-from-scratch. For the tuning epochs, we show in Figures 9 and 10 that fine-tuning using UM can converge within about 20 epochs, which is far less than 100 epochs. As for the major experiments conducted in our work, finetuning adopts 100 epochs for better exploring and understanding its learning dynamics for research purposes, this configuration is indicated in the training details of Section 4.1.
>
> We also provide an extra comparison to show the relative efficiency of our proposed UM/UMAP in the following Table 1 and Table 2. The results show that UM and UMAP can efficiently restore detection performance compared with the baseline. It is intuitively reasonable that finetuning would benefit from the well-trained model, allowing a faster convergence as the two phases consider the same task and training data.
>
> **Table 1: Fine-tuning for 20 epochs with DenseNet-101**
>
> | D$_{in}$ | Epoch | Method | AUROC$\uparrow$ | AUPR$\uparrow$ | FPR95$\downarrow$ | ID-ACC$\uparrow$ |
> |----------|-------|------------------|-------|-------|-------|---------|
> | CIFAR-10 | 100   | MSP              | 89.90 | 91.48 | 60.08 | 94.01   |
> | CIFAR-10 | 100   | ODIN             | 91.46 | 91.67 | 42.31 | 94.01   |
> | CIFAR-10 | 100   | Energy           | 92.07 | 92.72 | 42.69 | 94.01   |
> | CIFAR-10 | 100   | Energy+UM        | 93.73 | 94.27 | 33.29 | 92.80   |
> | CIFAR-10 | 100   | Energy+UMAP      | 93.97 | 94.38 | 30.71 | 94.01   |
> | CIFAR-10 | **20**    | MSP+UM           | 90.31 | 91.99 | 53.61 | 91.70   |
> | CIFAR-10 | **20**    | ODIN+UM          | 94.08 | 94.67 | 31.01 | 91.70   |
> | CIFAR-10 | **20**    | Energy+UM        | 93.60 | 94.32 | 33.03 | 91.70   |
> | CIFAR-10 | **20**    | MSP+UMAP         | 88.70 | 90.39 | 57.69 | 94.01   |
> | CIFAR-10 | **20**    | ODIN+UMAP        | 92.88 | 93.33 | 35.19 | 94.01   |
> | CIFAR-10 | **20**    | Energy+UMAP      | 92.88 | 93.39 | 35.60 | 94.01   |
>
> **Table 2: Fine-tuning for 20 epochs with WRN-40-4**
>
> | D$_{in}$ | Epoch | Method | AUROC$\uparrow$ | AUPR$\uparrow$ | FPR95$\downarrow$ | ID-ACC$\uparrow$ |
> |----------|-------|------------------|-------|-------|-------|---------|
> | CIFAR-10 | 100   | MSP              | 87.12 | 87.84 | 68.29 | 93.86   |
> | CIFAR-10 | 100   | ODIN             | 83.29 | 82.74 | 65.68 | 93.86   |
> | CIFAR-10 | 100   | Energy           | 87.69 | 88.16 | 58.47 | 93.86   |
> | CIFAR-10 | 100   | Energy+UM        | 91.74 | 92.67 | 40.40 | 92.68   |
> | CIFAR-10 | 100   | Energy+UMAP      | 88.84 | 89.31 | 50.23 | 93.86   |
> | CIFAR-10 | **20**    | MSP+UM           | 89.86 | 91.32 | 51.62 | 91.96   |
> | CIFAR-10 | **20**    | ODIN+UM          | 91.97 | 92.58 | 41.78 | 91.96   |
> | CIFAR-10 | **20**    | Energy+UM        | 92.95 | 93.64 | 36.21 | 91.96   |
> | CIFAR-10 | **20**    | MSP+UMAP         | 88.77 | 90.61 | 61.60 | 93.86   |
> | CIFAR-10 | **20**    | ODIN+UMAP        | 90.85 | 91.89 | 45.70 | 93.86   |
> | CIFAR-10 | **20**    | Energy+UMAP      | 91.66 | 92.49 | 42.94 | 93.86   |
>
>
> Considering the significance of the OOD awareness for those safety-critical areas, it is worthwhile to further excavate the OOD detection capability of the deployed well-trained model using our UM and UMAP. We will add the comparison and corresponding discussion in our appendix.

---

> ### Author Response · Authors · 2022-11-17
> **Would you mind confirming if you have further questions? Thanks!**
>
> Dear Reviewer Ga94,
>
> We appreciate your efforts and time in reviewing our paper! We have carefully considered your concerns/questions and provided as much refinement and experiments to address them regarding the submission. Would you mind checking our response, and confirming if you have further questions?
>
> Best,
>
> Authors

---

> ### Author Response · Authors · 2022-11-18
> **Would you mind checking our response? Welcome for more discussions.**
>
> Dear Reviewer Ga94,
>
> Thanks again for your time and efforts in reviewing our paper. As the end of Discussion Stage 1 is approaching and the window for paper revision is closing, here is a summary of our previous response and update:
>
> - Clarified the cost of the proposed methods (see Appendix F).
> - Clarified how we choose UM/UMAP (see Appendix F).
> - Clarified the terminology to describe the proposed algorithms (see Appendix D).
> - Add more explanation of some acronym words (see revision and Appendix A).
> - Revised more ambiguous words to avoid confusion (see revision).
>
> **We humbly expect you could check our detailed responses with our updated version, and confirm whether our response has addressed your concerns. More discussions are always welcome. Please let us know if there are any further questions or suggestions that we could clarify or improve.**
>
> Best,
>
> Authors

---

> ### Author Response · Authors · 2022-11-24
> **Need further clarification? Looking forward to your responses or further suggestions/comments!**
>
> Dear Reviewer Ga94,
>
> We appreciate your efforts and time in reviewing our paper! We have carefully considered and addressed your initial concerns regarding our submission. Would you mind checking our response, and confirming if you have further questions? If any, we will address them to improve our submission.
>
> Best,
>
> Authors of Paper1069

---

> ### Author Response · Authors · 2022-11-25
> **Need further clarification? Looking forward to your responses or further suggestions/comments!**
>
> Dear Reviewer Ga94,
>
> We appreciate your efforts and time in reviewing our paper! We have carefully considered and addressed your initial concerns regarding our submission. Would you mind checking our response, and confirming if you have further questions? If any, we will address them to improve our submission.
>
> Best,
>
> Authors of Paper1069

---

> ### Author Response · Authors · 2022-11-26
> **Need further clarification? Looking forward to your responses or further suggestions/comments!**
>
> Dear Reviewer Ga94,
>
> We appreciate your efforts and time in reviewing our paper! We have carefully considered and addressed your initial concerns regarding our submission. Would you mind checking our response, and confirming if you have further questions? If any, we will address them to improve our submission.
>
> Best,
>
> Authors of Paper1069

---

> ### Author Response · Authors · 2022-11-27
> **Need further clarification? Looking forward to your responses or further suggestions/comments!**
>
> Dear Reviewer Ga94,
>
> We appreciate your efforts and time in reviewing our paper! We have carefully considered and addressed your initial concerns regarding our submission. Would you mind checking our response, and confirming if you have further questions? If any, we will address them to improve our submission.
>
> Best,
>
> Authors of Paper1069

---

> ### Author Response · Authors · 2022-11-30
> **Need further clarification? Looking forward to your responses or further suggestions/comments!**
>
> Dear Reviewer Ga94,
>
> We appreciate your efforts and time in reviewing our paper! We have carefully considered and addressed your initial concerns regarding our submission.
>
> We understand you might be quite busy. However, as the discussion deadline is approaching, would you mind checking our response and confirming whether you have any further questions?
>
> Best,
>
> Authors of Paper1069

---

> ### Author Response · Authors · 2022-12-02
> **Would you mind confirming if you have further questions? Thanks!**
>
> Dear Reviewer Ga94,
>
> We appreciate your efforts and time in reviewing our paper! We have carefully considered and addressed your initial concerns/questions regarding our submission.
>
> We understand you might be quite busy. However, as the discussion deadline is approaching and we have not received any feedback from you yet since the discussion phase began, would you mind checking our response (e.g., A1-A12 in the [details](https://openreview.net/forum?id=K2OixmPDou3&noteId=3WLz7Ok6ZA), and [revision summary](https://openreview.net/forum?id=K2OixmPDou3&noteId=Cy5l4oeMb6n)) and confirming whether you have any further questions?
>
> If you have any more questions, we are happy to discuss them in the openreview system and will do our best to address them!
>
> Best,
>
> Authors of Paper1069

---

> ### Author Response · Authors · 2022-12-04
> **As the discussion phase is near the end, would you mind confirming if you have further questions?**
>
> Dear Reviewer Ga94,
>
> We appreciate your efforts and time in reviewing our paper! We have carefully considered and addressed your initial concerns/questions regarding our submission.
>
> We understand you might be quite busy. However, as the discussion deadline is approaching and we have not received any feedback from you yet since the discussion phase began, would you mind checking our response (e.g., A1-A12 in the [[details]](https://openreview.net/forum?id=K2OixmPDou3&noteId=3WLz7Ok6ZA), and [[revision summary]](https://openreview.net/forum?id=K2OixmPDou3&noteId=Cy5l4oeMb6n)) and confirming whether you have any further questions?
>
> If you have any more questions, we are happy to discuss them in the openreview system and will do our best to address them!
>
> Best,
>
> Authors of Paper1069

---

> ### Author Response · Authors · 2022-12-06
> **As the discussion phase is near the end, would you mind confirming if you have further questions?**
>
> Dear Reviewer Ga94,
>
> We appreciate your efforts and time in reviewing our paper! We have carefully considered and addressed your initial concerns regarding our submission.
>
> We understand you might be quite busy. However, as the discussion deadline is approaching and we have not received any feedback from you yet since the discussion phase began, would you mind checking our response (e.g., A1-A12 in the [[details]](https://openreview.net/forum?id=K2OixmPDou3&noteId=3WLz7Ok6ZA), and [[revision summary]](https://openreview.net/forum?id=K2OixmPDou3&noteId=Cy5l4oeMb6n)) and confirming whether you have any further questions?
>
> If you have any more questions, we are happy to discuss at the openreview system and will do our best to address them!
>
> Best,
>
> Authors of Paper1069

---

### Author Response · Authors · 2022-11-16
**General response -- thanks to all reviewers for the valuable feedback**


We are very glad to see that the reviewers find our observation **interesting**, **exciting**, and **important** (R3, R4), and the methods are **reasonable for identifying and forgetting the atypical samples** (R3, R4), and also have good **adaptability** (R1), and the experiments are **comprehensive** and can **support our methods well** (R2, R3, R4). We are also pleased that the reviewers find the description of setups **well-detailed for reproducibility** (R4) and the paper is **well-written** (R3).

We have addressed the reviewers' comments and concerns in **individual responses to each reviewer**. The reviews allowed us to improve our draft and the changes made in the revised draft are summarized below:

From Reviewer Ga94

- Clarified the cost of the proposed methods (see Appendix F).
- Clarified how we choose UM/UMAP (see Appendix F).
- Clarified the terminology to describe the proposed algorithms (see Appendix D).
- Add the explanation of some acronym words (see revision and Appendix A).
- Revised some ambiguous words to avoid confusion (see revision).


From Reviewer PAPe

- Added supportive experiments to tamp the association between our motivation and the proposed methods (see Appendix F).
- Expanded discussion about the hyper-parameter mask rate (see Appendix F).
- Clarified the arrangement of experiments of CIFAR-100 (see Section 4.2).
- Corrected some clarity issues (see revision).


From Reviewer miMq

- Clarified the description of the layer-wise mask (see Section 3.2).
- Expanded discussion and visualization about mining the atypical samples with our mask (see Appendix B).
- Expanded the discussion on overfitting and added a comparison between overfitting methods and the proposed methods (see Appendix C).
- Added supportive experiments to show atypical samples can indeed undermine OOD performance (see Appendix F).


From Reviewer 3ovB

- Expanded discussion on the difference between overfitting and the proposed methods and added comparison between overfitting and UM/UMAP (see Appendix C).
- Clarified the relationship between the observation and the proposed methods (see Appendix F).
- Added supportive experiments to support our motivation (see Appendix F).
- Expanded the description of UMAP (see Section 3.3).
- Correct the typo in Equation 3 (see Section 3.3).

**We appreciate your comments and time!** We have tried our best to address your concerns and revised the paper following the suggestions. **Would you mind checking it and confirming if you have further questions?**

---

### Decision · Program_Chairs · 2023-01-20

**Decision:**

Reject

**Justification For Why Not Higher Score:**

The paper is a good submission overall, and the writing clarity is great. My main reservation is about the thoroughness of experiments (mainly CIFAR-level) and the lack of ablation studies on key hyperparameters. However, I see this paper as very promising and should become a good accept case in the next major ML venue, after integrating their generated new results and revising accordingly

I also feel sorry for the authors that their reviewers were not sufficiently engaged in the discussion.

**Justification For Why Not Lower Score:**

N/A

**Metareview: Summary, Strengths And Weaknesses:**

The paper proposes Unleashing Mask (UM) with the goal of maintaining a model's capability to distinguish between out-of-distribution (OOD) and in-distribution (ID) training examples. The key observation is that there exists a historical training stage where the model has a higher OOD detection performance than the final well-trained, one that is very interesting and intuitive. while the atypical example memorization near the end of training improves model accuracy at the main target task, it harms the model's OOD detection performance. The authors hence specify a binary layer-wise mask that serves to force the forgetting of atypical examples.

The paper received highly divergent scores. Unfortunately, the reviewers were generally inactive in engaging in post-rebuttal discussions despite multiple reminders (including AC's direct email reminders to some of them). AC hence reads this paper carefully on his/her own.

First, AC sides with the authors that the comments of Reviewer Ga94 do not faithfully reflect this work's technical quality. Two important misunderstandings are spotted: (1) Reviewer Ga94 kept mentioning and comparing noisy label learning literature with this work, which is out of scope; (2) Reviewer Ga94 thought noisy/OoD data were used in training, which is factually wrong. Hence, it is suspected that Reviewer Ga94 put down his/her comments with a relatively rushed reading; and hence his/her score was weighted down towards the final decision.

Second, AC shares the concern with Reviewer PAPe that the key parameters of mask ratio might become a hidden issue, that was under-discussed in the current draft. The authors indeed reported an ablation study in Figure 10, but restricted to one setting of WideResNet-40-4 + CIFAR-10. Moreover, WideResNet-40-4 is clearly too much parameterized for CIFAR-10, and hence high sparsity masks  (i.e., from 96% to 99%)  unsurprisingly work in general. However, if the dataset scales up and the model is no longer "overparameterized" this much, e.g., switching to CIFAR to ImageNet as the main target set, I am uncertain whether this hyperparameter insensitivity remains a true statement (i.e., I bet 99% mask will at least affect performance). Unfortunately, all experiments in this paper adopt CIFAR-10/CIFAR-100 as the main target experiments and ImageNet was only used as validation or auxiliary OoD data.

Third, AC appreciates the authors providing additional experiments in quantitatively characterizing the negative effect of learning on those atypical samples by comparing with a counterpart that learning only with the typical sample: https://openreview.net/forum?id=K2OixmPDou3&noteId=5KrglufT1YA those should definitely become part of, or mentioned by your next version main text. Would be helpful in clarifying your motivation better given that was also not theory-backed.